# Global Carbon Budget 2016
Significant text differences from the Global Carbon Budget 2015 are shown in red
Corinne Le Quéré[1], Robbie M. Andrew[2], Josep G. Canadell[3], Stephen Sitch[4], Jan Ivar Korsbakken[2], Glen P.
Peters[2], Andrew C. Manning[5], Thomas A. Boden[6], Pieter P. Tans[7], Richard A. Houghton[8], Ralph F. Keeling[9],
Simone Alin[10], Oliver D. Andrews[1], Peter Anthoni[11], Leticia Barbero[12], Laurent Bopp[13], Frédéric Chevallier[13],
Louise P. Chini[14], Philippe Ciais[13], Kim Currie[15], Christine Delire[16], Scott C. Doney[17], Pierre Friedlingstein[18],
Thanos Gkritzalis[19], Ian Harris[20], Judith Hauck[21], Vanessa Haverd[22], Mario Hoppema[21], Kees Klein
Goldewijk[23], Atul K. Jain[24], Etsushi Kato[25], Arne Körtzinger[26], Peter Landschützer[27], Nathalie Lefèvre[28],
Andrew Lenton[28], Sebastian Lienert[30], Danica Lombardozzi[31], Joe R. Melton[31], Nicolas Metzl[28], Frank
Millero[33], Pedro M. S. Monteiro[34], David R. Munro[35], Julia E. M. S. Nabel[27], Shin-ichiro Nakaoka[36], Kevin
O'Brien[37], Are Olsen[38,39], Abdirahman M. Omar[38,39], Tsuneo Ono[41], Denis Pierrot[12], Benjamin Poulter[43,43],
Christian Rödenbeck[44], Joe Salisbury[45], Ute Schuster[4], Jörg Schwinger[38,39], Roland Séférian[16], Ingunn
Skjelvan[39,40], Benjamin D. Stocker[46,47], Adrienne J. Sutton[37,10], Taro Takahashi[48], Hanqin Tian[49], Bronte
Tilbrook[50], Ingrid T. van der Laan-Luijkx[51], Guido R. van der Werf[52], Nicolas Viovy[13], Anthony P. Walker[53],
Andrew J. Wiltshire[54], Sönke Zaehle[44]
[1]Tyndall Centre for Climate Change Research, University of East Anglia, Norwich Research Park, Norwich
NR4 7TJ, UK
[2]Center for International Climate and Environmental Research – Oslo (CICERO), Norway
[3]Global Carbon Project, CSIRO Oceans and Atmosphere, GPO Box 3023, Canberra, ACT 2601, Australia
[4]College of Life and Environmental Sciences, University of Exeter, Exeter EX4 4RJ, UK
[5]Centre for Ocean and Atmospheric Sciences, School of Environmental Sciences, University of East Anglia,
Norwich Research Park, Norwich NR4 7TJ, UK
[6]Carbon Dioxide Information Analysis Center (CDIAC), Oak Ridge National Laboratory, Oak Ridge, TN, USA
[7]National Oceanic & Atmospheric Administration, Earth System Research Laboratory (NOAA/ESRL),
Boulder, CO 80305, USA
[8]Woods Hole Research Centre (WHRC), Falmouth, MA 02540, USA
[9]University of California, San Diego, Scripps Institution of Oceanography, La Jolla, CA 92093-0244, USA
[10]National Oceanic & Atmospheric Administration/Pacific Marine Environmental Laboratory (NOAA/PMEL),
7600 Sand Point Way NE, Seattle, WA 98115, USA
[11]Karlsruhe Institute of Technology, Institute of Meteorology and Climate Research/Atmospheric
Environmental Research, 82467 Garmisch-Partenkirchen, Germany
[12]National Oceanic & Atmospheric Administration/Atlantic Oceanographic & Meteorological Laboratory
(NOAA/AOML), Miami, FL 33149, USA
[13]Laboratoire des Sciences du Climat et de l'Environnement, Institut Pierre-Simon Laplace, CEA-CNRS-
UVSQ, CE Orme des Merisiers, 91191 Gif sur Yvette Cedex, France
[14]Department of Geographical Sciences, University of Maryland, College Park, Maryland 20742, USA
[15]National Institute of Water and Atmospheric Research (NIWA), Dunedin 9054, New Zealand
[16]Centre National de Recherche Météorologique, Unite mixte de recherche 3589 Météo-France/CNRS, 42
Avenue Gaspard Coriolis, 31100 Toulouse, France
[17]Woods Hole Oceanographic Institution (WHOI), Woods Hole, MA 02543, USA
[18]College of Engineering, Mathematics and Physical Sciences, University of Exeter, Exeter EX4 4QF, UK
[19]Flanders Marine Institute, InnovOcean, Wandelaarkaai 7, 8400 Ostend, Belgium
[20]Climatic Research Unit, University of East Anglia, Norwich Research Park, Norwich, NR4 7TJ, UK
[21]Alfred Wegener Institute Helmholtz Centre for Polar and Marine Research, Postfach 120161, 27515
Bremerhaven, Germany
[22]CSIRO Oceans and Atmosphere, GPO Box 1700, Canberra, ACT 2601, Australia

[23]PBL Netherlands Environmental Assessment Agency, The Hague/Bilthoven and Utrecht University, Utrecht, The Netherlands

[24]Department of Atmospheric Sciences, University of Illinois, Urbana, IL 61821, USA

[25]Institute of Applied Energy (IAE), Minato-ku, Tokyo 105-0003, Japan

[26]GEOMAR Helmholtz Centre for Ocean Research Kiel, Düsternbrooker Weg 20, 24105 Kiel, Germany

[27]Max Planck Institute for Meteorology, Bundesstr. 53, 20146 Hamburg, Germany

[28]Sorbonne Universités (UPMC, Univ Paris 06), CNRS, IRD, MNHN, LOCEAN/IPSL Laboratory, 75252 Paris, France

[29]CSIRO Oceans and Atmosphere, PO Box 1538, Hobart, Tasmania, Australia

[30]Institute of Climate and Environmental Physics and Oeschger Centre for Climate Change Research, University of Bern, Bern, Switzerland

[31]National Center for Atmospheric Research, Climate and Global Dynamics, Terrestrial Sciences Section, Boulder, CO 80305, USA

[32]Climate Research Division, Environment and Climate Change Canada, Victoria, Canada

[33]Department of Ocean Sciences, RSMAS/MAC, University of Miami, 4600 Rickenbacker Causeway, Miami, FL 33149, USA

[34]Ocean Systems and Climate, CSIR-CHPC, Cape Town, 7700, South Africa

[35]Department of Atmospheric and Oceanic Sciences and Institute of Arctic and Alpine Research, University of Colorado, Campus Box 450, Boulder, CO 80309-0450, USA

[36]Center for Global Environmental Research, National Institute for Environmental Studies (NIES), 16-2 Onogawa, Tsukuba, Ibaraki 305-8506, Japan

[37]Joint Institute for the Study of the Atmosphere and Ocean, University of Washington, Seattle, WA 98195, USA

[38]Geophysical Institute, University of Bergen, Allégaten 70, 5007 Bergen, Norway

[39]Bjerknes Centre for Climate Research, Allégaten 70, 5007 Bergen, Norway

[40]Uni Climate - Uni Research AS, Allégaten 55, 5007 Bergen, Norway

[41]National Research Institute for Far Sea Fisheries, Japan Fisheries Research and Education Agency 2-12-4 Fukuura, Kanazawa-Ku, Yokohama 236-8648, Japan

[42]NASA Goddard Space Flight Center, Biospheric Science Laboratory, Greenbelt, Maryland 20771, USA

[43]Department of Ecology, Montana State University, Bozeman, MT 59717, USA

[44]Max Planck Institute for Biogeochemistry, P.O. Box 600164, Hans-Knöll-Str. 10, 07745 Jena, Germany

[45]University of New Hampshire, Ocean Process Analysis Laboratory, 161 Morse Hall, 8 College Road, Durham, NH 03824, USA

[46]Climate and Environmental Physics, and Oeschger Centre for Climate Change Research, University of Bern, Bern, Switzerland

[47]Imperial College London, Life Science Department, Silwood Park, Ascot, Berkshire SL5 7PY, UK

[48]Lamont-Doherty Earth Observatory of Columbia University, Palisades, NY 10964, USA

[49]School of Forestry and Wildlife Sciences, Auburn University, 602 Ducan Drive, Auburn, AL 36849, USA

[50]CSIRO Oceans and Atmosphere and Antarctic Climate and Ecosystems Co-operative Research Centre, Hobart, Tasmania, Australia

[51]Department of Meteorology and Air Quality, Wageningen University & Research, PO Box 47 6700AA Wageningen, The Netherlands

[52]Faculty of Earth and Life Sciences, VU University Amsterdam, The Netherlands

[53]Environmental Sciences Division & Climate Change Science Institute, Oak Ridge National Laboratory, Oak Ridge, Tennessee, USA

[54]Met Office Hadley Centre, FitzRoy Road, Exeter EX1 3PB, UK

**Abstract**

Accurate assessment of anthropogenic carbon dioxide ($CO_2$) emissions and their redistribution among the atmosphere, ocean, and terrestrial biosphere – the 'global carbon budget' – is

important to better understand the global carbon cycle, support the development of climate
policies, and project future climate change. Here we describe data sets and methodology to
quantify all major components of the global carbon budget, including their uncertainties, based on
the combination of a range of data, algorithms, statistics and model estimates and their
interpretation by a broad scientific community. We discuss changes compared to previous
estimates, consistency within and among components, alongside methodology and data
limitations. $CO_2$ emissions from fossil fuels and industry ($E_{FF}$) are based on energy statistics and
cement production data, respectively, while emissions from land-use change ($E_{LUC}$), mainly
deforestation, are based on combined evidence from land-cover change data, fire activity
associated with deforestation, and models. The global atmospheric $CO_2$ concentration is
measured directly and its rate of growth ($G_{ATM}$) is computed from the annual changes in
concentration. The mean ocean $CO_2$ sink ($S_{OCEAN}$) is based on observations from the 1990s, while
the annual anomalies and trends are estimated with ocean models. The variability in $S_{OCEAN}$ is
evaluated with data products based on surveys of ocean $CO_2$ measurements. The global residual
terrestrial $CO_2$ sink ($S_{LAND}$) is estimated by the difference of the other terms of the global carbon
budget and compared to results of independent Dynamic Global Vegetation Models. We compare
the mean land and ocean fluxes and their variability to estimates from three atmospheric inverse
methods for three broad latitude bands. All uncertainties are reported as ±1σ, reflecting the
current capacity to characterise the annual estimates of each component of the global carbon
budget. For the last decade available (2006-2015), $E_{FF}$ was 9.3 ± 0.5 GtC yr$^{-1}$, $E_{LUC}$ 1.0 ± 0.5 GtC yr$^{-1}$,
$G_{ATM}$ 4.5 ± 0.1 GtC yr$^{-1}$, $S_{OCEAN}$ 2.6 ± 0.5 GtC yr$^{-1}$, and $S_{LAND}$ 3.2 ± 0.8 GtC yr$^{-1}$. For year 2015 alone,
the growth in $E_{FF}$ was approximately zero and emissions remained at 9.9 ± 0.5 GtC yr$^{-1}$, showing a
slowdown in growth of these emissions compared to the average growth of 1.8 % yr$^{-1}$ that took
place during 2006-2015. Also for 2015, $E_{LUC}$ was 1.3 ± 0.5 GtC yr$^{-1}$, $G_{ATM}$ was 6.2 ± 0.2 GtC yr$^{-1}$,
$S_{OCEAN}$ was 3.0 ± 0.5 GtC yr$^{-1}$ and $S_{LAND}$ was 2.0 ± 0.9 GtC yr$^{-1}$. $G_{ATM}$ was higher in 2015 compared to
the past decade (2006-2015), reflecting a smaller $S_{LAND}$ for that year. The global atmospheric $CO_2$
concentration reached 399.4 ± 0.1 ppm averaged over 2015. For 2016, preliminary data indicate
that the growth in $E_{FF}$ will be approximately zero based on national emissions projections for
China and USA, and projections of Gross Domestic Product corrected for recent changes in the
carbon intensity of the economy for the rest of the world. In spite of an unchanged $E_{FF}$ in 2016,
the growth rate in atmospheric $CO_2$ concentration is expected to be near record-high because of
the smaller residual terrestrial sink ($S_{LAND}$) in response to El Niño conditions in 2015-2016. From
this projection of $E_{FF}$ and assumed constant $E_{LUC}$ for 2016, cumulative emissions of $CO_2$ will reach
$570 \pm 55$ GtC ($2085 \pm 205$ GtCO$_2$) for 1870-2016, about 75% from $E_{FF}$ and 25% from $E_{LUC.}$ This living
data update documents changes in the methods and data sets used in this new carbon budget
compared with previous publications of this data set (Le Quéré et al., 2015b; 2015a; 2014; 2013).
All observations presented here can be downloaded from the Carbon Dioxide Information Analysis
Center (doi: 10.3334/CDIAC/GCP_2016).
**1    Introduction**
The concentration of carbon dioxide ($CO_2$) in the atmosphere has increased from approximately
277 parts per million (ppm) in 1750 (Joos and Spahni, 2008), the beginning of the Industrial Era, to
$399.4 \pm 0.1$  ppm in 2015 (Dlugokencky and Tans, 2016). The Mauna Loa station, which holds the
longest running record of direct measurements of atmospheric $CO_2$ concentration (Tans and
Keeling, 2014), went above 400 ppm for the first time in May 2013 (Scripps, 2013). The global
monthly average concentration was above 400 ppm in March through May 2015 and again since
November 2015 (Dlugokencky and Tans, 2016; Fig. 1). The atmospheric $CO_2$ increase above
preindustrial levels was, initially, primarily caused by the release of carbon to the atmosphere
from deforestation and other land-use change activities (Ciais et al., 2013). While emissions from
fossil fuels started before the Industrial Era, they only became the dominant source of
anthropogenic emissions to the atmosphere from around 1920 and their relative share has
continued to increase until present. Anthropogenic emissions occur on top of an active natural
carbon cycle that circulates carbon between the reservoirs of the atmosphere, ocean, and
terrestrial biosphere on time scales from sub-daily to millennia, while exchanges with geologic
reservoirs occur at longer timescales (Archer et al., 2009).
The global carbon budget presented here refers to the mean, variations, and trends in the
perturbation of $CO_2$ in the atmosphere, referenced to the beginning of the Industrial Era. It
quantifies the input of $CO_2$ to the atmosphere by emissions from human activities, the growth rate
of atmospheric $CO_2$ concentration, and the resulting changes in the storage of carbon in the land
and ocean reservoirs in response to increasing atmospheric $CO_2$ levels, climate and variability, and
other anthropogenic and natural changes (Fig. 2). An understanding of this perturbation budget
over time and the underlying variability and trends of the natural carbon cycle are necessary to
understand the response of natural sinks to changes in climate, $CO_2$ and land-use change drivers,
and the permissible emissions for a given climate stabilization target.
The components of the $CO_2$ budget that are reported annually in this paper include separate
estimates for (1) the $CO_2$ emissions from fossil fuel combustion and oxidation and cement
production ($E_{FF}$; GtC yr$^{-1}$), (2) the $CO_2$ emissions resulting from deliberate human activities on land
leading to land-use change ($E_{LUC}$; GtC yr$^{-1}$), (3) the growth rate of atmospheric $CO_2$ concentration
($G_{ATM}$; GtC yr$^{-1}$), and the uptake of $CO_2$ by the 'CO$_2$ sinks' in (4) the ocean ($S_{OCEAN}$; GtC yr$^{-1}$) and (5)
on land ($S_{LAND}$; GtC yr$^{-1}$). The $CO_2$ sinks as defined here include the response of the land and ocean
to elevated $CO_2$ and changes in climate and other environmental conditions. The global emissions
and their partitioning among the atmosphere, ocean and land are in balance:

$$E_{FF} + E_{LUC} = G_{ATM} + S_{OCEAN} + S_{LAND}. \qquad (1)$$

$G_{ATM}$ is usually reported in ppm yr$^{-1}$, which we convert to units of carbon mass, GtC yr$^{-1}$, using 1
ppm = 2.12 GtC (Ballantyne et al., 2012; Prather et al., 2012; Table 1). We also include a
quantification of $E_{FF}$ by country, computed with both territorial and consumption based
accounting (see Methods).
Equation (1) partly omits two kinds of processes. The first is the net input of $CO_2$ to the
atmosphere from the chemical oxidation of reactive carbon-containing gases from sources other
than the combustion of fossil fuels (e.g. fugitive anthropogenic $CH_4$ emissions, industrial
processes, and biogenic emissions from changes in vegetation, fires, wetlands, etc.), primarily
methane ($CH_4$), carbon monoxide (CO), and volatile organic compounds such as isoprene and
terpene (Gonzalez-Gaya et al., 2016). CO emissions are currently implicit in $E_{FF}$ while fugitive
anthropogenic $CH_4$ emissions are not and thus their inclusion would result in a small increase in
$E_{FF}$. The second is the anthropogenic perturbation to carbon cycling in terrestrial freshwaters,
estuaries, and coastal areas, that modifies lateral fluxes from land ecosystems to the open ocean,
the evasion $CO_2$ flux from rivers, lakes and estuaries to the atmosphere, and the net air-sea
anthropogenic $CO_2$ flux of coastal areas (Regnier et al., 2013). The inclusion of freshwater fluxes of
anthropogenic $CO_2$ would affect the estimates of, and partitioning between, $S_{LAND}$ and $S_{OCEAN}$ in Eq.
(1) in complementary ways, but would not affect the other terms. These flows are omitted in the
absence of annual information on the natural versus anthropogenic perturbation terms of these
loops of the carbon cycle, and they are discussed in Section 2.7.
The $CO_2$ budget has been assessed by the Intergovernmental Panel on Climate Change (IPCC) in all
assessment reports (Ciais et al., 2013; Denman et al., 2007; Prentice et al., 2001; Schimel et al.,
1995; Watson et al., 1990), and by others (e.g. Ballantyne et al., 2012). These assessments
included budget estimates for the decades of the 1980s, 1990s (Denman et al., 2007) and, most
recently, the period 2002-2011 (Ciais et al., 2013). The IPCC methodology has been adapted and
used by the Global Carbon Project (GCP, www.globalcarbonproject.org), which has coordinated a
cooperative community effort for the annual publication of global carbon budgets up to year 2005
(Raupach et al., 2007; including fossil emissions only), year 2006 (Canadell et al., 2007), year 2007
(published online;  GCP, 2007), year 2008 (Le Quéré et al., 2009), year 2009 (Friedlingstein et al.,
2010), year 2010 (Peters et al., 2012b), year 2012 (Le Quéré et al., 2013; Peters et al., 2013), year
2013 (Le Quéré et al., 2014), year 2014 (Friedlingstein et al., 2014; Le Quéré et al., 2015b), and
most recently year 2015 (Jackson et al., 2016; Le Quéré et al., 2015a). Each of these papers
updated previous estimates with the latest available information for the entire time series. From
2008, these publications projected fossil fuel emissions for one additional year.
We adopt a range of ±1 standard deviation ($\sigma$) to report the uncertainties in our estimates,
representing a likelihood of 68% that the true value will be within the provided range if the errors
have a Gaussian distribution. This choice reflects the difficulty of characterising the uncertainty in
the $CO_2$ fluxes between the atmosphere and the ocean and land reservoirs individually,
particularly on an annual basis, as well as the difficulty of updating the $CO_2$ emissions from land-
use change. A likelihood of 68% provides an indication of our current capability to quantify each
term and its uncertainty given the available information. For comparison, the Fifth Assessment
Report of the IPCC (AR5) generally reported a likelihood of 90% for large data sets whose
uncertainty is well characterised, or for long time intervals less affected by year-to-year variability.
Our 68% uncertainty value is near the 66% which the IPCC characterises as 'likely' for values falling
into the ±1$\sigma$ interval. The uncertainties reported here combine statistical analysis of the
underlying data and expert judgement of the likelihood of results lying outside this range. The
limitations of current information are discussed in the paper and have been examined in detail
elsewhere (Ballantyne et al., 2015).
All quantities are presented in units of gigatonnes of carbon (GtC, $10^{15}$ gC), which is the same as
petagrams of carbon (PgC; Table 1). Units of gigatonnes of $CO_2$ (or billion tonnes of $CO_2$) used in
policy are equal to 3.664 multiplied by the value in units of GtC.
This paper provides a detailed description of the data sets and methodology used to compute the
global carbon budget estimates for the period preindustrial (1750) to 2015 and in more detail for
the period 1959 to 2015. We also provide decadal averages starting in 1960 including the last
decade (2006-2015), results for the year 2015, and a projection of $E_{FF}$ for year 2016. Finally we
provide cumulative emissions from fossil fuels and land-use change since year 1750, the
preindustrial period, and since year 1870, the reference year for the cumulative carbon estimate
used by the IPCC (AR5) based on the availability of global temperature data (Stocker et al., 2013).
This paper will be updated every year using the format of 'living data' to keep a record of budget
versions and the changes in new data, revision of data, and changes in methodology that lead to
changes in estimates of the carbon budget. Additional materials associated with the release of
each new version will be posted at the Global Carbon Project (GCP) website
(http://www.globalcarbonproject.org/carbonbudget), with fossil fuel emissions also available
through the Global Carbon Atlas (http://www.globalcarbonatlas.org). With this approach, we aim
to provide the highest transparency and traceability in the reporting of $CO_2$, the key driver of
climate change.
**2    Methods**
Multiple organizations and research groups around the world generated the original
measurements and data used to complete the global carbon budget. The effort presented here is
thus mainly one of synthesis, where results from individual groups are collated, analysed and
evaluated for consistency. We facilitate access to original data with the understanding that
primary data sets will be referenced in future work (See Table 2 for 'How to cite' the data sets).
Descriptions of the measurements, models, and methodologies follow below and in depth
descriptions of each component are described elsewhere.
This is the 11[th] version of the global carbon budget and the fifth revised version in the format of a
living data update. It builds on the latest published global carbon budget of Le Quéré et al.
(2015a). The main changes are: (1) the inclusion of data to year 2015 (inclusive) and a projection
for fossil fuel emissions for year 2016; (2) the introduction of a projection for the full carbon
budget for year 2016 using our fossil fuel projection, combined with preliminary data (Keeling et
al., 2016) and analysis by others (Betts et al., 2016) of the growth rate in atmospheric $CO_2$
concentration; and (3) the use of BP data from 1990  (BP, 2016b) to estimate emissions in China to
ensure all recent revisions in Chinese statistics are incorporated. The main methodological
differences between annual carbon budgets are summarised in Table 3.
**2.1  $CO_2$ emissions from fossil fuels and industry ($E_{FF}$)**
**2.1.1  Emissions from fossil fuels and industry and their uncertainty**
The calculation of global and national $CO_2$ emissions from fossil fuels, including gas flaring and
cement production ($E_{FF}$), relies primarily on energy consumption data, specifically data on
hydrocarbon fuels, collated and archived by several organisations (Andres et al., 2012). These
include the Carbon Dioxide Information Analysis Center (CDIAC), the International Energy Agency
(IEA), the United Nations (UN), the United States Department of Energy (DoE) Energy Information
Administration (EIA), and more recently also the Planbureau voor de Leefomgeving (PBL)
Netherlands Environmental Assessment Agency.  Where available, we use national emissions
estimated by the countries themselves and reported to the UNFCCC for the period 1990-2014 (40
countries). We assume that national emissions reported to the UNFCCC are the most accurate
because national experts have access to additional and country-specific information, and because
these emission estimates are periodically audited for each country through an established
international methodology overseen by the UNFCCC. We also use global and national emissions
estimated by CDIAC (Boden and Andres, 2016). The CDIAC emission estimates are the only data
set that extends back in time to 1751 with consistent and well-documented emissions from fossil
fuels, cement production, and gas flaring for all countries and their uncertainty (Andres et al.,
2014; Andres et al., 2012; Andres et al., 1999); this makes the data set a unique resource for
research of the carbon cycle during the fossil fuel era.
The global emissions presented here are from CDIAC's analysis, which provides an internally-
consistent global estimate including bunker fuels, minimising the effects of lower-quality energy
trade data. Thus the comparison of global emissions with previous annual carbon budgets is not
influenced by the use of national data from UNFCCC reports.
During the period 1959-2013, the emissions from fossil fuels estimated by CDIAC are based
primarily on energy data provided by the UN Statistics Division (UN, 2014a; b; Table 4). When
necessary, fuel masses/volumes are converted to fuel energy content using coefficients provided
by the UN and then to $CO_2$ emissions using conversion factors that take into account the
relationship between carbon content and energy (heat) content of the different fuel types (coal,
oil, gas, gas flaring) and the combustion efficiency (to account, for example, for soot left in the
combustor or fuel otherwise lost or discharged without oxidation). Most data on energy
consumption and fuel quality (carbon content and heat content) are available at the country level
(UN, 2014a). In general, $CO_2$ emissions for equivalent primary energy consumption are about 30%
higher for coal compared to oil, and 70% higher for coal compared to natural gas (Marland et al.,

3    2007).

Recent revisions in energy data for China (Korsbakken et al., 2016) have not yet fully propagated
to the UN energy statistics used by CDIAC, but are available through the BP energy statistics (BP,
2016b). We thus use the BP energy statistics (BP, 2016b) and estimate the emissions by fuel type
using the BP methodology (BP, 2016a) to be consistent with the format of the CDIAC data.
Emissions in China calculated from the BP statistics differ from those provided by CDIAC emissions
mostly between 1997 and 2009. We propagate these new estimates for China through to the
global total to ensure consistency.
Our emission totals for the UNFCCC-reporting countries were recorded as in the UNFCCC
submissions, which have a slightly larger system boundary than CDIAC. Additional emissions come
from carbonates other than in cement manufacture, and thus UNFCCC totals will be to be slightly
higher than CDIAC totals in general, although there are multiple sources of differences. We use
the CDIAC method to report emissions by fuel type (e.g. all coal oxidation is reported under 'coal',
regardless of whether oxidation results from combustion as an energy source), which differs
slightly from UNFCCC.
For the most recent 1-2 years when the UNFCCC estimates (1 year) and UN statistics (2 years)
used by CDIAC are not yet available, we generated preliminary estimates based on the BP annual
energy review by applying the growth rates of energy consumption (coal, oil, gas) for 2015 to the
national and global emissions from the UN national data in 2014, and for 2014 and 2015 to the
CDIAC national and global emissions in 2013 (BP, 2015). BP's sources for energy statistics overlap
with those of the UN data, but are compiled more rapidly from about 70 countries covering about
96% of global emissions. We use the BP values only for the year-to-year rate of change, because
the rates of change are less uncertain than the absolute values and to avoid discontinuities in the
time-series when linking the UN-based data with the BP data. These preliminary estimates are
replaced by the more complete UNFCCC or CDIAC data based on UN statistics when they become
available. Past experience and work by others (Andres et al., 2014; Myhre et al., 2009) shows that
projections based on the BP rate of change are within the uncertainty provided (see Sect. 3.2 and
Supplementary Information from Peters et al., 2013).
Estimates of emissions from cement production by CDIAC are based on data on growth rates of
cement production from the US Geological Survey up to year 2013 (USGS, 2016a). For 2014 and
2015 we use estimates of cement production made by the USGS for the top 18 countries
(representing 85% of global production; USGS, 2016b), while for all other countries we use the
2013 values (zero growth). Some fraction of the CaO and MgO in cement is returned to the
carbonate form during cement weathering but this is neglected here.
Estimates of emissions from gas flaring by CDIAC are calculated in a similar manner as those from
solid, liquid, and gaseous fuels, and rely on the UN Energy Statistics to supply the amount of flared
or vented fuel.  For the most recent 1-2 emission years, flaring is assumed constant from the most
recent available year of data (2014 for countries that report to the UNFCCC, 2013 for the
remainder). The basic data on gas flaring report atmospheric losses during petroleum production
and processing that have large uncertainty and do not distinguish between gas that is flared as
$CO_2$ or vented as $CH_4$. Fugitive emissions of $CH_4$ from the so-called upstream sector (e.g., coal
mining and natural gas distribution) are not included in the accounts of $CO_2$ emissions except to
the extent that they are captured in the UN energy data and counted as gas 'flared or lost'.
The published CDIAC data set includes 255 countries and regions. This list includes countries that
no longer exist, such as the USSR and East Pakistan. For the carbon budget, we reduce the list to
219 countries by reallocating emissions to the currently defined territories. This involved both
aggregation and disaggregation, and does not change global emissions. Examples of aggregation
include merging East and West Germany to the currently defined Germany. Examples of
disaggregation include reallocating the emissions from former USSR to the resulting independent
countries. For disaggregation, we use the emission shares when the current territories first
appeared. The disaggregated estimates should be treated with care when examining countries'
emissions trends prior to their disaggregation. For the most recent years, 2014 and 2015, the BP
statistics are more aggregated, but we retain the detail of CDIAC by applying the growth rates of
each aggregated region in the BP data set to its constituent individual countries in CDIAC.
Estimates of $CO_2$ emissions show that the global total of emissions is not equal to the sum of
emissions from all countries. This is largely attributable to emissions that occur in international
territory, in particular the combustion of fuels used in international shipping and aviation (bunker
fuels), where the emissions are included in the global totals but are not attributed to individual
countries. In practice, the emissions from international bunker fuels are calculated based on
where the fuels were loaded, but they are not included with national emissions estimates. Other
differences occur because globally the sum of imports in all countries is not equal to the sum of
exports and because of inconsistent national reporting, differing treatment of oxidation of non-
fuel uses of hydrocarbons (e.g. as solvents, lubricants, feedstocks, etc.), and changes in stock
(Andres et al., 2012).
The uncertainty of the annual emissions from fossil fuels and industry for the globe has been
estimated at ±5% (scaled down from the published ±10 % at ±2σ to the use of ±1σ bounds
reported here; Andres et al., 2012). This is consistent with a more detailed recent analysis of
uncertainty of ±8.4% at ±2σ (Andres et al., 2014) and at the high-end of the range of ±5-10% at
±2σ reported by Ballantyne et al. (2015). This includes an assessment of uncertainties in the
amounts of fuel consumed, the carbon and heat contents of fuels, and the combustion efficiency.
While in the budget we consider a fixed uncertainty of ±5% for all years, in reality the uncertainty,
as a percentage of the emissions, is growing with time because of the larger share of global
emissions from non-Annex B countries (emerging economies and developing countries) with less
precise statistical systems (Marland et al., 2009). For example, the uncertainty in Chinese
emissions has been estimated at around ±10% (for ±1σ; Gregg et al., 2008), and important
potential biases have been identified suggesting China's emissions could be overestimated in
published studies (Liu et al. 2015). Generally, emissions from mature economies with good
statistical bases have an uncertainty of only a few per cent (Marland, 2008). Further research is
needed before we can quantify the time evolution of the uncertainty, and its temporal error
correlation structure. We note that even if they are presented as 1σ estimates, uncertainties of
emissions are likely to be mainly country-specific systematic errors related to underlying biases of
energy statistics and to the accounting method used by each country. We assign a medium
confidence to the results presented here because they are based on indirect estimates of
emissions using energy data (Durant et al., 2010). There is only limited and indirect evidence for
emissions, although there is a high agreement among the available estimates within the given
uncertainty (Andres et al., 2014; Andres et al., 2012), and emission estimates are consistent with a
range of other observations (Ciais et al., 2013), even though their regional and national
partitioning is more uncertain (Francey et al., 2013).

## 2.1.2  Emissions embodied in goods and services

National emission inventories take a territorial (production) perspective and 'include greenhouse gas emissions and removals taking place within national territory and offshore areas over which the country has jurisdiction' (Rypdal et al., 2006). That is, emissions are allocated to the country where and when the emissions actually occur. The territorial emission inventory of an individual country does not include the emissions from the production of goods and services produced in other countries (e.g. food and clothes) that are used for consumption. Consumption-based emission inventories for an individual country is another attribution point of view that allocates global emissions to products that are consumed within a country, and are conceptually calculated as the territorial emissions minus the 'embedded' territorial emissions to produce exported products plus the emissions in other countries to produce imported products (Consumption = Territorial – Exports + Imports). The difference between the territorial- and consumption-based emission inventories is the net transfer (exports minus imports) of emissions from the production of internationally traded products. Consumption-based emission attribution results (e.g. Davis and Caldeira, 2010) provide additional information to territorial-based emissions that can be used to understand emission drivers (Hertwich and Peters, 2009), quantify emission transfers by the trade of products between countries (Peters et al., 2011b) and potentially design more effective and efficient climate policy (Peters and Hertwich, 2008).

We estimate consumption-based emissions from 1990-2014 by enumerating the global supply chain using a global model of the economic relationships between economic sectors within and between every country (Andrew and Peters, 2013; Peters et al., 2011a). Our analysis is based on the economic and trade data from the Global Trade and Analysis Project (GTAP; Narayanan et al., 2015), and we make detailed estimates for the years 1997 (GTAP version 5), 2001 (GTAP6), and 2004, 2007, and 2011 (GTAP9.1) (using the methodology of Peters et al., 2011b). The results cover 57 sectors and up to 141 countries and regions. The detailed results are then extended into an annual time-series from 1990 to the latest year of the GDP data (2014 in this budget), using GDP data by expenditure in current exchange rate of US dollars (USD; from the UN National Accounts main Aggregrates database; UN, 2014c) and time series of trade data from GTAP (based on the methodology in Peters et al., 2011b ).

We estimate the sector-level $CO_2$ emissions using our own calculations based on the GTAP data and methodology, include flaring and cement emissions from CDIAC, and then scale the national

totals (excluding bunker fuels) to match the CDIAC estimates from the most recent carbon budget.
We do not include international transportation in our estimates of national totals, but include
them in the global total. The time-series of trade data provided by GTAP covers the period 1995-
2013 and our methodology uses the trade shares as this data set. For the period 1990-1994 we
assume the trade shares of 1995, while for 2014 we assume the trade shares of 2013.
Comprehensive analysis of the uncertainty of consumption emissions accounts is still lacking in
the literature, although several analyses of components of this uncertainty have been made (e.g.
Dietzenbacher et al., 2012; Inomata and Owen, 2014; Karstensen et al., 2015; Moran and Wood,
2014). For this reason we do not provide an uncertainty estimate for these emissions, but based
on model comparisons and sensitivity analysis, they are unlikely to be larger than for the
territorial emission estimates (Peters et al., 2012a). Uncertainty is expected to increase for more
detailed results, and to decrease with aggregation (Peters et al., 2011b; e.g. the results for Annex
B countries will be more accurate than the sector results for an individual country).
The consumption-based emissions attribution method considers the $CO_2$ emitted to the
atmosphere in the production of products, but not the trade in fossil fuels (coal, oil, gas). It is also
possible to account for the carbon trade in fossil fuels (Andrew et al., 2013), but we do not
present those data here. Peters et al. (2012a) additionally considered trade in biomass.
The consumption data do not modify the global average terms in Eq. (1), but are relevant to the
anthropogenic carbon cycle as they reflect the trade-driven movement of emissions across the
Earth's surface in response to human activities. Furthermore, if national and international climate
policies continue to develop in an un-harmonised way, then the trends reflected in these data will
need to be accommodated by those developing policies.

### 2.1.3   Growth rate in emissions

We report the annual growth rate in emissions for adjacent years (in percent per year) by
calculating the difference between the two years and then comparing to the emissions in the first
year: $\left[\frac{E_{FF(t_{0+1})}-E_{FF(t_0)}}{E_{FF(t_0)}}\right]\times\%yr^{-1}$. This is the simplest method to characterise a one-year growth
compared to the previous year and is widely used. We apply a leap-year adjustment to ensure
valid interpretations of annual growth rates. This affects the growth rate by about 0.3% yr$^{-1}$ ($\frac{1}{365}$)
and causes growth rates to go up approximately 0.3% if the first year is a leap year and down 0.3%
if the second year is a leap year.
The relative growth rate of $E_{FF}$ over time periods of greater than one year can be re-written using
its logarithm equivalent as follows:

$$\frac{1}{E_{FF}} \frac{dE_{FF}}{dt} = \frac{d(lnE_{FF})}{dt} \tag{2}$$

Here we calculate relative growth rates in emissions for multi-year periods (e.g. a decade) by
fitting a linear trend to $ln(E_{FF})$ in Eq. (2), reported in percent per year. We fit the logarithm of $E_{FF}$
rather than $E_{FF}$ directly because this method ensures that computed growth rates satisfy Eq. (6).
This method differs from previous papers (Canadell et al., 2007; Le Quéré et al., 2009; Raupach et
al., 2007) that computed the fit to $E_{FF}$ and divided by average $E_{FF}$ directly, but the difference is very
small (<0.05% yr$^{-1}$) in the case of $E_{FF}$.

### 2.1.4  Emissions projections

Energy statistics from BP are normally available around June for the previous year. To gain insight
on emission trends for the current year (2016), we provide an assessment of global emissions for
$E_{FF}$ by combining individual assessments of emissions for China and the USA (the two biggest
emitting countries), and the rest of the world.
We specifically estimate emissions in China because the data indicate a significant departure from
the long-term trends in the carbon intensity of the economy used in emissions projections in
previous global carbon budgets (e.g. Le Quéré et al. 2015a), resulting from a rapid deceleration in
emissions growth against continued growth in economic output. This departure could be
temporary (Jackson et al., 2016). Our 2016 estimate for China uses:  (1) coal consumption
estimates from the China Coal Transportation and Distribution Association for January through
August (CCTD, 2016) (2) estimated consumption of natural gas (NDRC, 2016a) and domestic
production plus net imports of petroleum (NDRC, 2016b) for January through July from the
National Development and Reform Commission, and (3) production of cement reported for
January to August (National Bureau of Statistics of China, 2016; NBS, 2016). Using these data, we
estimate the change in emissions for the corresponding months in 2016 compared to 2015
assuming a 2% improvement in coal energy content for 2016 in line with recent years. We then
assume that the relative changes during the first months will persist throughout the year. The
main sources of uncertainty are from the incomplete data on stock changes, the carbon content
of coal and the assumption of persistent behaviour for the rest of the year. These are discussed
further in section 3.2.1.
For the USA, we use the forecast of the U.S. Energy Information Administration (EIA) for emissions
from fossil fuels (IEA, 2015). This is based on an energy forecasting model which is revised
monthly, and takes into account heating-degree days, household expenditures by fuel type,
energy markets, policies, and other effects. We combine this with our estimate of emissions from
cement production using the monthly U.S. cement data from USGS for January-July, assuming
changes in cement production over the first seven months apply throughout the year. While the
EIA's forecasts for current full-year emissions have on average been revised downwards, only
seven such forecasts are available, so we conservatively use the full range of adjustments
following revision, and additionally assume symmetrical uncertainty to give –2.5% to +2.5%
around the central forecast.
For the rest of the world, we use the close relationship between the growth in GDP and the
growth in emissions (Raupach et al., 2007) to project emissions for the current year. This is based
on the so-called Kaya identity (also called IPAT identity, the acronym standing for human impact
(I) on the environment, which is equal to the product of P= population, A= affluence, T=
technology), whereby $E_{FF}$ (GtC yr$^{-1}$) is decomposed by the product of GDP (USD yr$^{-1}$) and the fossil
fuel carbon intensity of the economy ($I_{FF}$; GtC USD$^{-1}$) as follows:

$$E_{FF} = GDP \times I_{FF} \qquad (3)$$

Such product-rule decomposition identities imply that the relative growth rates of the multiplied
quantities are additive. Taking a time derivative of Equation (3) gives:

$$\frac{dE_{FF}}{dt} = \frac{d(GDP \times I_{FF})}{dt} \qquad (4)$$

and applying the rules of calculus:

$$\frac{dE_{FF}}{dt} = \frac{dGDP}{dt} \times I_{FF} + GDP \times \frac{dI_{FF}}{dt} \qquad (5)$$

finally, dividing (5) by (3) gives :

$$\frac{1}{E_{FF}}\frac{dE_{FF}}{dt} = \frac{1}{GDP}\frac{dGDP}{dt} + \frac{1}{I_{FF}}\frac{dI_{FF}}{dt} \qquad (6)$$

where the left hand term is the relative growth rate of $E_{FF}$, and the right hand terms are the
relative growth rates of GDP and $I_{FF}$, respectively, which can simply be added linearly to give
overall growth rate. The growth rates are reported in percent by multiplying each term by 100. As
preliminary estimates of annual change in GDP are made well before the end of a calendar year,
making assumptions on the growth rate of $I_{FF}$ allows us to make projections of the annual change
in $CO_2$ emissions well before the end of a calendar year. The $I_{FF}$ is based on GDP in constant PPP
(purchasing power parity) from the IEA up to 2013 (IEA/OECD, 2015)  and extended using the IMF
growth rates for 2014 and 2015 (IMF, 2016). Interannual variability in $I_{FF}$ is the largest source of
uncertainty in the GDP-based emissions projections. We thus use the standard deviation of the
annual $I_{FF}$ for the period 2006-2015 as a measure of uncertainty, reflecting a ±1σ as in the rest of
the carbon budget. This is ±1.0% $yr^{-1}$ for the rest of the world (global emissions minus China and
USA).
The 2016 projection for the world is made of the sum of the projections for China, USA, and the
rest. The uncertainty is added in quadrature among the three regions. The uncertainty here
reflects the best of our expert opinion.
**2.2    $CO_2$ emissions from land use, land-use change and forestry ($E_{LUC}$)**
Land-use change emissions reported here ($E_{LUC}$) include $CO_2$ fluxes from deforestation,
afforestation, logging (forest degradation and harvest activity), shifting cultivation (cycle of cutting
forest for agriculture, then abandoning), and regrowth of forests following wood harvest or
abandonment of agriculture. Only some land management activities are included in our land-use
change emissions estimates (Table 5). Some of these activities lead to emissions of $CO_2$ to the
atmosphere, while others lead to $CO_2$ sinks. $E_{LUC}$ is the net sum of all anthropogenic activities
considered. Our annual estimate for 1959-2010 is from a bookkeeping method (Sect. 2.2.1)
primarily based on net forest area change and biomass data from the Forest Resource Assessment
(FRA) of the Food and Agriculture Organisation (FAO) which is only available at intervals of five
years. We use the bookkeeping method based on FAO FRA 2010 here (Houghton et al., 2012), and
present preliminary results of an update using the FAO FRA 2015 (Houghton and Nassikas, in
preparation). Inter-annual variability in emissions due to deforestation and degradation have been
coarsely estimated from satellite-based fire activity in tropical forest areas (Section 2.2.2; Giglio et
al., 2013; van der Werf et al., 2010). The bookkeeping method is used to quantify the $E_{LUC}$ over the
time period of the available data, and the satellite-based deforestation fire information to
incorporate interannual variability ($E_{LUC}$ flux annual anomalies) from tropical deforestation fires.
The satellite-based deforestation and degradation fire emissions estimates are available for years
1997-2015. We calculate the global annual anomaly in deforestation and degradation fire
emissions in tropical forest regions for each year, compared to the 1997-2010 period, and add this
annual flux anomaly to the $E_{LUC}$ estimated using the published bookkeeping method that is
available up to 2010 only and assumed constant at the 2010 value during the period 2011-2015.
We thus assume that all land management activities apart from deforestation and degradation do
not vary significantly on a year-to-year basis. Other sources of interannual variability (e.g. the
impact of climate variability on regrowth fluxes) are accounted for in $S_{LAND}$. In addition, we use
results from Dynamic Global Vegetation Models (see Section 2.2.3 and Table 6) that calculate net
land-use change $CO_2$ emissions in response to land-cover change reconstructions prescribed to
each model, to help quantify the uncertainty in $E_{LUC}$, and to explore the consistency of our
understanding. The three methods are described below, and differences are discussed in Section

16   3.2.

### 2.2.1  Bookkeeping method

Land-use change $CO_2$ emissions are calculated by a bookkeeping method approach (Houghton,
2003) that keeps track of the carbon stored in vegetation and soils before deforestation or other
land-use change, and the changes in forest age classes, or cohorts, of disturbed lands after land-
use change including possible forest regrowth after deforestation. It tracks the $CO_2$ emitted to the
atmosphere immediately during deforestation, and over time due to the follow-up decay of soil
and vegetation carbon in different pools, including wood products pools after logging and
deforestation. It also tracks the regrowth of vegetation and associated build-up of soil carbon
pools after land-use change. It considers transitions between forests, pastures and cropland;
shifting cultivation; degradation of forests where a fraction of the trees is removed; abandonment
of agricultural land; and forest management such as wood harvest and, in the USA, fire
management. In addition to tracking logging debris on the forest floor, the bookkeeping method
tracks the fate of carbon contained in harvested wood products that is eventually emitted back to
the atmosphere as $CO_2$, although a detailed treatment of the lifetime in each product pool is not
performed (Earles et al., 2012). Harvested wood products are partitioned into three pools with
different turnover times. All fuel-wood is assumed burnt in the year of harvest (1.0 yr$^{-1}$). Pulp and
paper products are oxidized at a rate of 0.1 yr$^{-1}$, timber is assumed to be oxidized at a rate of 0.01
yr$^{-1}$, and elemental carbon decays at 0.001 yr$^{-1}$. The general assumptions about partitioning wood
products among these pools are based on national harvest data (Houghton, 2003).
The primary land-cover change and biomass data for the bookkeeping method analysis is the
Forest Resource Assessment of the FAO which provides statistics on forest-cover change and
management at intervals of five years (FAO, 2010). The data is based on countries' self-reporting
some of which include satellite data in more recent assessments (Table 4). Changes in land cover
other than forest are based on annual, national changes in cropland and pasture areas reported
by the FAO Statistics Division (FAOSTAT, 2010). Land-use change country data are aggregated by
regions. The carbon stocks on land (biomass and soils), and their response functions subsequent
to land-use change, are based on FAO data averages per land cover type, per biome and per
region. Similar results were obtained using forest biomass carbon density based on satellite data
(Baccini et al., 2012). The bookkeeping method does not include land ecosystems' transient
response to changes in climate, atmospheric $CO_2$ and other environmental factors, and the
growth/decay curves are based on contemporary data that will implicitly reflect the effects of $CO_2$
and climate at that time. Published results from the bookkeeping method are available from 1850
to 2010, with preliminary results available to 2015.
**2.2.2   Fire-based interannual variability in E$_{LUC}$**
Land-use change associated $CO_2$ emissions calculated from satellite-based fire activity in tropical
forest areas (van der Werf et al., 2010) provide information on emissions due to tropical
deforestation and degradation that are complementary to the bookkeeping approach. They do
not provide a direct estimate of E$_{LUC}$ as they do not include non-combustion processes such as
respiration, wood harvest, wood products or forest regrowth. Legacy emissions such as
decomposition from on-ground debris and soils are not included in this method either. However,
fire estimates provide some insight in the year-to-year variations in the sub-component of the
total E$_{LUC}$ flux that result from immediate $CO_2$ emissions during deforestation caused, for example,
by the interactions between climate and human activity (e.g. there is more burning and clearing of
forests in dry years) that are not represented by other methods. The 'deforestation fire emissions'
assume an important role of fire in removing biomass in the deforestation process, and thus can
be used to infer gross instantaneous $CO_2$ emissions from deforestation using satellite-derived data

1. on fire activity in regions with active deforestation. The method requires information on the
2. fraction of total area burned associated with deforestation versus other types of fires, and this
3. information can be merged with information on biomass stocks and the fraction of the biomass
4. lost in a deforestation fire to estimate $CO_2$ emissions. The satellite-based deforestation fire
5. emissions are limited to the tropics, where fires result mainly from human activities. Tropical
6. deforestation is the largest and most variable single contributor to $E_{LUC}$.
7. Fire emissions associated with deforestation and tropical peat burning are based on the Global
8. Fire Emissions Database (GFED4; accessed July 2016) described in van der Werf et al. (2010) but
9. with updated burned area (Giglio et al., 2013) as well as burned area from relatively small fires
10. that are detected by satellite as thermal anomalies but not mapped by the burned area approach
11. (Randerson, 2012). The burned area information is used as input data in a modified version of the
12. satellite-driven Carnegie Ames Stanford Approach (CASA) biogeochemical model to estimate
13. carbon emissions associated with fires, keeping track of what fraction of fire emissions was due to
14. deforestation (see van der Werf et al., 2010). The CASA model uses different assumptions to
15. compute decay functions compared to the bookkeeping method, and does not include historical
16. emissions or regrowth from land-use change prior to the availability of satellite data. Comparing
17. coincident CO emissions and their atmospheric fate with satellite-derived CO concentrations
18. allows for some validation of this approach (e.g. van der Werf et al., 2008). Results from the fire-
19. based method to estimate land-use change emissions anomalies added to the bookkeeping mean
20. $E_{LUC}$ estimate are available from 1997 to 2015. Our combination of land-use change $CO_2$ emissions
21. where the variability of annual $CO_2$ deforestation emissions is diagnosed from fires assumes that
22. year-to-year variability is dominated by variability in deforestation.

23. ### 2.2.3 Dynamic Global Vegetation Models (DGVMs)

24. Land-use change $CO_2$ emissions have been estimated using an ensemble of DGVMs simulations.
25. New model experiments up to year 2015 have been coordinated by the project 'Trends and
26. drivers of the regional-scale sources and sinks of carbon dioxide (TRENDY; Sitch et al., 2015)'. We
27. use only models that have estimated land-use change $CO_2$ emissions following the TRENDY
28. protocol (see Section 2.5.2). Models use their latest configurations, summarised in Tables 5 and 6.

29. Two sets of simulations were performed with the DGVMs, first forced with historical changes in
30. land cover distribution, climate, atmospheric $CO_2$ concentration, and N deposition, and second, as
31. further described below with a time-invariant pre-industrial land cover distribution, allowing to

estimate, by difference with the first simulation, the dynamic evolution of biomass and soil carbon
pools in response to prescribed land-cover change. Because of the limited availability of the land
use forcing  (see below), 14 DGVMs performed historical simulations with time-invariant land
cover distribution, but only 5 DGMs managed to simulate realistic simulations with time varying
land cover change.  These latter DGVMs accounted for deforestation and (to some extent)
regrowth, the most important components of $E_{LUC}$, but they do not represent all processes
resulting directly from human activities on land (Table 5). All DGVMs represent processes of
vegetation growth and mortality, as well as decomposition of dead organic matter associated with
natural cycles, and include the vegetation and soil carbon response to increasing atmospheric $CO_2$
levels and to climate variability and change. In addition, eight models explicitly simulate the
coupling of C and N cycles and account for atmospheric N deposition (Table 5), with three of those
models used for land-use change simulations. The DGVMs are independent from the other budget
terms except for their use of atmospheric $CO_2$ concentration to calculate the fertilization effect of
$CO_2$ on primary production.
For this global carbon budget, the DGVMs used the HYDE land-use change data set (Klein
Goldewijk et al., 2011), which provides annual, half-degree, fractional data on cropland and
pasture. These data are based on annual FAO statistics of change in agricultural area available to
2012 (FAOSTAT, 2010). For the years 2013 to 2015, the HYDE data were extrapolated by country
for pastures and cropland separately based on the trend in agricultural area over the previous 5
years. The more comprehensive LUH dataset (Hurtt et al., 2011), that also includes fractional data
on primary vegetation and secondary vegetation, as well as all underlying transitions between
land-use states, has not been made available yet for this year. Hence the reduced ensemble of
DGVMs that can simulate the LUC flux from the HYDE dataset only. The HYDE data are
independent from the data set used in the bookkeeping method (Houghton, 2003 and updates),
which is based primarily on forest area change statistics (FAO, 2010). The HYDE land-use change
data set does not indicate whether land-use changes occur on forested or non-forested land, it
only provides the changes in agricultural areas. Hence, it is implemented differently within each
model (e.g. an increased cropland fraction in a grid cell can either be at the expense of grassland,
or forest, the latter resulting in deforestation; land cover fractions of the non-agricultural land
differ between models). Thus the DGVM forest area and forest area change over time is not
consistent with the Forest Resource Assessment of the FAO forest area data used for the
bookkeeping model to calculate $E_{LUC}$. Similarly, model-specific assumptions are applied to convert
deforested biomass or deforested area, and other forest product pools, into carbon in some
models (Table 5).
The DGVM model runs were forced by either 6 hourly CRU-NCEP or by monthly CRU temperature,
precipitation, and cloud cover fields (transformed into incoming surface radiation) based on
observations and provided on a 0.5°x0.5° grid and updated to 2015 (Harris et al., 2014; Viovy,
2016). The forcing data include both gridded observations of climate and global atmospheric $CO_2$,
which change over time (Dlugokencky and Tans, 2016), and N deposition (as used in some models;
Table 5). As mentioned before, $E_{LUC}$ is diagnosed in each model by the difference between a model
simulation with prescribed historical land cover change and a simulation with constant,
preindustrial land cover distribution. Both simulations were driven by changing atmospheric $CO_2$,
climate, and in some models N deposition over the period 1860-2015. Using the difference
between these two DGVM simulations to diagnose $E_{LUC}$ is not fully consistent with the definition
of $E_{LUC}$ in the bookkeeping method (Gasser and Ciais, 2013; Stocker and Joos, 2015). The DGVM
approach to diagnose land-use change $CO_2$ emissions would be expected to produce
systematically higher $E_{LUC}$ emissions than the bookkeeping approach if all the parameters of the
two approaches were the same, which is not the case (see Section 2.5.2).
**2.2.4   Other published $E_{LUC}$ methods**
Other methods have been used to estimate $CO_2$ emissions from land-use change. We describe
some of the most important methodological differences between the approach used here and
other published methods, and for completion, we explain why they are not used in the budget.
Different definitions and boundary conditions (e.g. the inclusion of fire management) for $E_{LUC}$ can
lead to significantly different estimates within models (Gasser and Ciais, 2013; Hansis et al., 2015;
Pongratz et al., 2014) as well as between models and other approaches (Houghton et al., 2012;
Smith et al., 2014b). FAO uses the IPCC approach called 'Tier 1-type' (e.g. Tubiello et al., 2015) to
produce a 'Land use – forest land' estimate from the Forest Resources Assessment data updated
from the one used in the bookkeeping method described in Section 2.2.1 (MacDicken, 2015). The
Tier 1-type method applies a nationally reported mean forest carbon stock change (above and
below ground living biomass) to nationally reported net forest area change, across all forest land
combined (planted and natural forests). The methods implicitly assume instantaneous loss or gain
of mean forest. Thus the IPCC Tier 1-type approach provides an estimate of attributable emissions
from the process of land-cover change, but it does not distribute these emissions through time. It
also captures some of what the global modelling approach considers residual carbon flux ($S_{LAND}$), it
does not consider loss of soil carbon, and there are no legacy fluxes. Land use fluxes estimated
with this method were 0.47 GtC yr$^{-1}$ in 2001-2010 and 0.22 GtC yr$^{-1}$ in 2011-2015 (Federici et al.,
2015). This estimate is not directly comparable with $E_{LUC}$ used here because of the different
boundary conditions and processes included.
Recent advances in satellite data leading to higher resolution area change data (e.g. Hansen et al.,
2013) and estimates of biomass in live vegetation (e.g. Baccini et al., 2012; Saatchi, 2011), have
led to several satellite-based estimates of $CO_2$ emissions due to tropical deforestation (typically
gross loss of forest area; Achard and House, in press).  These include estimates of 1.0 GtC yr$^{-1}$ for
2000 to 2010 (Baccini et al., 2012), 0.8 GtC yr$^{-1}$ for 2000 to 2005 (Harris, 2012), 0.9 GtC yr$^{-1}$ for
2000 to 2010 for net area change, and 1.3 GtC yr$^{-1}$ 2000 to 2010 (Tyukavina et al., 2015). These
estimates include belowground carbon biomass using a scaling factor. Some estimate soil carbon
loss, some assume instantaneous emissions, some do not account for regrowth fluxes, and none
account for legacy fluxes from land-use change prior to the availability of satellite data. They are
mostly estimates of tropical deforestation only, and do not capture regrowth flux after
abandonment, or planting (Achard and House, in press). These estimates are also difficult to
compare with $E_{LUC}$ used here because they do not fully include legacy fluxes and forest regrowth.
**2.2.5    Uncertainty assessment for $E_{LUC}$**
Differences between the bookkeeping, the addition of fire-based interannual variability to the
bookkeeping, and DGVM methods originate from three main sources: the land cover change data
set, the different approaches used in models, and the different processes represented (Table 5).
We examine the results from the DGVM models and of the bookkeeping method to assess the
uncertainty in $E_{LUC}$.
The uncertainties in annual $E_{LUC}$ estimates are examined using the standard deviation across
models, which averages 0.3 GtC yr$^{-1}$ from 1959 to 2015 (Table 7). The mean of the multi-model
$E_{LUC}$ estimates is consistent with a combination of the bookkeeping method and fire-based
emissions (Le Quéré et al. 2014), with the multi-model mean and bookkeeping method differing
by less than 0.5 GtC yr$^{-1}$ over 85% of the time. Based on this comparison, we assess that an
uncertainty of ±0.5 GtC yr$^{-1}$ provides a semi-quantitative measure of uncertainty for annual
emissions, and reflects our best value judgment that there is at least 68% chance (±1σ) that the
true land-use change emission lies within the given range, for the range of processes considered
here. This is consistent with the uncertainty analysis of Houghton et al. (2012), which partly
reflects improvements in data on forest area change using data, and partly more complete
understanding and representation of processes in models.
The uncertainties in the decadal $E_{LUC}$ estimates are also examined using the DGVM ensemble,
although they are likely correlated between decades. The correlations between decades come
from (1) common biases in system boundaries (e.g. not counting forest degradation in some
models); (2) common definition for the calculation of $E_{LUC}$ from the difference of simulations with
and without land-use change (a source of bias vs. the unknown truth); (3) common and uncertain
land-cover change input data which also cause a bias, though if a different input data set is used
each decade, decadal fluxes from DGVMs may be partly decorrelated; (4) model structural errors
(e.g. systematic errors in biomass stocks). In addition, errors arising from uncertain DGVM
parameter values would be random but they are not accounted for in this study, since no DGVM
provided an ensemble of runs with perturbed parameters.
Prior to 1959, the uncertainty in $E_{LUC}$ is taken as ±33%, which is the ratio of uncertainty to mean
from the 1960s in the bookkeeping method (Table 7), the first decade available. This ratio is
consistent with the mean standard deviation of DGMVs land-use change emissions over 1870-
1958 (0.32 GtC) over the multi-model mean (0.9 GtC).
**2.3    Growth rate in atmospheric CO$_2$ concentration (G$_{ATM}$)**
**2.3.1    Global growth rate in atmospheric CO$_2$ concentration**
The rate of growth of the atmospheric CO$_2$ concentration is provided by the US National Oceanic
and Atmospheric Administration Earth System Research Laboratory (NOAA/ESRL; Dlugokencky
and Tans, 2016), which is updated from Ballantyne et al. (2012). For the 1959-1980 period, the
global growth rate is based on measurements of atmospheric CO$_2$ concentration averaged from
the Mauna Loa and South Pole stations, as observed by the CO$_2$ Program at Scripps Institution of
Oceanography (Keeling et al., 1976). For the 1980-2015 time period, the global growth rate is
based on the average of multiple stations selected from the marine boundary layer sites with well-
mixed background air (Ballantyne et al., 2012), after fitting each station with a smoothed curve as
a function of time, and averaging by latitude band (Masarie and Tans, 1995). The annual growth
rate is estimated by Dlugokencky and Tans (2016) from atmospheric $CO_2$ concentration by taking
the average of the most recent December-January months corrected for the average seasonal
cycle and subtracting this same average one year earlier. The growth rate in units of ppm yr$^{-1}$ is
converted to units of GtC yr$^{-1}$ by multiplying by a factor of 2.12 GtC per ppm (Ballantyne et al.,
2012) for consistency with the other components.
The uncertainty around the annual growth rate based on the multiple stations data set ranges
between 0.11 and 0.72 GtC yr$^{-1}$, with a mean of 0.61 GtC yr$^{-1}$ for 1959-1979 and 0.19 GtC yr$^{-1}$ for
1980-2015, when a larger set of stations were available (Dlugokencky and Tans, 2016). It is based
on the number of available stations, and thus takes into account both the measurement errors
and data gaps at each station. This uncertainty is larger than the uncertainty of ±0.1 GtC yr$^{-1}$
reported for decadal mean growth rate by the IPCC because errors in annual growth rate are
strongly anti-correlated in consecutive years leading to smaller errors for longer time scales. The
decadal change is computed from the difference in concentration ten years apart based on a
measurement error of 0.35 ppm. This error is based on offsets between NOAA/ESRL
measurements and those of the World Meteorological Organization World Data Centre for
Greenhouse Gases (NOAA/ESRL, 2015) for the start and end points (the decadal change
uncertainty is the $\sqrt{(2(0.35 ppm)^2)}(10\ yr)^{-1}$ assuming that each yearly measurement error is
independent). This uncertainty is also used in Table 8.
The contribution of anthropogenic CO and $CH_4$ is neglected from the global carbon budget (see
Sect. 2.7.1). We assign a high confidence to the annual estimates of $G_{ATM}$ because they are based
on direct measurements from multiple and consistent instruments and stations distributed
around the world (Ballantyne et al., 2012).
In order to estimate the total carbon accumulated in the atmosphere since 1750 or 1870, we use
an atmospheric $CO_2$ concentration of 277 ± 3 ppm or 288 ± 3 ppm, respectively, based on a cubic
spline fit to ice core data (Joos and Spahni, 2008). The uncertainty of ±3 ppm (converted to ±1σ) is
taken directly from the IPCC's assessment (Ciais et al., 2013). Typical uncertainties in the growth
rate in atmospheric $CO_2$ concentration from ice core data are ±1-1.5 GtC per decade as evaluated
from the Law Dome data (Etheridge et al., 1996) for individual 20-year intervals over the period
from 1870 to 1960 (Bruno and Joos, 1997).
## 2.4   Ocean CO$_2$ sink
Estimates of the global ocean CO$_2$ sink are based on a combination of a mean CO$_2$ sink estimate
for the 1990s from observations, and a trend and variability in the ocean CO$_2$ sink for 1959-2015
from seven global ocean biogeochemistry models. We use two observation-based estimates of
S$_{OCEAN}$ available for recent decades to provide a qualitative assessment of confidence in the
reported results.
### 2.4.1   Observation-based estimates
A mean ocean CO$_2$ sink of 2.2 ± 0.4 GtC yr$^{-1}$ for the 1990s was estimated by the IPCC (Denman et
al., 2007) based on indirect observations and their spread: ocean/land CO$_2$ sink partitioning from
observed atmospheric O$_2$/N$_2$ concentration trends (Keeling et al., 2011; Manning and Keeling,
2006), an oceanic inversion method constrained by ocean biogeochemistry data (Mikaloff Fletcher
et al., 2006), and a method based on penetration time scale for CFCs (McNeil et al., 2003). This is
comparable with the sink of 2.0 ± 0.5 GtC yr$^{-1}$ estimated by Khatiwala et al. (2013) for the 1990s,
and with the sink of 1.9 to 2.5 GtC yr$^{-1}$ estimated from a range of methods for the period 1990-
2009 (Wanninkhof et al., 2013), with uncertainties ranging from ±0.3 GtC yr$^{-1}$ to ±0.7 GtC yr$^{-1}$. The
most direct way for estimating the observation-based ocean sink is from the product of (sea-air
pCO$_2$ difference) x (gas transfer coefficient). Estimates based on sea-air pCO$_2$ are fully consistent
with indirect observations (Zeng et al., 2005), but their uncertainty is larger mainly due to
difficulty in capturing complex turbulent processes in the gas transfer coefficient (Sweeney et al.,
2007) and because of uncertainties in the pre-industrial river outgas of CO$_2$ (Jacobson et al., 2007).
Both observation-based estimates (Landschutzer et al., 2015; Rödenbeck et al., 2014a) compute
the ocean CO$_2$ sink and its variability using interpolated measurements of surface ocean fugacity
of CO$_2$ (pCO2 corrected for the non-ideal behaviour of the gas; Pfeil et al., 2013). The
measurements were from the Surface Ocean CO$_2$ Atlas version 4, which is an update of version 3
(Bakker et al., 2016) and contains data to 2015 (see data attribution Table 1A). In contrast to last
year's global carbon budget, where preliminary data was used for the past year, data used here
are fully quality-controlled following standard SOCAT procedures. The SOCAT v4 were mapped
using a data-driven diagnostic method (Rödenbeck et al., 2013) and a combined self-organising
map and feed-forward neural network (Landschützer et al., 2014). The global observation-based
estimates were adjusted to remove a background (not part of the anthropogenic ocean flux)
ocean source of $CO_2$ to the atmosphere of 0.45 GtC yr$^{-1}$ from river input to the ocean (Jacobson et
al., 2007), to make them comparable to $S_{OCEAN}$ which only represents the annual uptake of
anthropogenic $CO_2$ by the ocean. Several other data-based products are available, but they show
large discrepancies with observed variability that need to be resolved. Here we used the two data
products that had the best fit to observations, distinctly better than most in their representation
of tropical and global variability (Rödenbeck et al., 2015).
We use the data-based product of Khatiwala et al. (2009) updated by Khatiwala et al. (2013) to
estimate the anthropogenic carbon accumulated in the ocean during 1765-1958 (60.2 GtC) and
1870-1958 (47.5 GtC), and assume an oceanic uptake of 0.4 GtC for 1750-1765 (for which time no
data are available) based on the mean uptake during 1765-1770. The estimate of Khatiwala et al.
(2009) is based on regional disequilibrium between surface $pCO_2$ and atmospheric $CO_2$, and a
Green's function utilizing transient ocean tracers like CFCs and $^{14}C$ to ascribe changes through
time. It does not include changes associated with changes in ocean circulation, temperature and
climate, but these are thought to be small over the time period considered here (Ciais et al.,
2013). The uncertainty in cumulative uptake of ±20 GtC (converted to ±1σ) is taken directly from
the IPCC's review of the literature (Rhein et al., 2013), or about ±30% for the annual values
(Khatiwala et al., 2009).
**2.4.2    Global Ocean Biogeochemistry models**
The trend in the ocean $CO_2$ sink for 1959-2015 is computed using a combination of seven global
ocean biogeochemistry models (Table 6). The models represent the physical, chemical and
biological processes that influence the surface ocean concentration of $CO_2$ and thus the air-sea
$CO_2$ flux. The models are forced by meteorological reanalysis and atmospheric $CO_2$ concentration
data available for the entire time period. Models do not include the effects of anthropogenic
changes in nutrient supply, which could lead to an increase of up to about 0.3 GtC yr$^{-1}$ over the
industrial period (Duce et al., 2008). They compute the air-sea flux of $CO_2$ over grid boxes of 1° to
4° in latitude and longitude. The ocean $CO_2$ sink for each model is normalised to the observations,
by dividing the annual model values by their modeled average over 1990-1999 and multiplying
this with the observation-based estimate of 2.2 GtC yr$^{-1}$ (obtained from Keeling et al., 2011;
Manning and Keeling, 2006; McNeil et al., 2003; Mikaloff Fletcher et al., 2006). The ocean $CO_2$ sink
for each year ($t$) in GtC yr$^{-1}$ is therefore:

$$S_{OCEAN}(t) = \frac{1}{n} \sum_{m=1}^{m=n} \frac{S_{OCEAN}^{m}(t)}{S_{OCEAN}^{m}(1990-1999)} \times 2.2 \qquad (7)$$

where $n$ is the number of models. This normalisation ensures that the ocean $CO_2$ sink for the
global carbon budget is based on observations, whereas the trends and annual values in $CO_2$ sinks
are from model estimates. The normalisation based on a ratio assumes that if models over or
underestimate the sink in the 1990s, it is primarily due to the process of diffusion, which depends
on the gradient of $CO_2$. Thus a ratio is more appropriate than an offset as it takes into account the
time-dependence of $CO_2$ gradients in the ocean. The mean uncorrected ocean $CO_2$ sink from the
models for 1990-1999 ranges between 1.7 and 2.4 GtC yr$^{-1}$, with a multi model mean of 2.0 GtC yr$^{-}$
$^1$.
### 2.4.3   Uncertainty assessment for $S_{OCEAN}$
The uncertainty around the mean ocean sink of anthropogenic $CO_2$ was quantified by Denman et
al. (2007) for the 1990s (see Section 2.4.1). To quantify the uncertainty around annual values, we
examine the standard deviation of the normalised model ensemble. We use further information
from the two data-based products to assess the confidence level. The average standard deviation
of the normalised ocean model ensemble is 0.16 GtC yr$^{-1}$ during 1980-2010 (with a maximum of
0.33), but it increases as the model ensemble goes back in time, with a standard deviation of 0.22
GtC yr$^{-1}$ across models in the 1960s. We estimate that the uncertainty in the annual ocean $CO_2$
sink is about ± 0.5 GtC yr$^{-1}$ from the fractional uncertainty of the data uncertainty of ± 0.4 GtC yr$^{-1}$
and standard deviation across models of up to ± 0.33 GtC yr$^{-1}$, reflecting both the uncertainty in
the mean sink from observations during the 1990's (Denman et al., 2007; Section 2.4.1) and in the
interannual variability as assessed by models.
We examine the consistency between the variability of the model-based and the data-based
products to assess confidence in $S_{OCEAN}$. The interannual variability of the ocean fluxes (quantified
as the standard deviation) of the two data-based estimates for 1986-2015 (where they overlap) is
± 0.34 GtC yr$^{-1}$ (Rödenbeck et al., 2014b) and ± 0.41 GtC yr$^{-1}$ (Landschützer et al., 2015), compared
to ± 0.29 GtC yr$^{-1}$ for the normalised model ensemble. The standard deviation includes a
component of trend and decadal variability in addition to interannual variability, and their relative
influence differs across estimates. The phase is generally consistent between estimates, with a
higher ocean $CO_2$ sink during El Niño events. The annual data-based estimates correlate with the
ocean $CO_2$ sink estimated here with a correlation of r = 0.71 (0.51 to 0.77 for individual models),
and r = 0.81 (0.66 to 0.79) for the data-based estimates of Rödenbeck et al. (2014) and
Landschützer et al. (2015), respectively (simple linear regression), with their mutual correlation at
0.65. The agreement is better for decadal variability than for interannual variability. The use of
annual data for the correlation may reduce the strength of the relationship because the dominant
source of variability associated with El Niño events is less than one year. We assess a medium
confidence level to the annual ocean $CO_2$ sink and its uncertainty because they are based on
multiple lines of evidence, and the results are consistent in that the interannual variability in the
model and data-based estimates are all generally small compared to the variability in the growth
rate of atmospheric $CO_2$ concentration.

## 2.5    Terrestrial $CO_2$ sink

The difference between, on the one hand fossil fuel ($E_{FF}$) and land-use change emissions ($E_{LUC}$),
and on the other hand the growth rate in atmospheric $CO_2$ concentration ($G_{ATM}$) and the ocean
$CO_2$ sink ($S_{OCEAN}$), is attributable to the net sink of $CO_2$ in terrestrial vegetation and soils ($S_{LAND}$),
within the given uncertainties (Eq. 1). Thus, this sink can be estimated as the residual of the other
terms in the mass balance budget, as well as directly calculated using DGVMs. The residual land
sink ($S_{LAND}$) is thought to be in part because of the fertilising effect of rising atmospheric $CO_2$ on
plant growth, N deposition and effects of climate change such as the lengthening of the growing
season in northern temperate and boreal areas. $S_{LAND}$ does not include gross land sinks directly
resulting from land-use change (e.g. regrowth of vegetation) as these are estimated as part of the
net land use flux ($E_{LUC}$). System boundaries make it difficult to attribute exactly $CO_2$ fluxes on land
between $S_{LAND}$ and $E_{LUC}$ (Erb et al., 2013), and by design most of the uncertainties in our method
are allocated to $S_{LAND}$ for those processes that are poorly known or represented in models.

### 2.5.1   Residual of the budget

For 1959-2015, the terrestrial carbon sink was estimated from the residual of the other budget
terms by rearranging Eq. (1):

$$S_{LAND} = E_{FF} + E_{LUC} - (G_{ATM} + S_{OCEAN}) \hspace{2cm} (8)$$

The uncertainty in $S_{LAND}$ is estimated annually from the root sum of squares of the uncertainty in
the right-hand terms assuming the errors are not correlated. The uncertainty averages to ± 0.8
GtC yr$^{-1}$ over 1959-2015 (Table 7). $S_{LAND}$ estimated from the residual of the budget includes, by
definition, all the missing processes and potential biases in the other components of Eq. (8).
**2.5.2   DGVMs**
A comparison of the residual calculation of $S_{LAND}$ in Eq. (8) with estimates from DGVMs as used to
estimate $E_{LUC}$ in Sect. 2.2.3, but here excluding the effects of changes in land cover (using a
constant pre-industrial land cover distribution), provides an independent estimate of the
consistency of $S_{LAND}$ with our understanding of the functioning of the terrestrial vegetation in
response to $CO_2$ and climate variability (Table 7). As described in Sect. 2.2.3, the DGVM runs that
exclude the effects of changes in land cover include all climate variability and $CO_2$ effects over
land, but do not include reductions in $CO_2$ sink capacity associated with human activity directly
affecting changes in vegetation cover and management, which by design is allocated to $E_{LUC}$. This
effect has been estimated to have led to a reduction in the terrestrial sink by 0.5 GtC yr$^{-1}$ since
1750 (Gitz and Ciais, 2003). The models in this configuration estimate the mean and variability of
$S_{LAND}$ based on atmospheric $CO_2$ and climate, and thus both terms can be compared to the budget
residual. We apply three criteria for minimum model realism by including only those models
with (1) steady state after spin up, (2) where available, net land fluxes ($S_{LAND} - E_{LUC}$) that is a
carbon sink over the 1990s as constrained by global atmospheric and oceanic observations
(McNeill et al 2003, Manning and Keeling 2006, Mikaloff-Fletcher et al., 2006), and (3) where
available global $E_{LUC}$ that is a carbon source over the 1990s.  Fourteen models met criteria (1) and
five of the models that provided $E_{LUC}$ met all three criteria.
The annual standard deviation of the $CO_2$ sink across the DGVMs averages to ± 0.8 GtC yr$^{-1}$ for the
period 1959 to 2015. The model mean, over different decades, correlates with the budget residual
with r = 0.68 (0.51 to r = 0.77 for individual models). The standard deviation is similar to that of
the five model ensembles presented in Le Quéré et al. (2009), but the correlation is improved
compared to r = 0.54 obtained in the earlier study. The DGVM results suggest that the sum of our
knowledge on annual $CO_2$ emissions and their partitioning is plausible (see Discussion), and
provide insight on the underlying processes and regional breakdown. However as the standard
deviation across the DGVMs (0.8 GtC yr$^{-1}$ on average) is of the same magnitude as the combined
uncertainty due to the other components ($E_{FF}$, $E_{LUC}$, $G_{ATM}$, $S_{OCEAN}$; Table 7), the DGVMs do not
provide further reduction of uncertainty on the annual terrestrial $CO_2$ sink compared to the
residual of the budget (Eq. 8). Yet, DGVM results are largely independent from the residual of the
budget, and it is worth noting that the residual method and ensemble mean DGVM results are
consistent within their respective uncertainties. We attach a medium confidence level to the
annual land $CO_2$ sink and its uncertainty because the estimates from the residual budget and
averaged DGVMs match well within their respective uncertainties, and the estimates based on the
residual budget are primarily dependent on $E_{FF}$ and $G_{ATM}$, both of which are well constrained.
## 2.6    The atmospheric perspective
The world-wide network of atmospheric measurements can be used with atmospheric inversion
methods to constrain the location of the combined total surface $CO_2$ fluxes from all sources,
including fossil and land-use change emissions and land and ocean $CO_2$ fluxes. The inversions
assume $E_{FF}$ to be well known, and they solve for the spatial and temporal distribution of land and
ocean fluxes from the residual gradients of $CO_2$ between stations that are not explained by
emissions. Inversions used atmospheric $CO_2$ data to the end of 2015 (including preliminary values
in some cases), and three atmospheric $CO_2$ inversions (Table 6) to infer the total $CO_2$ flux over land
regions, and the distribution of the total land and ocean $CO_2$ fluxes for the mid-high latitude
northern hemisphere (30°N-90°N), Tropics (30°S-30°N) and mid-high latitude region of the
southern hemisphere (30°S-90°S). We focus here on the largest and most consistent sources of
information, and use these estimates to comment on the consistency across various data streams
and process-based estimates.
### 2.6.1    Atmospheric inversions
The three inversion systems used in this release are the CarbonTracker (Peters et al., 2010), the
Jena CarboScope (Rödenbeck, 2005), and CAMS (Chevallier et al., 2005). See Table 6 for version
numbers. They are based on the same Bayesian inversion principles that interpret the same, for
the most part, observed time series (or subsets thereof), but use different methodologies that
represent some of the many approaches used in the field. This mainly concerns the time
resolution of the estimates (i.e. weekly or monthly), spatial breakdown (i.e. grid size), assumed
correlation structures, and mathematical approach. The details of these approaches are
documented extensively in the references provided. Each system uses a different transport
model, which was demonstrated to be a driving factor behind differences in atmospheric-based
flux estimates, and specifically their global distribution (Stephens et al., 2007).
The three inversions use atmospheric $CO_2$ observations from various flask and in situ networks.
They prescribe spatial and global $E_{FF}$ that can vary from that presented here. The CarbonTracker
and CAMS inversions prescribed the same global $E_{FF}$ than in section 2.1.1, during 2010-2015 for
CarbonTracker, and during 1979-2015 in CAMS. The Jena CarboScope inversion uses $E_{FF}$ from
EDGAR (2011) v4.2. Different spatial and temporal distributions of $E_{FF}$ were prescribed in each
inversion.
Given their prescribed map of $E_{FF}$, each inversion estimates natural fluxes from a similar set of
surface $CO_2$ measurement stations, and CarbonTracker additionally uses two sites of aircraft $CO_2$
vertical profiles over the Amazon and Siberia, regions where surface observations are sparse. The
atmospheric transport models of each inversion are TM5 for CarbonTracker, TM3 for Jena
CarboScope, and LMDZ for CAMS. These three models are based on the same ECMWF wind fields.
The three inversions use different prior natural fluxes, which partly influences their optimized
fluxes. CAMS assumes that the prior land flux is zero on the annual mean in each grid cell of the
transport model, so that any sink or source on land is entirely reflecting the information brought
by atmospheric measurements. CarbonTracker simulates a small prior sink on land from the
SIBCASA model that results from regrowth following fire disturbances of an otherwise net zero
biosphere. Jena CarboScope assumes a prior sink on land as well from the LPJ model. Inversion
results for the sum of natural ocean and land fluxes (Fig. 8) are more constrained in the Northern
hemisphere (NH) than in the Tropics, because of the higher measurement stations density in the
NH.
Finally, results from atmospheric inversions include the natural $CO_2$ fluxes from rivers (which need
to be taken into account to allow comparison to other sources), and chemical oxidation of
reactive carbon-containing gases (which are neglected here). These inverse estimates are not truly
independent of the other estimates presented here as the atmospheric observations include a set
of observations used to estimate the global growth rate in atmospheric $CO_2$ concentration
(Section 2.3). However they provide new information on the regional distribution of fluxes.
We focus the analysis on two known strengths of the inverse approach: the derivation of the year-
to-year changes in total land fluxes ($E_{LUC} + S_{LAND}$) consistent with the whole network of
atmospheric observations, and the spatial breakdown of land and ocean fluxes ($E_{LUC} + S_{LAND} +$
$S_{OCEAN}$) across large regions of the globe. The spatial breakdown is discussed in Section 3.1.3.
**2.7    Processes not included in the global carbon budget**
**2.7.1    Contribution of anthropogenic CO and CH$_4$ to the global carbon budget**
Anthropogenic emissions of CO and CH$_4$ to the atmosphere are eventually oxidized to CO$_2$ and
thus are part of the global carbon budget. These contributions are omitted in Eq. (1), but an
attempt is made in this section to estimate their magnitude, and identify the sources of
uncertainty. Anthropogenic CO emissions are from incomplete fossil fuel and biofuel burning and
deforestation fires. The main anthropogenic emissions of fossil CH$_4$ that matter for the global
carbon budget are the fugitive emissions of coal, oil and gas upstream sectors (see below). These
emissions of CO and CH$_4$ contribute a net addition of fossil carbon to the atmosphere.
In our estimate of E$_{FF}$ we assumed (Section 2.1.1) that all the fuel burned is emitted as CO$_2$, thus
CO anthropogenic emissions and their atmospheric oxidation into CO$_2$ within a few months are
already counted implicitly in E$_{FF}$ and should not be counted twice (same for E$_{LUC}$ and
anthropogenic CO emissions by deforestation fires). Anthropogenic emissions of fossil CH$_4$ are not
included in E$_{FF}$, because these fugitive emissions are not included in the fuel inventories. Yet they
contribute to the annual CO$_2$ growth rate after CH$_4$ gets oxidized into CO$_2$. Anthropogenic
emissions of fossil CH$_4$ represent 15% of total CH$_4$ emissions (Kirschke et al., 2013) that is 0.061
GtC yr$^{-1}$ for the past decade. Assuming steady state, these emissions are all converted to CO$_2$ by
OH oxidation, and thus explain 0.06 GtC yr$^{-1}$ of the global CO$_2$ growth rate in the past decade.
Other anthropogenic changes in the sources of CO and CH$_4$ from wildfires, biomass, wetlands,
ruminants or permafrost changes are similarly assumed to have a small effect on the CO$_2$ growth
rate.
**2.7.2    Anthropogenic carbon fluxes in the land to ocean aquatic continuum**
The approach used to determine the global carbon budget considers only anthropogenic CO$_2$
emissions and their partitioning among the atmosphere, ocean and land. In this analysis, the land
and ocean reservoirs that take up anthropogenic CO$_2$ from the atmosphere are conceived as
independent carbon storage repositories. This approach thus omits that carbon is continuously
displaced along the land-ocean aquatic continuum (LOAC) comprising freshwaters, estuaries and
coastal areas (Bauer et al., 2013; Regnier et al., 2013). A significant fraction of this lateral carbon
flux is entirely 'natural' and is thus a steady state component of the pre-industrial carbon cycle.
However, changes in environmental conditions and land use change have caused an increase in
the lateral transport of C into the LOAC – a perturbation that is relevant for the global carbon
budget presented here.
The results of the analysis of Regnier et al. (2013) can be summarized in three points of relevance
to the anthropogenic $CO_2$ budget. First, the anthropogenic carbon input from land to
hydrosphere, $F_{LH}$, estimated at $1 \pm 0.5$ GtC yr$^{-1}$ implies that a portion of the anthropogenic $CO_2$
taken up by land ecosystems by perturbed Net Primary Productivity is not sequestered in soil and
biomass pools but exported to the LOAC. Second, some of the exported anthropogenic carbon
remains stored in the LOAC ($\Delta C_{LOAC}$, $0.55 \pm 0.3$ GtC yr$^{-1}$) and some is released back to the
atmosphere as $CO_2$ ($E_{LOAC}$, $0.35 \pm 0.2$ GtC yr$^{-1}$), the magnitude of these fluxes resulting from the
combined effects of freshwaters, estuaries and coastal seas. Third, a small fraction of
anthropogenic carbon displaced by the LOAC is transferred to the open ocean where it
accumulates ($F_{HO}$, $0.1 \pm > 0.05$ GtC yr$^{-1}$). The anthropogenic perturbation of the carbon fluxes from
land to ocean does not contradict the method used in Section 2.5 to define the ocean sink and
residual terrestrial sink. However, it does point to the need to account for the fate of
anthropogenic carbon once it is removed from the atmosphere by land ecosystems (summarized
in Fig 2). In theory, direct estimates of changes of the ocean inorganic carbon inventory over time
would see the land flux of anthropogenic carbon and would thus have a bias relative to air-sea flux
estimates and tracer based reconstructions. However, currently the value is small enough to be
not noticeable relative to the errors in the individual techniques.
The residual terrestrial sink in a budget that accounts for the LOAC will be larger than $S_{LAND}$, as the
flux is partially offset by the net source of $CO_2$ to the atmosphere, i.e. $E_{LOAC}$, of $0.35 \pm 0.3$ GtC yr$^{-1}$
from rivers, estuaries and coastal seas:

$$S_{LAND+LOAC} = E_{FF} + E_{LUC} - (G_{ATM} + S_{OCEAN}) + E_{LOAC} \qquad (9)$$

The residual terrestrial sink ($S_{LAND}$) is $3.0 \pm 0.8$ GtC yr$^{-1}$ for 2006-2015 as calculated according to Eq.
(8; Table 7) while $S_{LAND+LOAC}$ is $3.4 \pm 0.9$ GtC yr$^{-1}$ over the same time period. A fraction of
anthropogenic $CO_2$ taken up by land ecosystems is exported to the LOAC ($F_{LH}$). With the LOAC
included, we now have:

$$\Delta C_{TE} = S_{LAND+LOAC} - E_{LUC} - F_{LH} \qquad (10)$$

where $\Delta C_{TE}$ is the change in annual terrestrial ecosystems carbon storage, including land
vegetation, litter and soil, $\Delta C_{TE}$ is 1.4 GtC yr$^{-1}$ for the period 2006-2015 (Eqs (9) and (10)). It is
notably smaller than what would be calculated in a traditional budget that ignores the LOAC. In
this case, the change in carbon storage is estimated as 2.1± 0.7 Gt C yr$^{-1}$ from the difference
between $S_{LAND}$ (3.0 Gt ± 0.8 C yr$^{-1}$) and $E_{LUC}$ (1.0 ± 0.5 Gt C yr$^{-1}$;Table 8). All estimates of LOAC are
given with low confidence, because they originate from a single source. The carbon budget
presented here implicitly incorporates the fluxes from the LOAC with $S_{LAND}$. We do not attempt to
separate these fluxes because the uncertainties in either estimate are too large, and there is
insufficient information available to estimate the LOAC fluxes on an annual basis.
## 3   Results
### 3.1   Global carbon budget averaged over decades and its variability
The global carbon budget averaged over the last decade (2006-2015) is shown in Fig. 2. For this
time period, 91 of the total emissions ($E_{FF}$ + $E_{LUC}$) were caused by fossil fuels and industry, and 9%
by land-use change. The total emissions were partitioned among the atmosphere (44%), ocean
(26%) and land (30%). All components except land-use change emissions have grown since 1959
(Figs. 3 and 4), with important interannual variability in the growth rate in atmospheric $CO_2$
concentration and in the land $CO_2$ sink (Fig. 4), and some decadal variability in all terms (Table 8).
### 3.1.1   $CO_2$ emissions
Global $CO_2$ emissions from fossil fuels and industry have increased every decade from an average
of 3.1 ± 0.2 GtC yr$^{-1}$ in the 1960s to an average of 9.3 ± 0.5 GtC yr$^{-1}$ during 2006-2015 (Table 8 and
Fig. 5). The growth rate in these emissions decreased between the 1960s and the 1990s, from
4.5% yr$^{-1}$ in the 1960s (1960-1969), 2.8% yr$^{-1}$ in the 1970s (1970-1979), 1.9% yr$^{-1}$ in the 1980s
(1980-1989), and to 1.1% yr$^{-1}$ in the 1990s (1990-1999). After this period, the growth rate began
increasing again in the 2000s at an average growth rate of 3.5% yr$^{-1}$, decreasing to 1.8% yr$^{-1}$ for
the last decade (2006-2015). In contrast, $CO_2$ emissions from land-use change have remained
constant at around 1.5 ± 0.5 GtC yr$^{-1}$ between 1960-1999 and 1.0 ± 0.5 GtC yr$^{-1}$ during 2000-2015.
The decrease in emissions from land-use change between the 1990s and 2000s is highly uncertain.
This decrease is not found in the current ensemble of the DGVMs (Fig. 6), which are otherwise
consistent with the bookkeeping method within their respective uncertainty (Table 7). The
decrease is also not found in the study of tropical deforestation of Achard et al. (2014) where the
fluxes in the 1990s were similar to those of the 2000s and outside our uncertainty range. A new
study based on FAO data to 2015 (Federici et al., 2015) suggests that $E_{LUC}$ decreased during 2011-
2015 compared to 2001-2010.

### 3.1.2  Partitioning among the atmosphere, ocean and land

Emissions are partitioned among the atmosphere, ocean and land (Eq. 1). The growth rate in
atmospheric $CO_2$ level increased from 1.7 ± 0.1 GtC yr$^{-1}$ in the 1960s to 4.5 ± 0.1 GtC yr$^{-1}$ during
2006-2015 with important decadal variations (Table 8). Both ocean and land $CO_2$ sinks increased
roughly in line with the atmospheric increase, but with significant decadal variability on land
(Table 8). The ocean $CO_2$ sink increased from 1.2 ± 0.5 GtC yr$^{-1}$ in the 1960s to 2.6 ± 0.5 GtC yr$^{-1}$
during 2006-2015, with interannual variations of the order of a few tenths of GtC yr$^{-1}$ generally
showing an increased ocean sink during El Niño events (i.e. 1982-1983, 1997-1998, 2015-2016)
(Fig. 7; Rödenbeck et al., 2014). Although there is some coherence between the ocean models and
data products and among data products regarding the mean, decadal variability and trend, the
ocean models and data products show poor agreement for interannual variability (Section 2.4.3
and Fig. 7). As shown in Fig. 7, the two data products and most model estimates produce a mean
$CO_2$ sink for the 1990s that is below the mean assessed by the IPCC from indirect (but arguably
more reliable) observations (Denman et al., 2007; Section 2.4.1). This discrepancy suggests we
may need to reassess estimates of the mean ocean carbon sinks, with some implications for the
cumulative carbon budget (Landschützer et al., 2016).
The residual terrestrial $CO_2$ sink increased from 1.7 ± 0.7 GtC yr$^{-1}$ in the 1960s to 3.2 ± 0.8 GtC yr$^{-1}$
during 2006-2015, with important interannual variations of up to 2 GtC yr$^{-1}$ generally showing a
decreased land sink during El Niño events, overcompensating the increase in ocean sink and
accounting for the enhanced growth rate in atmospheric $CO_2$ concentration during El Niño events.
The high uptake anomaly around year 1991 is thought to be caused by the effect of the volcanic
eruption of Mount Pinatubo on climate and is not generally reproduced by the DGVMs, but it is
assigned to the land by the two inverse systems that include this period (Fig. 6). The larger land
$CO_2$ sink during 2006-2015 compared to the 1960s is reproduced by all the DGVMs in response to
combined atmospheric $CO_2$ increase, climate and variability, consistent with the budget residual
and reflecting a common knowledge of the processes (Table 7). The DGVM ensemble mean of 2.8
± 0.7 GtC yr$^{-1}$ also reproduces the observed mean for the period 2006-2015 calculated from the
budget residual (Table 7).
The total $CO_2$ fluxes on land ($E_{LUC}$ + $S_{LAND}$) constrained by the atmospheric inversions show in
general very good agreement with the global budget estimate, as expected given the strong
constrains of $G_{ATM}$ and the small relative uncertainty assumed on $S_{OCEAN}$ and $E_{FF}$ by inversions. The
total land flux is of similar magnitude for the decadal average, with estimates for 2006-2015 from
the three inversions of 2.2, 2.3 and 3.4 GtC yr$^{-1}$ compared to 2.1 ± 0.7 GtC yr$^{-1}$ for the total flux
computed with the carbon budget from other terms in Eq. 1 (Table 7). The total land sink from the
three inversions is 1.8, 1.8 and 3.0 GtC yr$^{-1}$ when including a mean river flux adjustment of 0.45
GtC yr$^{-1}$, though the exact adjustment is in fact smaller because the anthropogenic contribution to
river fluxes is only a fraction of the total river flux (Section 2.7.2). The interannual variability of the
inversions also matched the residual-based $S_{LAND}$ closely (Fig. 6). The total land flux from the
DGVM multi-model mean also compares well with the estimate from the carbon budget and
atmospheric inversions, with a decadal mean of 1.7 ± 0.5 GtC yr$^{-1}$ (Table 7; 2006-2015), although
individual models differ by several GtC for some years (Fig. 6).
### 3.1.3  Regional distribution
Fig 8 shows the partitioning of the total surface fluxes excluding emissions from fossil fuels and
industry ($S_{LAND}$ + $S_{OCEAN}$ - $E_{LUC}$) according to the process models in the ocean and on land, and to
the three atmospheric inversions. The total surface fluxes provide information on the regional
distribution of those fluxes by latitude bands (Fig. 8). The global mean $CO_2$ fluxes from process
models for 2006-2015 is 4.2 ± 0.6 GtC yr$^{-1}$. This is comparable to the fluxes of 4.8 ± 0.5 GtC yr$^{-1}$
inferred from the remainder of the carbon budget ($E_{FF}$ – $G_{ATM}$ in Equation 1; Table 8) within their
respective uncertainties. The total $CO_2$ fluxes from the three inversions range between 4.6 and 4.9
GtC yr$^{-1}$, consistent with the carbon budget as expected from the constraints on the inversions.
In the South (south of 30°S), the atmospheric inversions and process models all suggest a $CO_2$ sink
for 2006-2015 of between 1.2 and 1.6 GtC yr$^{-1}$ (Fig. 8), although the details of the interannual
variability are not fully consistent across methods. The interannual variability in the South is low
because of the dominance of ocean area with low variability compared to land areas.
In the Tropics (30°S-30°N), both the atmospheric inversions and process models suggest the
carbon balance in this region is close to neutral over the past decade, with fluxes for 2006-2015
ranging between –0.5 and +0.6 GtC yr$^{-1}$. Both the process based models and the inversions

1 consistently allocate more year-to-year variability of $CO_2$ fluxes to the Tropics compared to the

2 North (north of 30°N; Fig. 8), this variability being dominated by land fluxes.

3 In the North (north of 30°N), the inversions and process models are not in agreement on the

4 magnitude of the $CO_2$ sink with the ensemble mean of the process models suggesting a total

5 northern hemisphere sink for 2006-2015 of 2.3 ± 0.4 GtC yr$^{-1}$ while the three inversions estimate a

6 sink of 2.7, 3.8 and 3.8 GtC yr$^{-1}$. The mean difference can only partly be explained by the influence

7 of river fluxes, which is seen by the inversions but not included in the process models, as this flux

8 in the Northern Hemisphere would be less than 0.45 GtC yr$^{-1}$, particularly when only the

9 anthropogenic contribution to river fluxes is accounted for. The CarbonTracker inversion is close

10 to the one standard deviation of the process models for the mean sink during their overlap period.

11 CAMS and Jena CarboScope give a higher sink in the North than the process models, and a

12 correspondingly higher source in the Tropics. Differences between CarbonTracker and CAMS, Jena

13 CarboScope may be related to differences in inter-hemispheric mixing time of their transport

14 models, and other inversion settings. Differences between the mean fluxes of CAMS, Jena

15 CarboScope and the ensemble of process models cannot be simply explained. They could either

16 reflect a bias in these two inversions, or missing processes or biases in the process models, such as

17 the lack of adequate parameterizations for forest management in the North and for forest

18 degradation emissions in Tropics for the DGVMs.

19 The estimated contribution of the North and its uncertainty from process models is sensitive both

20 to the ensemble of process models used and to the specifics of each inversion. All three inversions

21 show substantial differences in variability and/or trend, and one inversion substantial difference

22 in the mean Northern sink.

23 ### 3.2 Global carbon budget for year 2015

24 ### 3.2.1 $CO_2$ emissions

25 Global $CO_2$ emissions from fossil fuels and industry remained nearly constant at 9.9 ± 0.5 GtC in

26 2015 (Fig. 5), distributed among coal (41%), oil (34%), gas (19%), cement (5.6%) and gas flaring

27 (0.7%). Compared to the previous year, emissions from coal and cement decreased by −1.8% and

28 −1.9%, respectively, while emissions from oil and gas increased by 1.9% and 1.7%, respectively.

29 Due to lack of data, gas flaring in 2014 and 2015 are assumed the same as 2013.

Growth in emissions in 2015 was not statistically different from zero, at 0.06% higher than in
2014, in stark contrast with the decadal average of 1.8% $yr^{-1}$ (2006-2015). Growth in 2015 is in the
range of our projection change of -0.6 [-1.6 to +0.5]% made last year (Le Quéré et al., 2015a)
based on national emissions projections for China and the USA, and projections of gross domestic
product corrected for $I_{FF}$ improvements for the rest of the world. However, the specific projection
for 2015 for China made last year (likely range of –4.6% to –1.1%) was for a larger decrease in
emissions than realised (–0.7%). This is due to lower decline in coal production in the last four
months of the year compared to January-August and to improvements in energy content of coal
at the top of the range.
In 2015, the largest contributions to global $CO_2$ emissions were from China (29%), the USA (15%),
the EU (28 member states; 10%), and India (6.3%). The percentages are the fraction of the global
emissions including bunker fuels (3.2%). These four regions account for 59% of global emissions.
Growth rates for these countries from 2014 to 2015 were –0.7% (China), –2.6% (USA), +1.4%
(EU28), and +5.2% (India). The per-capita $CO_2$ emissions in 2015 were 1.3 tC $person^{-1}$ $yr^{-1}$ for the
globe, and were 4.6 (USA), 2.1 (China), 1.9 (EU28) and 0.5 (India) tC $person^{-1}$ $yr^{-1}$ for the four
highest emitting countries (Fig. 5e).
Territorial emissions in Annex B countries decreased by –0.2% $yr^{-1}$ on average during 1990-2014.
Trends observed for consumption emissions were less monotonic, with 0.8% $yr^{-1}$ growth over
1990-2007 and a –1.5% $yr^{-1}$ decrease over 2007-2014 (Fig. 5c). In non-Annex B countries during
1990-2014, territorial emissions have grown at 4.7% $yr^{-1}$, while consumption emissions have
grown at 4.4% $yr^{-1}$. In 1990, 65% of global territorial emissions were emitted in Annex B countries
(33% in non-Annex B, and 2% in bunker fuels used for international shipping and aviation), while
in 2014 this had reduced to 37% (60% in non-Annex B, and 3% in bunker fuels). In terms of
consumption emissions this split was 67% in 1990 and 41% in 2014 (33% to 59% in non-Annex B).
The difference between territorial and consumption emissions (the net emission transfer via
international trade) from non-Annex B to Annex B countries has increased from near zero in 1990
to 0.3 GtC $yr^{-1}$ around 2005 and remained relatively stable afterwards until the last year available
(2014; Fig. 5). The increase in net emission transfers of 0.30 GtC $yr^{-1}$ between 1990 and 2014
compares with the emission reduction of 0.4 GtC $yr^{-1}$ in Annex B countries. These results show the
importance of net emission transfer via international trade from non-Annex B to Annex B
countries, and the stabilisation of emissions transfer when averaged over Annex B countries
during the past decade. In 2014, the biggest emitters from a consumption perspective were China
(25% of the global total), USA (16%), EU28 (12%), and India (5%).
Based on fire activity, the global $CO_2$ emissions from land-use change are estimated as 1.3 ± 0.5
GtC in 2015, slightly above the 2006-2015 average of 1.0 ± 0.5 GtC yr$^{-1}$. The slight rise in $E_{LUC}$ in
2015 is consistent with estimates of peat fires in Asia based on atmospheric data (Yin et al., in
press). However, the estimated annual variability is not generally consistent between methods,
except that all methods estimate that variability in $E_{LUC}$ is small relative to the variability from
$S_{LAND}$ (Fig. 6a). This could be partly due to the design of the DGVM experiments, which use flux
differences between simulations with and without land-cover change, and thus their variability
may differ e.g. due to fires in forest regions where the contemporary forest cover is smaller than
pre-industrial cover used in the 'without land cover change' runs.
### 3.2.2   Partitioning among the atmosphere, ocean and land
The growth rate in atmospheric $CO_2$ concentration was 6.2 ± 0.2 GtC in 2015 (2.92+/-0.09 ppm;
Fig. 4;Dlugokencky and Tans, 2016). This is well above the 2006-2015 average of 4.5 ± 0.1 GtC yr$^{-1}$
and reflects the large interannual variability in the growth rate of atmospheric $CO_2$ concentration
associated with El Niño events.
The ocean $CO_2$ sink was 3.0 ± 0.5 GtC yr$^{-1}$ in 2015, an increase of 0.15 GtC yr$^{-1}$ over 2015 according
to ocean models. Five of the seven ocean models produce an increase in the ocean $CO_2$ sink in
2015 compared to 2014, with near zero changes in the last two models (Fig. 7). However, of the
two data products available disagree over changes in the last year, Rödenbeck et al. (2014b)
produce a decrease of -0.4 GtC yr$^{-1}$ while Landschützer et al. (2015) produce an increase of 0.3 GtC
yr$^{-1}$. Thus there is no overall consistency in the annual change in the ocean $CO_2$ sink, although
there is an indication of increasing convergence among products for the assessment of multi-year
changes, as suggested by the time-series correlations reported in Section 2.4.3 (see also
Landschützer et al., 2015). An increase in the ocean $CO_2$ in 2015 sink would be consistent with the
observed El Niño conditions and continued rising atmospheric $CO_2$. All estimates suggest an ocean
$CO_2$ sink for 2015 that is larger than their 2006-2015 average.
The terrestrial $CO_2$ sink calculated as the residual from the carbon budget was 2.0 ± 0.9 GtC in
2015, well below the 3.2 ± 0.8 GtC yr$^{-1}$ averaged over 2006-2015 (Fig. 4), and reflecting the onset
of the El Niño conditions in the second half of 2015. The DGVM model mean produces a sink of 1.0
± 1.4 GtC in 2015, also well below the 2006-2015 average (Table 7). Both models and inversions
suggest that the lower sink in 2015 primarily originated in the tropics (Fig. 8).
**3.3   Emission projections and the global carbon budget for year 2016**
**3.3.1   CO$_2$ emissions**
Using separate projections for China, the USA, and the rest of the world as described in Section
2.1.4, we project that the growth in global CO$_2$ emissions from fossil fuels and cement production
will be near zero in 2016, with a change of +0.2% (range of –1.0% to +1.7%) from 2015 levels and
no leap year adjustment (Table 9). Our method is imprecise and contains several assumptions that
could influence the results beyond the given range, and as such is indicative only. Within the given
assumptions, global emissions remain nearly constant at 9.9 ± 0.5 GtC (36.4 ± 1.8 GtCO$_2$) in 2016,
but are still 63% above emissions in 1990. The drivers of the trends in E$_{FF}$ are discussed elsewhere
(Peters et al., submitted).
For China, the expected change based on available data during January to July or August (see
Section 2.1.4) is for an increase in emissions of – 0.6% (range of –3.4% to +1.1%) in 2016
compared to 2015, based on estimated decreases in coal consumption (–1.2%) and estimated
growth in apparent oil (+5.5%) and natural gas (+8.8%) consumption and in cement production
(+2.5%). The uncertainty range considers the spread between different data sources, and
differences between July/August and end-year data observed in 2014 and 2015. The estimated
reduction in coal consumption also incorporates an assumed 2% increase in the energy density of
coal—based on increases in the last two years, which are assumed to continue given production
limits in 2016 that are likely to affect production of low-quality coal more—and the uncertainty
range also reflects uncertainty in this figure.
For the USA, the EIA emissions projection for 2016 combined with cement data from USGS gives a
decrease of –1.5% (range of –4.0 to +1.0%) compared to 2015.
For the rest of the world, the expected growth for 2016 of +1.1% (range of –0.3 to +2.6%) is
computed using the GDP projection for the world excluding China and the USA of 2.4% made by
the IMF (IMF, 2016) and a decrease in I$_{FF}$ of –1.2% yr$^{-1}$ which is the average from 2006-2015. The
uncertainty range is based on the standard deviation of the interannual variability in I$_{FF}$ during
2006-2015 of ±1.0% yr$^{-1}$ and our estimate of uncertainty in the IMF's GDP forecast of ±0.5%.

### 3.3.2 Partitioning among the atmosphere, ocean and land

The growth in atmospheric $CO_2$ concentration ($G_{ATM}$) was projected to be high again in 2016, at 6.7 ± 1.1 GtC (3.15 ± 0.53 ppm) for the Mauna Loa station (Betts et al., 2016). Growth at Mauna Loa is closely correlated with the global growth ($r^2$=0.90 for 1959-2015). Therefore, the global growth rate in atmospheric $CO_2$ concentration is also expected to be high in 2016. The observed global growth in atmospheric $CO_2$ concentration between December 2015 and June 2016 was already 2.1 ppm (Dlugokencky and Tans, 2016) after seasonal adjustment (for a 6 months period), supporting the projection of Betts et al., even with a return to El Niño neutral or possible emerging La Niña conditions for the second half of 2016.

Combining projected $E_{FF}$ and $G_{ATM}$ suggests a total for the combined land and ocean ($S_{LAND} + S_{OCEAN} - E_{LUC}$) of about 3 GtC only. $S_{OCEAN}$ was 3.0 GtC in 2015 and is expected to slightly increase in 2016 from a delayed response to El Niño conditions (Feely et al., 1999). $E_{LUC}$ was 1.3 GtC in 2015, above the decadal mean average of 1.0 GtC $yr^{-1}$, and is expected to return to average or below average in 2016 based on fire activity related to land management so far (up to August). Hence for 2016, the residual land sink $S_{LAND}$, should be well below its 2006-2015 average, and approximately balance $E_{LUC}$. This is consistent with our understanding of the response of the terrestrial vegetation to El Niño conditions and increasing atmospheric $CO_2$ concentrations.

### 3.4 Cumulative emissions

Cumulative emissions for 1870-2015 were 410 ± 20 GtC for $E_{FF}$, and 145± 50 GtC for $E_{LUC}$ based on the bookkeeping method of Houghton et al. (2012) for 1870-1996 and a combination with fire-based emissions for 1997-2015 as described in Section 2.2 (Table 10). The cumulative emissions are rounded to the nearest 5 GtC. The total cumulative emissions from fossil and land use change for 1870-2015 are 560 ± 55 GtC. These emissions were partitioned among the atmosphere (235 ± 5 GtC based on atmospheric measurements in ice cores of 288 ppm (Section 2.3.1; Joos and Spahni, 2008) and recent direct measurements of 399.1 ppm (Dlugokencky and Tans, 2016), ocean (155 ± 20 GtC using Khatiwala et al. (2013) prior to 1959 and Table 8 otherwise), and the land (165 ± 60 GtC by the difference).

Cumulative emissions for the early period 1750-1869 were 3 GtC for $E_{FF}$, and about 45 GtC for $E_{LUC}$ (rounded to nearest 5) of which 10 GtC were emitted in the period 1850-1870 (Houghton et al. 2012) and 30 GtC were emitted in the period 1750-1850 based on the average of four publications

(22 GtC by Pongratz et al. (2009); 15 GtC by van Minnen et al. (2009); 64 GtC by Shevliakova et al.
(2009) and 24 GtC by Zaehle et al. (2011)). The growth rate in atmospheric $CO_2$ concentration
during that time was about 25 GtC, and the ocean uptake about 20 GtC, implying a land uptake of
5 GtC. These numbers have large relative uncertainties but balance within the limits of our
understanding.
Cumulative emissions for 1750-2015 based on the sum of the two periods above (before rounding
to the nearest five GtC) were 415 ± 20 GtC for $E_{FF}$, and 190 ± 65 GtC for $E_{LUC}$, for a total of 605 ± 70
GtC, partitioned among the atmosphere (260 ± 5 GtC), ocean (175 ± 20 GtC), and the land (170 ±
70 GtC).
Cumulative emissions through to year 2016 can be estimated based on the 2016 projections of $E_{FF}$
(Section 3.2), the largest contributor, and assuming a constant $E_{LUC}$ of 1.0 GtC (average of last
decade). For 1870–2016, these are 570 ± 55 GtC (2085 ± 205 $GtCO_2$) for total emissions, with
about 75% contribution from $E_{FF}$ (420 ± 20 GtC) and about 25% contribution from $E_{LUC}$ (150 ± 50
GtC). Cumulative emissions since year 1870 are higher than the emissions of 515 [445 to 585] GtC
reported in the IPCC (Stocker et al., 2013) because they include an additional 55 GtC from
emissions in 2012-2016 (mostly from $E_{FF}$). The uncertainty presented here (±1σ) is smaller than
the range of 90% used by IPCC, but both estimates overlap within their uncertainty ranges.
**4    Discussion**
Each year when the global carbon budget is published, each component for all previous years is
updated to take into account corrections that are the result of further scrutiny and verification of
the underlying data in the primary input data sets. The updates have generally been relatively
small and focused on the most recent years, except for land-use change, where they are more
significant but still generally within the provided uncertainty range (Fig. 9). The difficulty in
accessing land-cover change data to estimate $E_{LUC}$ is the key problem to providing continuous
records of emissions in this sector. Current FAO estimates are based on statistics reported at the
country level and are not spatially-explicit. Advances in satellite recovery of land-cover change
could help to keep track of land-use change through time (Achard et al., 2014; Harris, 2012).
Revisions in $E_{LUC}$ for the 2008/2009 budget were the result of the release of FAO 2010, which
contained a major update to forest cover change for the period 2000-2005 and provided the data
for the following 5 years to 2010 (Fig. 9b). The differences this year could be attributable to both
the different data and the different methods. Comparison of global carbon budget components
released annually by GCP since 2006 show that update differences were highest at 0.82 GtC yr$^{-1}$
for the growth rate in atmospheric $CO_2$ concentration (from a one-off correction back to year
1979), 0.24 GtC yr$^{-1}$ for fossil fuels and industry, and 0.52 GtC yr$^{-1}$ for the ocean $CO_2$ sink (from a
change from one to multiple models; Fig. 9d). The update for the residual land $CO_2$ sink was also
large (Fig. 9e), with a maximum value of 0.83 GtC yr$^{-1}$, directly reflecting revisions in other terms
of the budget.
Our capacity to separate the carbon budget components can be evaluated by comparing the land
$CO_2$ sink estimated through two approaches: (1) the budget residual ($S_{LAND}$), which includes errors
and biases from all components, and (2) the land $CO_2$ sink estimate by the DGVM ensemble, which
is based on our understanding of processes of how the land responds to increasing $CO_2$, climate
and variability. Furthermore, the inverse model estimates based on atmospheric $CO_2$ observations
can provide constraints on the total land flux ($S_{LAND} - E_{LUC}$). These estimates are generally close
(Fig. 6), both for the global mean and for the interannual variability. The annual estimates from
the DGVM of the residual terrestrial sink over 1959 to 2015 correlate with the annual budget
residual with r = 0.68 (Section 2.5.2; Fig. 6). The DGVMs produce a decadal mean and standard
deviation across models of 2.8 ± 0.7 GtC yr$^{-1}$ for the period 2006-2015, consistent with the
estimate of 3.0 ± 0.8 GtC yr$^{-1}$ produced with the budget residual (Table 7). New insights into total
surface fluxes arise from the comparison with the atmospheric inversions and their regional
breakdown already provide a semi-independent way to validate the results. The comparison
shows a first-order consistency between inversions and process models but with a lot of
discrepancies, particularly for the allocation of the mean land sink between the tropics and the
Northern hemisphere. Understanding these discrepancies and further analysis of regional carbon
budgets would provide additional information to quantify and improve our estimates, as has been
undertaken by the project REgional Carbon Cycle Assessment and Processes (RECAPP; Canadell et
al., 2012-2013).
Annual estimates of each component of the global carbon budgets have their limitations, some of
which could be improved with better data and/or better understanding of carbon dynamics. The
primary limitations involve resolving fluxes on annual time scales and providing updated estimates
for recent years for which data-based estimates are not yet available or only beginning to emerge.
Of the various terms in the global budget, only the burning of fossil fuels and the growth rate in
atmospheric $CO_2$ concentration terms are based primarily on empirical inputs supporting annual
estimates in this carbon budget. While these models represent the current state of the art, they
provide only simulated changes in primary carbon budget components. For example, the decadal
trends in global ocean uptake and the interannual variations associated with El Niño-Southern
Ocean Oscillation (e.g. ENSO) are not directly constrained by observations, although many of the
processes controlling these trends are sufficiently well known that the model-based trends still
have value as benchmarks for further validation. Data-based products for the ocean $CO_2$ sink
provide new ways to evaluate the model results, and could be used directly as data become more
rapidly available and methods for creating such products improve. However, there are still large
discrepancies among data-based estimates, in large part due to the lack of routine data sampling,
that preclude their direct use for now (see Rödenbeck et al., 2015). Estimates of land-use
emissions and their year-to-year variability have even larger uncertainty, and much of the
underlying data are not available as an annual update. Efforts are underway to work with annually
available satellite area change data or FAO reported data in combination with fire data and
modelling to provide annual updates for future budgets.
Our approach also depends on the reliability of the energy and land-cover change statistics
provided at the country level, which are potentially subject to biases. Thus it is critical to develop
multiple ways to estimate the carbon balance at the global and regional level, including estimates
from the inversion of atmospheric $CO_2$ concentration, the use of other oceanic and atmospheric
tracers, and the compilation of emissions using alternative statistics (e.g. sectors). It is also
important to challenge the consistency of information across observational streams, for example
to contrast the coherence of temperature trends with those of $CO_2$ sink trends. Multiple
approaches ranging from global to regional scale would greatly help increase confidence and
reduce uncertainty in $CO_2$ emissions and their fate.
**5    Conclusions**
The estimation of global $CO_2$ emissions and sinks is a major effort by the carbon cycle research
community that requires a combination of measurements and compilation of statistical estimates
and results from models. The delivery of an annual carbon budget serves two purposes. First,
there is a large demand for up-to-date information on the state of the anthropogenic perturbation
of the climate system and its underpinning causes. A broad stakeholder community relies on the
data sets associated with the annual carbon budget including scientists, policy makers, businesses,
journalists, and the broader society increasingly engaged in adapting to and mitigating human-
driven climate change. Second, over the last decade we have seen unprecedented changes in the
human and biophysical environments (e.g. increase in the growth of fossil fuel emissions, ocean
temperatures, and strength of the land sink), which call for more frequent assessments of the
state of the Planet, and by implications a better understanding of the future evolution of the
carbon cycle, and the requirements for climate change mitigation and adaptation. Both the ocean
and the land surface presently remove a large fraction of anthropogenic emissions. Any significant
change in the function of carbon sinks is of great importance to climate policymaking, as they
affect the excess carbon dioxide remaining in the atmosphere and therefore the compatible
emissions for any climate stabilization target. Better constraints of carbon cycle models against
contemporary data sets raises the capacity for the models to become more accurate at future
projections.
This all requires more frequent, robust, and transparent data sets and methods that can be
scrutinized and replicated. After 11 annual releases from the GCP, the effort is growing and the
traceability of the methods has become increasingly complex. Here, we have documented in
detail the data sets and methods used to compile the annual updates of the global carbon budget,
explained the rationale for the choices made, the limitations of the information, and finally
highlighted need for additional information where gaps exist.
This paper via 'living data' will help to keep track of new budget updates. The evolution over time
of the carbon budget is now a key indicator of the anthropogenic perturbation of the climate
system, and its annual delivery joins a set of other climate indicators to monitor the evolution of
human-induced climate change, such as the annual updates on the global surface temperature,
sea level rise, minimum Arctic sea ice extent among others.
**Data access**
The data presented here are made available in the belief that their wide dissemination will lead to
greater understanding and new scientific insights of how the carbon cycle works, how humans are
altering it, and how we can mitigate the resulting human-driven climate change. The free
availability of these data does not constitute permission for publication of the data. For research
projects, if the data are essential to the work, or if an important result or conclusion depends on
the data, co-authorship may need to be considered. Full contact details and information on how
to cite the data are given at the top of each page in the accompanying database, and summarised
in Table 2.
The accompanying database includes two Excel files organised in the following spreadsheets
(accessible with the free viewer http://www.microsoft.com/en-us/download/details.aspx?id=10):
File Global_Carbon_Budget_2015.xlsx includes:
1.  Summary
2.  The global carbon budget (1959-2015);
3.  Global $CO_2$ emissions from fossil fuels and cement production by fuel type, and the per-capita

9       emissions (1959-2015);

4.  $CO_2$ emissions from land-use change from the individual methods and models (1959-2015);
5.  Ocean $CO_2$ sink from the individual ocean models and data products (1959-2015);
6.  Terrestrial residual $CO_2$ sink from the DGVMs (1959-2015);
7.   Additional information on the carbon balance prior to 1959 (1750-2015).
File National_Carbon_Emissions_2015.xlsx includes:
1.  Summary
2.  Territorial country $CO_2$ emissions from fossil fuels and industry (1959-2015) from CDIAC,
extended to 2015 using BP data;
3.  Territorial country $CO_2$ emissions from fossil fuels and industry (1959-2015) from CDIAC with
UNFCCC data overwritten where available, extended to 2015 using BP data;
4.  Consumption country $CO_2$ emissions from fossil fuels and industry and emissions transfer
from the international trade of goods and services (1990-2014) using CDIAC/UNFCCC data
(worksheet 3 above) as reference;
5.  Emissions transfers (Consumption minus territorial emissions; 1990-2014);
6.  Country definitions;
7.  Details of disaggregated countries;
8.  Details of aggregated countries.
National emissions data are also available from the Global Carbon Atlas (globalcarbonatlas.org).
**Acknowledgments** We thank all people and institutions who provided the data used in this carbon
budget, C Enright and W Peters for their involvement in the development, use and analysis of the models
and data-products used here, and F Joos and S Khatiwala for providing historical data. We thank E.
Dlugokencky who provided the atmospheric $CO_2$ measurements used here, B. Pfeil, C. Landa, and S. Jones
of the Bjerknes Climate Data Centre and the ICOS Ocean Thematic Centre data management at the
University of Bergen who helped with gathering information from the SOCAT community, D. Bakker for
support to the SOCAT coordination, and all those involved in collecting and providing oceanographic $CO_2$
measurements used here, in particular for the new ocean data for years 2015: A. Andersson, N. Bates, R.
Bott, A. Cattrijsse, E. De Carlo, C. Dietrich, L. Gregor, C. Hunt, T. Johannessen, W.R. Joubert, A. Kuwata, S.K.
Lauvset, C. Lo Monaco, S. Maenner, D. Manzello, N. Monacci, S. Musielewicz, T. Newberger, A. Olsen, J.
Osborne, C. Sabine, S.C. Sutherland, C. Sweeney, K. Tadokoro, S. van Heuven, D. Vandemark, R.
Wanninkhof. We thank the institutions and funding agencies responsible for the collection and quality
control of the data included in SOCAT, and the support of the International Ocean Carbon Coordination
Project (IOCCP), the Surface Ocean Lower Atmosphere Study (SOLAS), and the Integrated Marine
Biogeochemistry, Ecosystem Research program (IMBER). We thank data providers to ObsPack
GLOBALVIEWplus v1.0 and NRT v3.0 for atmospheric $CO_2$ observations used in CTE2016-FT. This is NOAA-
PMEL contribution number 4576.
Finally we thank all funders who have supported the individual and joint contributions to this work (see
Table 11).

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

**Tables**
**Table 1.** Factors used to convert carbon in various units (by convention, Unit 1 = Unit 2 ˙
conversion).

| Unit 1 | Unit 2 | Conversion | Source |
|---|---|---|---|
| GtC (gigatonnes of carbon) | ppm (parts per million)[a] | 2.12[b] | Ballantyne et al. (2012) |
| GtC (gigatonnes of carbon) | PgC (petagrams of carbon) | 1 | SI unit conversion |
| $GtCO_2$ (gigatonnes of carbon dioxide) | GtC (gigatonnes of carbon) | 3.664 | 44.01/12.011 in mass equivalent |
| GtC (gigatonnes of carbon) | MtC (megatonnes of carbon) | 1000 | SI unit conversion |

[a] Measurements of atmospheric $CO_2$ concentration have units of dry-air mole fraction. 'ppm' is an
abbreviation for micromole/mol, dry air.
[b]The use of a factor of 2.12 assumes that all the atmosphere is well mixed within one year. In reality, only
the troposphere is well mixed and the growth rate of $CO_2$ concentration in the less well-mixed stratosphere
is not measured by sites from the NOAA network. Using a factor of 2.12 makes the approximation that the
growth rate of $CO_2$ concentration in the stratosphere equals that of the troposphere on a yearly basis.

1 **Table 2.** How to cite the individual components of the global carbon budget presented here.

| Component | Primary reference |
| --- | --- |
| Global emissions from fossil fuels and industry ($E_{FF}$), total and by fuel type | Boden and Andres (2016; CDIAC; cdiac.ornl.gov/trends/emis/meth_reg.html) |
| National territorial emissions from fossil fuels and industry ($E_{FF}$) | CDIAC source: Boden and Andres (2016; as above) UNFCCC source: (2016; http://unfccc.int/national_reports/annex_i_ghg_inventories/national_inventories_submissions/items/8108.php; accessed June 2016) |
| National consumption-based emissions from fossil fuels and industry ($E_{FF}$) by country (consumption) | Peters et al. (2011b) updated as described in this paper |
| Land-use change emissions ($E_{LUC}$) | Houghton et al. (2012) combined with Giglio et al. (2013) |
| Growth rate in atmospheric $CO_2$ concentration ($G_{ATM}$) | Dlugokencky and Tans (2016; NOAA/ESRL: www.esrl.noaa.gov/gmd/ccgg/trends/global; accessed July 2016) |
| Ocean and land $CO_2$ sinks ($S_{OCEAN}$ and $S_{LAND}$) | This paper for $S_{OCEAN}$ and $S_{LAND}$ and references in Table 6 for individual models. |

**Table 3.** Main methodological changes in the global carbon budget since first publication. Unless specified below, the methodology was identical to that described in the current paper. Furthermore, methodological changes introduced in one year are kept for the following years unless noted. Empty cells mea there were no methodological changes introduced that year.

| Publication year[a] | Fossil fuel emissions | | | LUC emissions | Reservoirs | | | Uncertainty & other changes |
|---|---|---|---|---|---|---|---|---|
| | Global | Country (territorial) | Country (consumption) | | Atmosphere | Ocean | Land | |
| 2006 Raupach et al. (2007) | | Split in regions | | | | | | |
| 2007 Canadell et al. (2007) | | | | $E_{LUC}$ based on FAO-FRA 2005; constant $E_{LUC}$ for 2006 | 1959-1979 data from Mauna Loa; data after 1980 from global average | Based on one ocean model tuned to reproduced observed 1990s sink | | ±1σ provided for all components |
| 2008 (online) 2009 Le Quéré et al. (2009) | | Split between Annex B and non-Annex B | Results from an independent study discussed | Constant $E_{LUC}$ for 2007 Fire-based emission anomalies used for 2006-2008 | | Based on four ocean models normalised to observations with constant delta | First use of five DGVMs to compare with budget residual | |
| 2010 Friedlingstein et al. (2010) | Projection for current year based on GDP | Emissions for top emitters | | $E_{LUC}$ updated with FAO-FRA 2010 | | | | |
| 2011 Peters et al. (2012b) 2012 Le Quéré et al. (2013) Peters et al. (2013) | | 129 countries from 1959 | Split between Annex B and non-Annex B 129 countries and regions from 1990-2010 based on GTAP8.0 | $E_{LUC}$ for 1997-2011 includes interannual anomalies from fire-based emissions | All years from global average | Based on 5 ocean models normalised to observations with ratio | Ten DGVMs available for $S_{LAND}$; First use of four models to compare with $E_{LUC}$ | |
| 2013 Le Quéré et al. (2014) | | 250 countries[b] | 134 countries and regions 1990-2011 based on GTAP8.1, with detailed estimates for years 1997, 2001, 2004, and 2007 | $E_{LUC}$ for 2012 estimated from 2001-2010 average | | Based on six models compared with two data-products to year 2011 | Coordinated DGVM experiments for $S_{LAND}$ and $E_{LUC}$ | Confidence levels; cumulative emissions; budget from 1750 |
| 2014 Le Quéré et al. (2015b) | Three years of BP data | Three years of BP data | Extended to 2012 with updated GDP data | $E_{LUC}$ for 1997-2013 includes interannual anomalies from fire-based emissions | | Based on seven models compared with three data-products to year 2013 | Based on ten models | Inclusion of breakdown of the sinks in three latitude bands and comparison with three atmospheric inversions |
| 2015 Le Quéré et al. (2015a) Jackson et al. (2016) | Projection for current year based Jan-Aug data | National emissions from UNFCCC extended to 2014 also provided (along with CDIAC) | Detailed estimates introduced for 2011 based on GTAP9 | | | Based on eight models compared with two data-products | Based on ten models with assessment of minimum realism | The decadal uncertainty for the DGVM ensemble mean now uses ±1σ of the decadal spread across models |
| 2016 (this study) | Two years of BP data; CHN emissions from 1990 from BP data | Added three small countries | | Preliminary $E_{LUC}$ using FRA-2015 shown for comparison; use of five DGVMs | | Based on seven models compared with two data-products | Based on fourteen models | Discussion of projection for full budget for current year |

[a]The naming convention of the budgets has changed. Up to and including 2010, the budget year (Carbon Budget 2010) represented the latest year of the data. From 2012, the budget year (Carbon Budget 2012) refers to the initial publication year.

[b]The CDIAC database has about 250 countries, but we show data for 219 countries since we aggregate and disaggregate some countries to be consistent with current country definitions (see Sect. 2.1.1 for more details).

1    **Table 4.** Data sources used to compute each component of the global carbon budget.

| Component | Process | Data source | Data reference |
|---|---|---|---|
| $E_{FF}$ (global and CDIAC national) | Fossil fuel combustion and gas flaring | UN Statistics Division to 2013 | UN (2014a, b) |
| | | BP for 2014-2015 | BP (BP, 2016b) |
| | Cement production | US Geological Survey | USGS (2016a) |
| | | | USGS (2016b) |
| $E_{LUC}$ | Land cover change (deforestation, afforestation, and forest regrowth) | Forest Resource Assessment (FRA) of the Food and Agriculture Organisation (FAO) | FAO (2010) |
| | Wood harvest | FAO Statistics Division | FAOSTAT (2010) |
| | Shifting agriculture | FAO FRA and Statistics Division | FAO (2010) |
| | | | FAOSTAT (2010) |
| | Interannual variability from peat fires and climate – land management interactions (1997-2013) | Global Fire Emissions Database (GFED4) | Giglio et al., (2013) |
| $G_{ATM}$ | Change in atmospheric $CO_2$ concentration | 1959-1980: $CO_2$ Program at Scripps Institution of Oceanography and other research groups | Keeling et al. (1976) |
| | | 1980-2015: US National Oceanic and Atmospheric Administration Earth System Research Laboratory | Dlugokencky and Tans (2016) |
| | | | Ballantyne et al. (2012) |
| $S_{OCEAN}$ | Uptake of anthropogenic $CO_2$ | 1990-1999 average: indirect estimates based on CFCs, atmospheric $O_2$, and other tracer observations | Manning and Keeling (2006) |
| | | | Keeling et al. (2011) |
| | | | McNeil et al. (2003) |
| | | | Mikaloff Fletcher et al. (2006) as assessed by the IPCC in Denman et al. (2007) |
| | Impact of increasing atmospheric $CO_2$, climate and variability | Ocean models | Table 6 |
| $S_{LAND}$ | Response of land vegetation to: | Budget residual | |
| | Increasing atmospheric $CO_2$ concentration | | |
| | Climate and variability | | |
| | Other environmental changes | | |

**Table 5.** Comparison of the processes included in the bookkeeping method and DGVM models in their estimates of $E_{LUC}$ and $S_{LAND}$. See Table 6 for model references. All models include deforestation and forest regrowth after abandonment of agriculture (or from afforestation activities on agricultural land). Processes relevant for $E_{LUC}$ are only described for the DGVMs used with land-cover change in this study (Fig. 6 top panel).

| | Bookkeeping | CABLE | CLASS-CTEM | CLM | DLEM | ISAM | JSBACH | JULES | LPJ-GUESS | LPJ | LPX-Bern | OCN | ORCHIDEE | SDGVM | VISIT |
|---|---|---|---|---|---|---|---|---|---|---|---|---|---|---|---|
| **Processes relevant for $E_{LUC}$** | | | | | | | | | | | | | | | |
| Wood harvest and forest degradation[a] | yes | | | | | yes | | no | no | no | | yes | | | |
| Shifting cultivation | yes[b] | | | | | no | | no | no | no | | no | | | |
| Cropland harvest | yes | | | | | yes | | no | yes | no | | yes | | | |
| Peat fires | no | | | | | no | | no | no | no | | no | | | |
| **Processes relevant also for $S_{LAND}$** | | | | | | | | | | | | | | | |
| Fire simulation and/or suppression | for US only | no | yes | yes | yes | no | yes | no | yes | yes | yes | no | no | yes | yes |
| Climate and variability | no | yes | yes | yes | yes | yes | yes | yes | yes | yes | yes | yes | yes | yes | yes |
| $CO_2$ fertilisation | no | yes | yes | yes | yes | yes | yes | yes | yes | yes | yes | yes | yes | yes | yes |
| Carbon-nitrogen interactions, including N deposition | no | yes | no | yes | yes | yes | no | no | yes | no | yes | yes | no | yes[c] | no |

[a]Refers to the routine harvest of established managed forests rather than pools of harvested products. [b]No in the recent update (Houghton and Nassikas, in prep.). [c]Very limited. Nitrogen uptake is simulated as a function of soil C, and Vcmax is an empirical function of canopy N. Does not consider N deposition.

1   **Table 6.** References for the process models and data products included in Figs. 6-8. All models and
2   products are updated with new data to end of year 2015.

| Model/data name | Reference | Change from Le Quéré et al. (2015) |
|---|---|---|
| *Dynamic global vegetation models* | | |
| CABLE | Zhang et al. (2013) | Not applicable (not used in 2015) |
| CLASS-CTEM | Melton and Arora (2016) | Not applicable (not used in 2015) |
| CLM | Oleson et al 2013 | No change |
| ISAM | Jain et al. (2013)[a] | No change |
| JSBACH | Reick et al. (2013)[b] | No change |
| JULES[c] | Clarke et al. (2011)[d] | Updated to code release 4.6 and configuration JULES-C-1.1. This version includes improvements to the seasonal cycle of soil respiration. |
| LPJ-GUESS | Smith et al. (2014a) | Use of CRU-NCEP. Crop representation in LPJ-GUESS was adopted from Olin et al. (2015), applying constant fertiliser rate and area fraction under irrigation, as in Elliott et al (2015). |
| LPJ[f] | Sitch et al. (2003)[g] | No change |
| LPX-Bern | Stocker et al. (2014)[h] | Not applicable (not used in 2015) |
| OCN | Zaehle and Friend (2010)[i] | Updated to v1.r278. Biological N fixation is now simulated dynamically according to the OPT scheme of Meyerholt et al. (2016) |
| ORCHIDEE | Krinner et al. (2005)[j] | Updated revision 3687, including a new hydrological scheme with 11 layers and a complete diffusion scheme; a new parameterization of photosynthesis; an improved scheme for representation of snow; a new representation of soil albedo based on satellite data. |
| SDGVM | Woodward et al (1995)[k] | Not applicable (not used in 2015) |
| VISIT | Kato et al. (2013)[l] | Updated to use CRU-NCEP shortwave radiation data instead of using internally estimated radiation from CRU cloudiness data. |
| *Data products for land-use change emissions* | | |
| Bookkeeping | Houghton et al. (2012) | No change |
| Bookkeeping using FAO2015 | Houghton and Nassikas, in prep | Not applicable (not used in 2015) |
| Fire-based emissions | van der Werf et al. (2010) | No change |
| *Ocean biogeochemistry models* | | |
| NEMO-PlankTOM5 | Buitenhuis et al. (2010)[m] | No change |
| NEMO-PISCES (IPSL) | Aumont and Bopp (2006) | No change |
| CCSM-BEC | Doney et al. (2009) | No change |
| MICOM-HAMOCC (NorESM-OC) | Schwinger et al. (2016) | No change |
| NEMO-PISCES (CNRM) | Séférian et al. (2013)[n] | No change |
| CSIRO | Oke et al. (2013) | No change |

| | | |
|---|---|---|
| MITgcm-REcoM2 | Hauck et al. (2016) | nanophytoplankton degradation rate set to 0.1 per day |

*Data products for ocean $CO_2$ flux*

| | | |
|---|---|---|
| Landschützer | Landschützer et al. (2015) | No change |
| Jena CarboScope | Rödenbeck et al. (2014b) | Updated to version oc_1.4 with Longer spin-up/down periods both before and after the data-constrained period. |

*Atmospheric inversions for total $CO_2$ fluxes (land-use-change + land + ocean $CO_2$ fluxes)*

| | | |
|---|---|---|
| CarbonTracker | Peters et al. (2010) | Updated to version CTE2016-FT with minor changes in the inversion set up |
| Jena CarboScope | Rödenbeck et al. (2003) | Updated to version s81_v3.8 |
| CAMS[o] | Chevallier et al. (2005) | Updated to version 15.2 with minor changes in the inversion set up |

[a]See also El-Masri et al. (2013)
[b]See also Goll et al (2015)
[c]Joint UK Land Environment Simulator
[d]See also Best et al. (2011)
[e]Smith et al. (2014)
[f]Lund-Potsdam-Jena
[g]Compared to published version, decreased LPJ wood harvest efficiency so that 50% of biomass was removed off-site
compared to 85% used in the 2012 budget. Residue management of managed grasslands increased so that 100% of
harvested grass enters the litter pool.
[h]Compared to published version: Changed several model parameters, due to new tuning with multiple observational
constraints. No mechanistic changes.
[i]See also Zaehle et al. (2011)
[j]Compared to published version, revised parameters values for photosynthetic capacity for boreal forests (following
assimilation of FLUXNET data), updated parameters values for stem allocation, maintenance respiration and biomass
export for tropical forests (based on literature) and, $CO_2$ down-regulation process added to photosynthesis.
[k]See also Woodward & Lomas (2004). Changed from publications include sub-daily photosynthesis downscaling and
other adjustment.
[l]see also Ito and Inatomi  (2012)
[m]With no nutrient restoring below the mixed layer depth
[n]Uses winds from Atlas et al. (2011)
[o]The CAMS (Copernicus Atmosphere Monitoring Service) v15.2  $CO_2$ inversion system, initially described by Chevallier
et al. (2005), relies on the global tracer transport model LMDZ (see also Supplementary Material Chevallier, 2015;
Hourdin et al., 2006).

**Table 7.** Comparison of results from the bookkeeping method and budget residuals with results from the DGVMs and inverse estimates for the periods 1960-1969, 1970-1979, 1980-1989, 1990-1999, 2000-2009, last decade and last year available. All values are in GtC yr$^{-1}$. The DGVM uncertainties represents ±1σ of the decadal or annual (for 2015 only) estimates from the individual models, for the inverse models all three results are given where available.

| | Mean (GtC yr$^{-1}$) | | | | | | |
|---|---|---|---|---|---|---|---|
| | 1960-1969 | 1970-1979 | 1980-1989 | 1990-1999 | 2000-2009 | 2006-2015 | 2015 |
| *Land-use change emissions ($E_{LUC}$)* | | | | | | | |
| Bookkeeping method | 1.5 ± 0.5 | 1.3 ± 0.5 | 1.4 ± 0.5 | 1.6 ± 0.5 | 1.0 ± 0.5 | 1.0 ± 0.5 | 1.3 ± 0.5 |
| DGVMs[a] | 1.2 ± 0.3 | 1.2 ± 0.3 | 1.2 ± 0.2 | 1.2 ± 0.2 | 1.1 ± 0.2 | 1.3 ± 0.3 | 1.2 ± 0.4 |
| *Residual terrestrial sink ($S_{LAND}$)* | | | | | | | |
| Budget residual | 1.7 ± 0.7 | 1.7 ± 0.8 | 1.6 ± 0.8 | 2.6 ± 0.8 | 2.6 ± 0.8 | 3.2 ± 0.8 | 2.0 ± 0.9 |
| DGVMs[a] | 1.2 ± 0.5 | 2.2 ± 0.5 | 1.7 ± 0.6 | 2.3 ± 0.5 | 2.8 ± 0.7 | 2.8 ± 0.7 | 1.0 ± 1.4 |
| *Total land fluxes ($S_{LAND} - E_{LUC}$)* | | | | | | | |
| Budget ($E_{FF}$-$G_{ATM}$-$S_{OCEAN}$) | 0.2 ± 0.5 | 0.4 ± 0.6 | 0.1 ± 0.6 | 1.0 ± 0.6 | 1.4 ± 0.6 | 2.1 ± 0.7 | 0.6 ± 0.7 |
| DGVMs[a] | -0.2 ± 0.7 | 1.1 ± 0.5 | 0.4 ± 0.5 | 1.1 ± 0.3 | 1.8 ± 0.4 | 1.7 ± 0.5 | -0.1 ± 1.4 |
| Inversions (CTE2016-FT/Jena CarboScope/CAMS)* | —/—/— | —/—/— | —/0.2*/0.9* | —/1.0*/1.9* | 1.5/1.6*/2.5* | 2.2*/2.3*/3.4* | 1.9*/2.6*/2.6* |

[a]Note that for DGVMs, the mean reported for the total land fluxes is not equal to the difference between the means reported for $S_{LAND}$ and $E_{LUC}$ as different set of models contributed to these two estimates (see section 2.2.3).

*Estimates are not corrected for the influence of river fluxes, which would reduce the fluxes by 0.45 GtC yr$^{-1}$ when neglecting the anthropogenic influence on land (Section 2.7.2). See Table 6 for model references.

**Table 8.** Decadal mean in the five components of the anthropogenic $CO_2$ budget for the periods 1960-1969, 1970-1979, 1980-1989, 1990-1999,
2000-2009, last decade and last year available. All values are in GtC yr$^{-1}$. All uncertainties are reported as ±1σ. A data set containing data for
each year during 1959-2014 is available on http://cdiac.ornl.gov/GCP/carbonbudget/2015/ . Please follow the terms of use and cite the
original data sources as specified on the data set.

| | Mean (GtC yr$^{-1}$) | | | | | | |
|---|---|---|---|---|---|---|---|
| | 1960-1969 | 1970-1979 | 1980-1989 | 1990-1999 | 2000-2009 | 2006-2015 | 2015 |
| *Emissions* | | | | | | | |
| Fossil fuels and industry ($E_{FF}$) | 3.1 ± 0.2 | 4.7 ± 0.2 | 5.5 ± 0.3 | 6.3 ± 0.3 | 8.0 ± 0.4 | 9.3 ± 0.5 | 9.9 ± 0.5 |
| Land-use change emissions ($E_{LUC}$) | 1.5 ± 0.5 | 1.3 ± 0.5 | 1.4 ± 0.5 | 1.6 ± 0.5 | 1.0 ± 0.5 | 1.0 ± 0.5 | 1.3 ± 0.5 |
| *Partitioning* | | | | | | | |
| Growth rate in atmospheric $CO_2$ concentration ($G_{ATM}$) | 1.7 ± 0.1 | 2.8 ± 0.1 | 3.4 ± 0.1 | 3.1 ± 0.1 | 4.0 ± 0.1 | 4.5 ± 0.1 | 6.2 ± 0.2 |
| Ocean sink ($S_{OCEAN}$) | 1.2 ± 0.5 | 1.5 ± 0.5 | 1.9 ± 0.5 | 2.2 ± 0.5 | 2.3 ± 0.5 | 2.6 ± 0.5 | 3.0 ± 0.5 |
| Residual terrestrial sink ($S_{LAND}$) | 1.7 ± 0.7 | 1.7 ± 0.8 | 1.6± 0.8 | 2.6 ± 0.8 | 2.6 ± 0.8 | 3.2 ± 0.8 | 2.0 ± 0.9 |

**Table 9.** Actual $CO_2$ emissions from fossil fuels and industry ($E_{FF}$) compared to projections made
the previous year based on world GDP (IMF October 2015) and the fossil fuel intensity of GDP ($I_{FF}$)
based on subtracting the $CO_2$ and GDP growth rates. The 'Actual' values are the latest estimate
available and the 'Projected' value for 2015 refers to those presented in this paper. A correction
for leap years is applied (Section 2.1.3).

| | $E_{FF}$ | | GDP | | $I_{FF}$ | |
|---|---|---|---|---|---|---|
| | Projected | Actual | Projected | Actual | Projected | Actual |
| 2009[a] | −2.8% | −1.1% | −1.1% | -0.05% | −1.7% | −1.1% |
| 2010[b] | >3% | 5.7% | 4.8% | 5.4% | >−1.7% | +0.3% |
| 2011[c] | 3.1±1.5% | 4.1% | 4.0% | 4.2% | −0.9±1.5% | −0.2% |
| 2012[d] | 2.6%[i] (1.9 to 3.5) | 1.7% | 3.3% | 3.5% | −0.7% | −1.8% |
| 2013[e] | 2.1% (1.1 to 3.1) | 1.1% | 2.9% | 3.3% | −0.8% | −2.2% |
| 2014[f] | 2.5% (1.3 to 3.5) | 0.8% | 3.3% | 3.0% | −0.7% | −2.6% |
| Change in method | | | | | | |
| | $E_{FF}$ | | $E_{FF}$ (China) | | $E_{FF}$ (USA) | | $E_{FF}$ (Rest of World) | |
| | Projected | Actual | Projected | Actual | Projected | Actual | Projected | Actual |
| 2015[g] | −0.6% (−1.6 to 0.5) | 0.05% | −3.9% (−4.6 to -1.1) | −0.8% | −1.5% (−5.5 to 0.3) | −2.6% | 1.2% (−0.2 to 2.6) | 1.2% |
| 2016[h] | +0.2%[i] (−1.0 to +1.7) | -- | −0.6%[i] (−3.4 to +1.1) | -- | −1.5%[i] (−4.0 to +1.0) | -- | +1.1%[i] (−0.3 to +2.6) | -- |

[a]Le Quéré et al. (2009). [b]Friedlingstein et al. (2010). [c]Peters et al. (2013). [d]Le Quéré et al. (2013). [e]Le Quéré et al.
(2014). [f]Friedlingstein et al. (2014) and Le Quéré et al. (2015b). [g]Jackson et al. (2016) and Le Quéré et al. (2015a). [h]This
study. [i]These numbers are not adjusted for leap years.
**Table 10.** Cumulative $CO_2$ emissions for the periods 1750-2014, 1870-2014 and 1870-2015 in
gigatonnes of carbon (GtC). We also provide the 1850-2005 time-period used in a number of
model evaluation publications. All uncertainties are reported as ±1σ. All values are rounded to
nearest 5 GtC as in Stocker et al. (2013), reflecting the limits of our capacity to constrain
cumulative estimates. Thus some columns will not exactly balance because of rounding errors.

| Units of GtC | 1750-2015 | 1850-2005 | 1870-2015 | 1870-2016 |
|---|---|---|---|---|
| *Emissions* | | | | |
| Fossil fuels and industry ($E_{FF}$) | 415 ± 20 | 320 ± 15 | 410 ± 20 | 420 ± 20* |
| Land-use change emissions ($E_{LUC}$) | 190 ± 65 | 150 ± 55 | 145 ± 50 | 150 ± 50* |
| Total emissions | 605 ± 70 | 470 ± 55 | 560 ± 55 | 570 ± 55* |
| *Partitioning* | | | | |
| Growth rate in atmospheric $CO_2$ concentration ($G_{ATM}$) | 260 ± 5 | 195 ± 5 | 235 ± 5 | |
| Ocean sink ($S_{OCEAN}$) | 175 ± 20 | 160 ± 20 | 155 ± 20 | |
| Residual terrestrial sink ($S_{LAND}$) | 170 ± 70 | 115 ± 60 | 165 ± 60 | |

*The extension to year 2016 uses the emissions projections for fossil fuels and industry for 2016 (Sect. 3.2) and
assumes a constant $E_{LUC}$ flux (Sect. 2.2).

2  **Table 11.** Funding supporting the production of the various components of the global carbon
3  budget (see also acknowledgements).

| Funder and grant number (where relevant) | author initials |
|---|---|
| Biological and Environmental Research Program, Office of Science, U. S. Department of Energy (contract no.  DE-AC05-00OR22725) | APW |
| BMBF, Germany | AK |
| BNP Paribas Climate Philanthropy Grant for the Global Carbon Atlas, France | PC |
| Copernicus Atmosphere Monitoring Service, European Centre for Medium-Range Weather Forecasts (ECMWF), European Commission | FC |
| CSIR | PMS |
| Department of Energy, US (grant no. DE-FC03-97ER62402/A010) | DL |
| FWO Flanders (formerly Hercules foundation) | TG |
| German Federal Ministry of Education and Research (grant no. 01LK1224I ICOS-D) | MH |
| German Research Foundation's Emmy Noether Program (grant no. PO1751/1-1) | JN |
| H2020 (CRESCENDO; grant no. 641816) | CD, RS, OA, PF |
| H2020 European Research Council (ERC) (QUINCY; grant no. 647204). | SZ |
| H2020 European Research Council Synergy grant (IMBALANCE-P; grant no. ERC-2013-SyG-610028) | PC |
| Helmholtz PostDoc Programme (Initiative and Networking Fund of the Helmholtz Association), Germany | JH |
| Institut National des Sciences de l'Univers (INSU) and Institut Paul Emile Victor (IPEV) for OISO cruises, France | NM |
| IRD/ EU Atlantos | NL |
| MAFF | OT |
| Ministry of Environment of Japan | SN |
| Ministry of Environment of Japan | SN |
| Ministry of Environment of Japan (grant no. ERTDF S-10) | EK |
| National Institute of Food and Agriculture/US Department of Agriculture (grant no. 2015-67003-23485) | DL |
| Natural Environment Research Council UK (RAGNARoCC) | US |
| Newton Fund through the Met Office Climate Science for Service Partnership Brazil (CSSP Brazil) | AJW |
| NIWA (National Institute of Water and Atmospheric Research) Core Funding, New Zealand | KC |
| NOAA's Climate Observation Division of the Climate Program Office (grant no. N8R1SE3P00); NOAA's Ocean Acidification Program (grant no. N8R3CEAP00), US | DP, LB |
| NOAA's Climate Observation Division of the Climate Program Office, NOAA, US Department of Commerce | SRA, AJS |
| Norwegian Environment Agency (grant no. 16078007) | IS |
| NSF (National Science Foundation; grant no. AGS-1048827), USA | SD |
| RCN | OMA |
| Research Council of Norway (grant no. 569980) | GPP, RMA |
| Research Council of Norway (project EVA; grant no. 229771) | JS |
| | |
| **Computing time** | |
| GENCI (Grand Équipement National de Calcul Intensif; allocation t2016012201), France | FC |
| Météo-France/DSI supercomputing centre | RS |

| | |
|---|---|
| Netherlands Organization for Scientific Research (NWO) (SH-312-14) | IvdL-L |
| Norwegian Metacenter for Computational Science (NOTUR, project nn2980k) and the Norwegian Storage Infrastructure (NorStore, project ns2980k) | JS |
| UEA High Performance Computing Cluster, UK | OA, CLQ |

**Fig. 1**

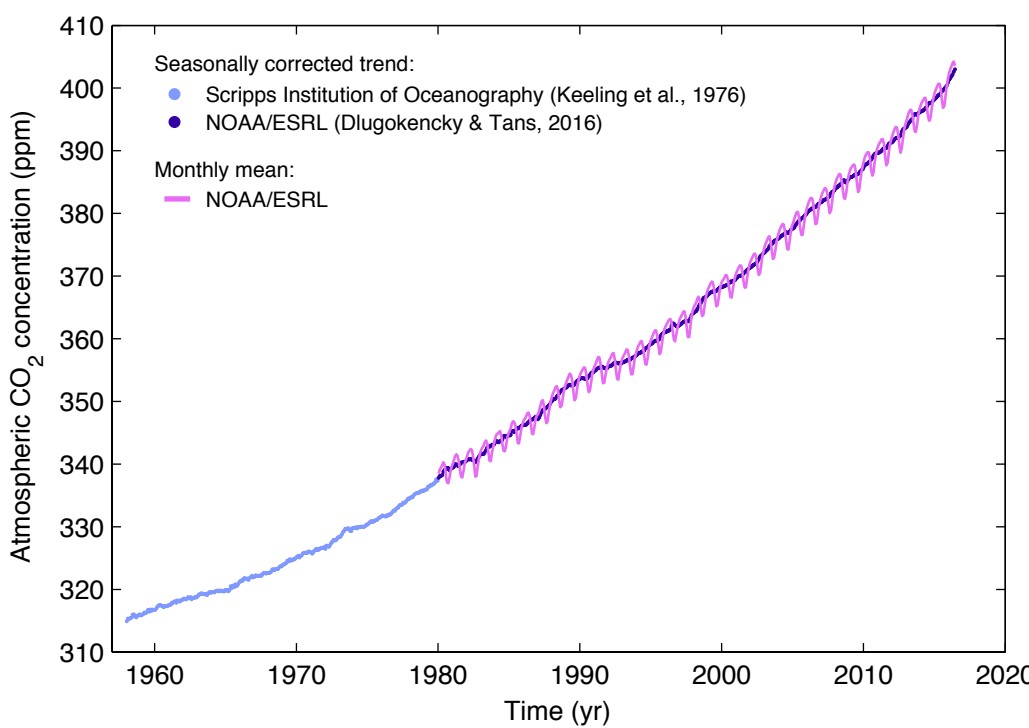

**Figure 1.** Surface average atmospheric $CO_2$ concentration, deseasonalised (ppm). The 1980-2016
monthly data are from NOAA/ESRL (Dlugokencky and Tans, 2016)  and are based on an average of
direct atmospheric $CO_2$ measurements from multiple stations in the marine boundary layer
(Masarie and Tans, 1995). The 1958-1979 monthly data are from the Scripps Institution of
Oceanography, based on an average of direct atmospheric $CO_2$ measurements from the Mauna
Loa and South Pole stations (Keeling et al., 1976). To take into account the difference of mean $CO_2$
between the NOAA/ESRL and the Scripps station networks used here, the Scripps surface average
(from two stations) was harmonised to match the NOAA/ESRL surface average (from multiple
stations) by adding the mean difference of 0.542 ppm, calculated here from overlapping data
during 1980-2012. The mean seasonal cycle is also shown from 1980 (in pink).
**Fig. 2**

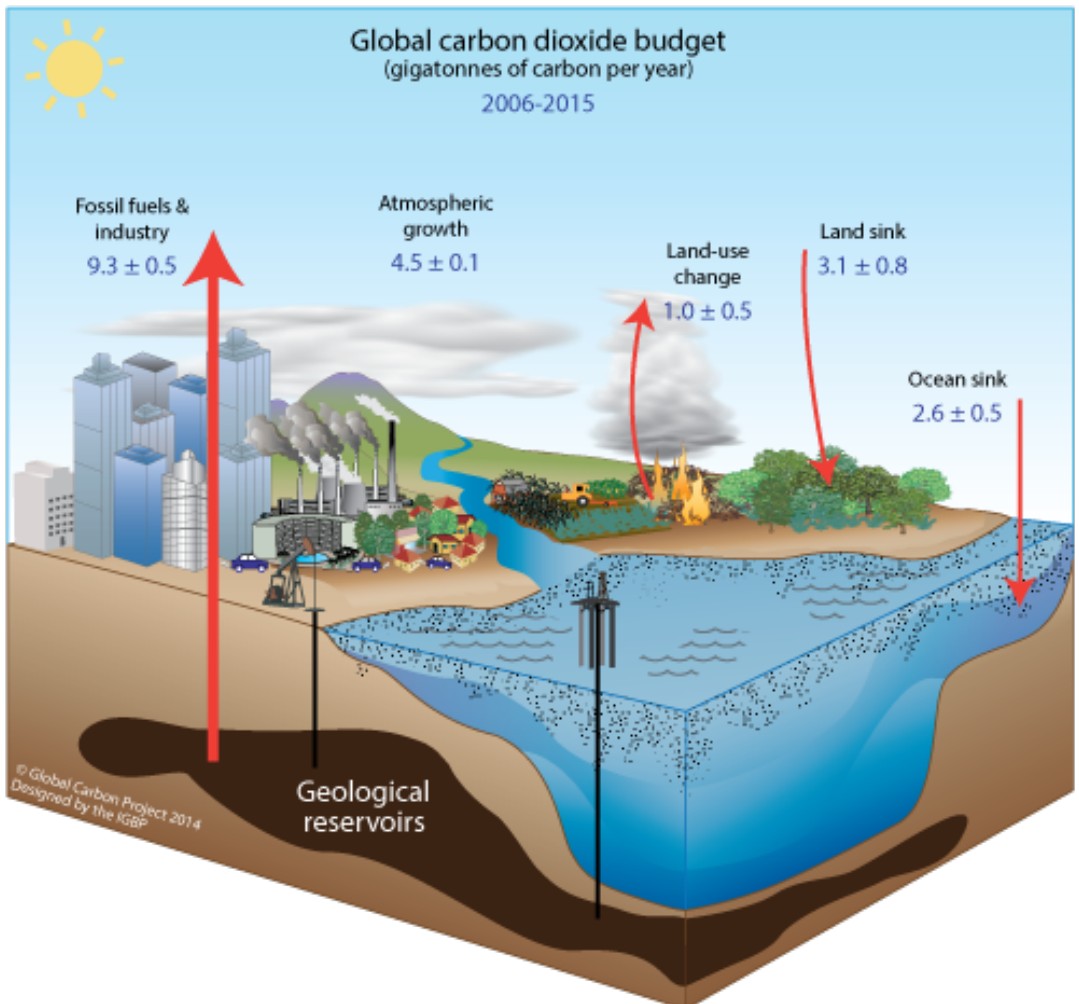

**Figure 2.**Schematic representation of the overall perturbation of the global carbon cycle caused by
anthropogenic activities, averaged globally for the decade 2006-2015. The arrows represent
emission from fossil fuels and industry ($E_{FF}$); emissions from deforestation and other land-use
change ($E_{LUC}$); the growth rate in atmospheric $CO_2$ concentration ($G_{ATM}$) and the uptake of carbon
by the 'sinks' in the ocean ($S_{OCEAN}$) and land ($S_{LAND}$) reservoirs. All fluxes are in units of GtC yr$^{-1}$,
with uncertainties reported as ±1σ (68% confidence that the real value lies within the given
interval) as described in the text. This figure is an update of one prepared by the International
Geosphere Biosphere Programme for the GCP, first presented in Le Quéré (2009).
**Fig. 3**

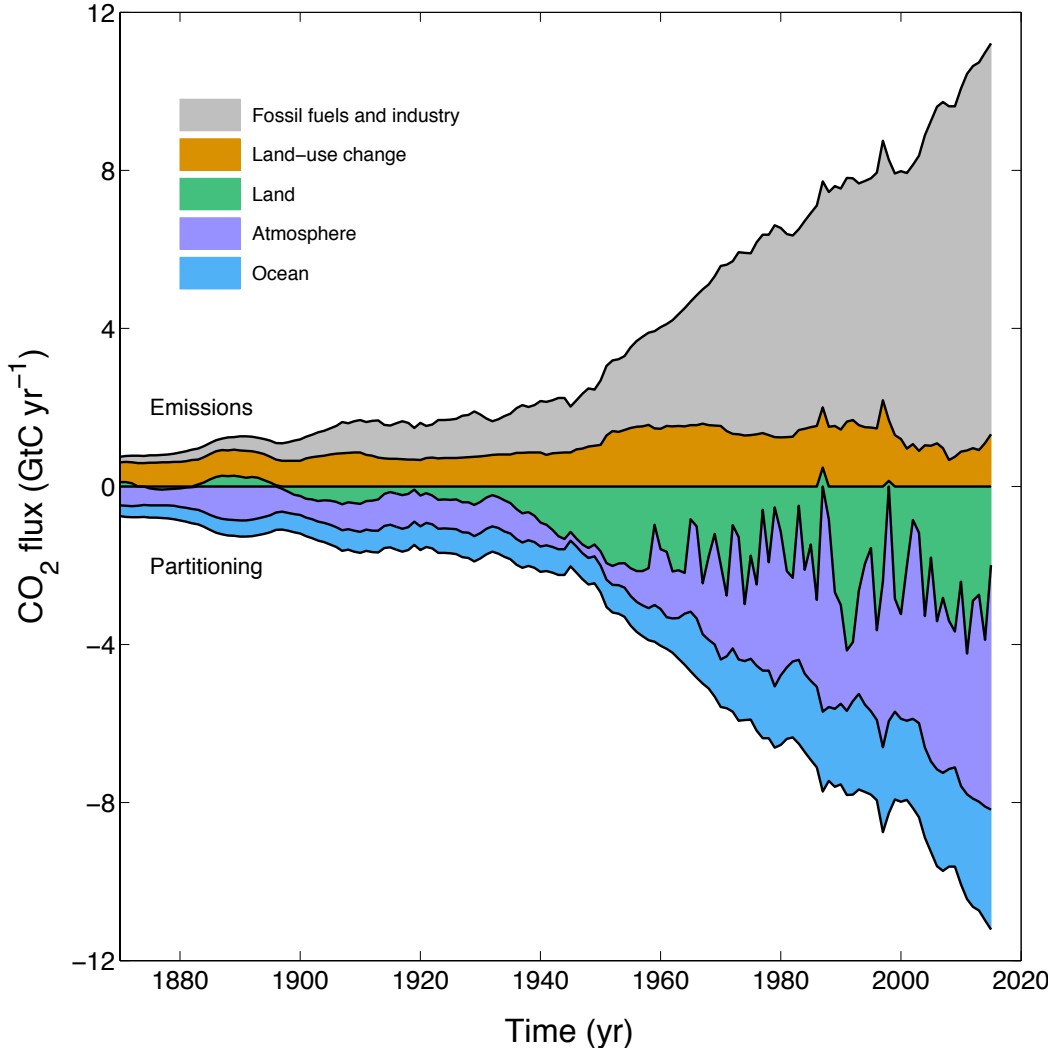

**Figure 3.** Combined components of the global carbon budget illustrated in Fig. 2 as a function of
time, for emissions from fossil fuels and industry ($E_{FF}$; grey) and emissions from land-use change
($E_{LUC}$; brown), as well as their partitioning among the atmosphere ($G_{ATM}$; purple), land ($S_{LAND}$;
green) and oceans ($S_{OCEAN}$; dark blue). All time series are in GtC yr$^{-1}$. $G_{ATM}$ and $S_{OCEAN}$ (and by
construction also $S_{LAND}$) prior to 1959 are based on different methods. The primary data sources
for fossil fuels and industry are from Boden and Andres (2016), with uncertainty of about ±5%
(±1σ); land-use change emissions are from Houghton et al. (2012) with uncertainties of about
±30%; growth rate in atmospheric $CO_2$ concentration prior to 1959 is from Joos and Spahni (2008)
with uncertainties of about ±1-1.5 GtC decade$^{-1}$ or ±0.1-0.15 GtC yr$^{-1}$ (Bruno and Joos, 1997), and
from Dlugokencky and Tans (2016) from 1959 with uncertainties of about ±0.2 GtC yr$^{-1}$; the ocean
sink prior to 1959 is from Khatiwala et al. (2013) with uncertainty of about ±30%, and from this
study from 1959 with uncertainties of about ±0.5 GtC yr$^{-1}$; and the residual land sink is obtained
by difference (Eq. 8), resulting in uncertainties of about ±50% prior to 1959 and ±0.8 GtC yr$^{-1}$ after
that. See the text for more details of each component and their uncertainties.
**Fig. 4**

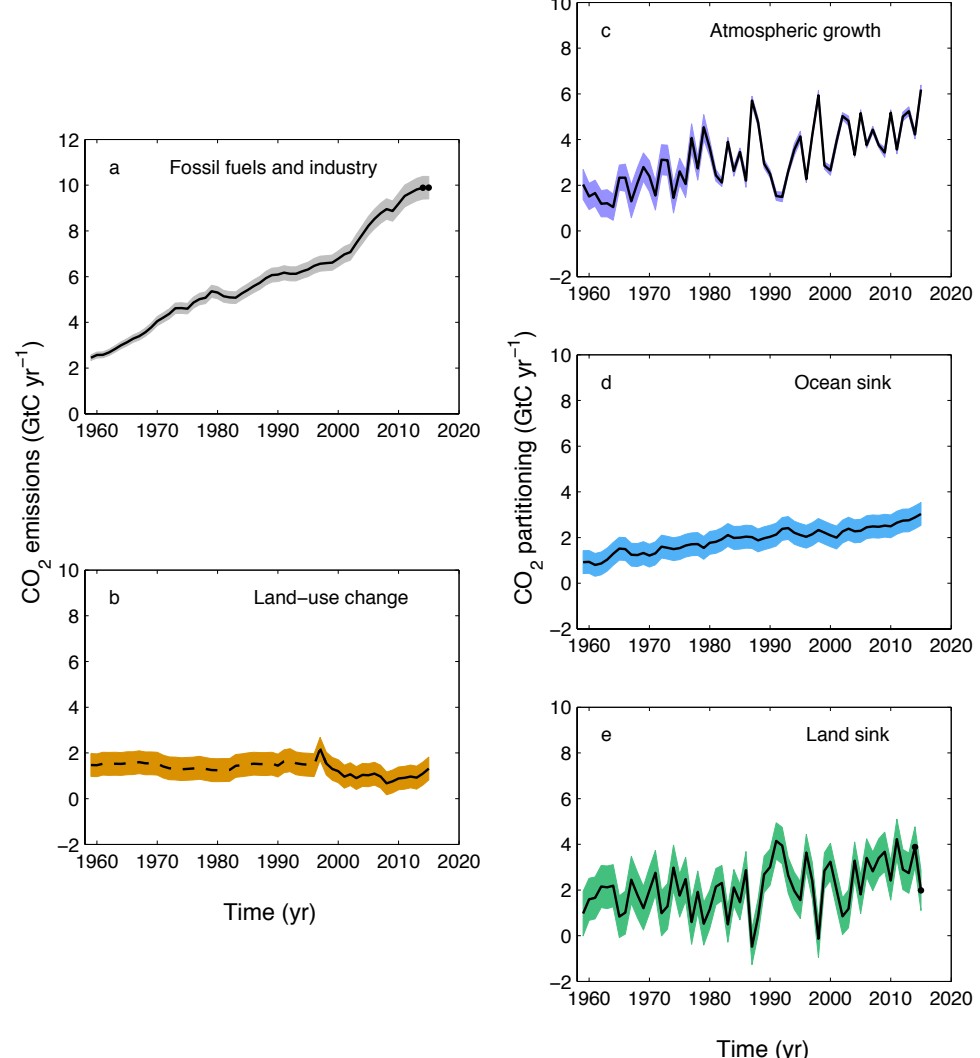

**Figure 4.** Components of the global carbon budget and their uncertainties as a function of time,
presented individually for (a) emissions from fossil fuels and industry ($E_{FF}$), (b) emissions from
land-use change ($E_{LUC}$), (c) growth rate in atmospheric $CO_2$ concentration ($G_{ATM}$), (d) the ocean $CO_2$
sink ($S_{OCEAN}$, positive indicates a flux from the atmosphere to the ocean), and (e) the land $CO_2$ sink
($S_{LAND}$, positive indicates a flux from the atmosphere to the land). All time series are in GtC yr$^{-1}$
with the uncertainty bounds representing ±1σ in shaded colour. Data sources are as in Fig. 3. The
black dots in panels (a) and (e) show values for 2014 and 2015 that originate from a different data
set to the remainder of the data, while the dashed line in panel (b) highlights the start of satellite
data use to estimate the interannual variability and extend the series in time (see text).
**Fig. 5**

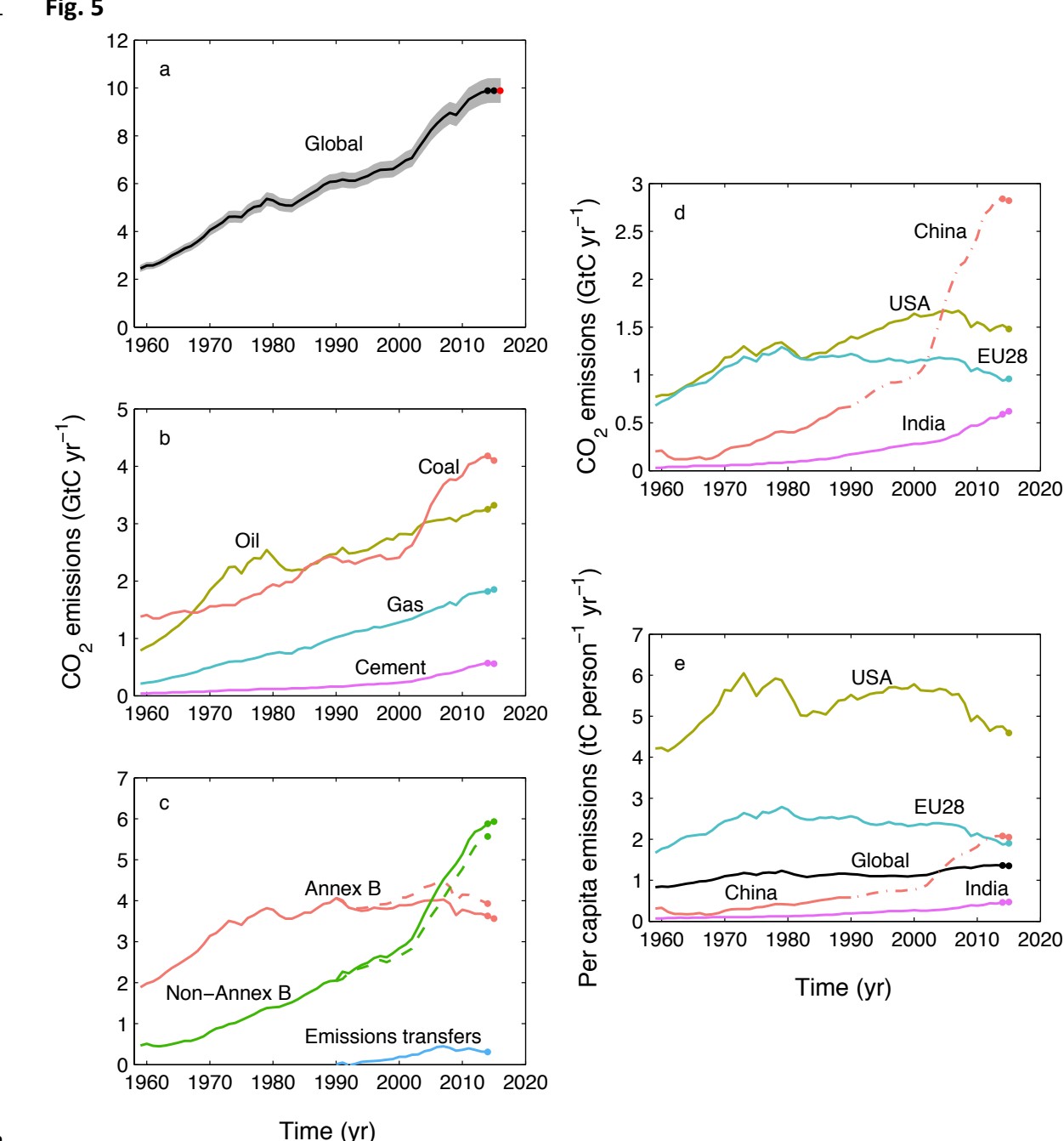

**Figure 5.** CO$_2$ emissions from fossil fuels and industry for (**a**) the globe, including an uncertainty of
± 5% (grey shading), the emissions extrapolated using BP energy statistics (black dots) and the
emissions projection for year 2016 based on GDP projection (red dot), (**b**) global emissions by fuel
type, including coal (salmon), oil (olive), gas (turquoise), and cement (purple), and excluding gas
flaring which is small (0.6% in 2013), (**c**) territorial (solid line) and consumption (dashed line)
emissions for the countries listed in Annex B of the Kyoto Protocol (salmon lines; mostly advanced
economies with emissions limitations) versus non-Annex B countries (green lines); also shown are
the emissions transfer from non-Annex B to Annex B countries (light blue line) (**d**) territorial CO$_2$
emissions for the top three country emitters (USA - olive; China - salmon; India - purple) and for
the European Union (EU; turquoise for the 28 member states of the EU as of 2012), and (**e**) per-
capita emissions for the top three country emitters and the EU (all colours as in panel (**d**)) and the
world (black). In panels (**b**) to (**e**), the dots show the data that were extrapolated from BP energy
statistics for 2014 and 2015. All time series are in GtC yr$^{-1}$ except the per-capita emissions (panel
(**e**)), which are in tonnes of carbon per person per year (tC person$^{-1}$ yr$^{-1}$). Territorial emissions are
primarily from Boden and Andres (2016) except national data for the USA and EU28 for 1990-
2014, which are reported by the countries to the UNFCCC as detailed in the text, and for China
from 1990 which are estimated here from BP energy statistics (the latter shown as a dash-dot
line); consumption-based emissions are updated from Peters et al. (2011a). See Section 2.1.1 for
details of the calculations and data sources.
**Fig. 6**

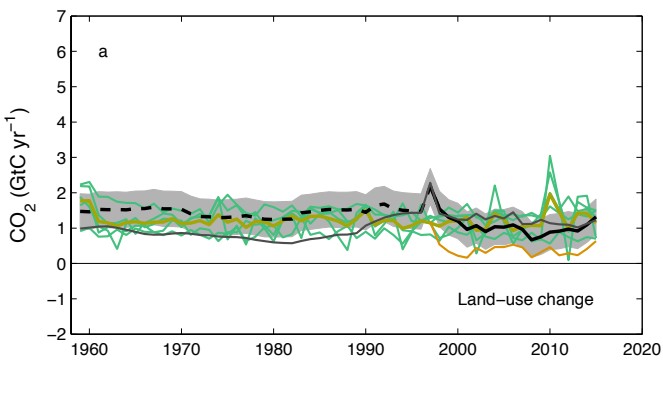

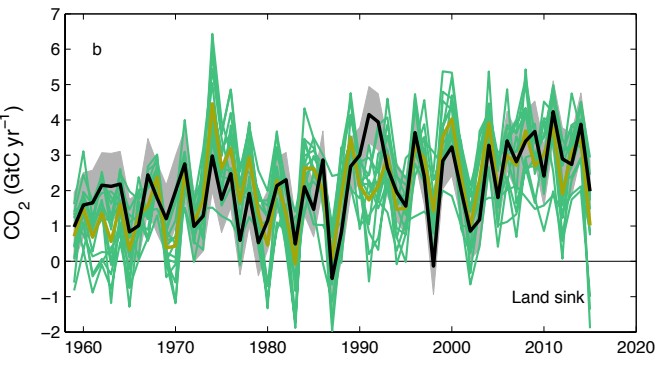

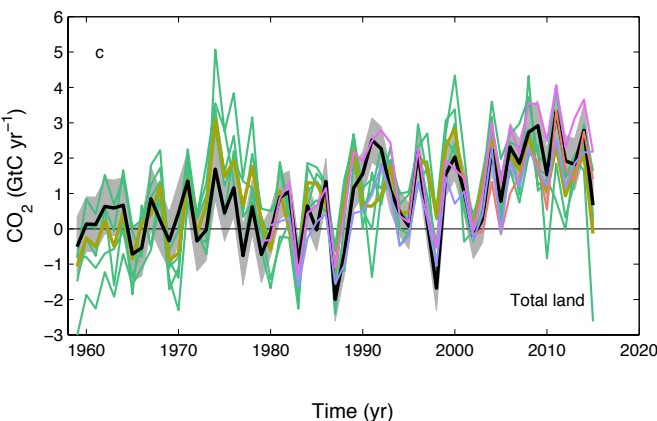

**Figure 6.** Atmosphere-land $CO_2$ flux. (a) Comparison of the global carbon budget values of $E_{LUC}$
(black with ±1σ uncertainty in grey shading), with $CO_2$ emissions from land-use change showing
individual DGVM model results (green) and the multi model mean (olive), and fire-based results
(orange); land-use change data prior to 1997 (dashed black) highlights the pre-satellite years;
preliminary results using the FAO FRA 2015 (Houghton and Nassikas, in preparation) are also
shown in dark grey. (**b**) Land $CO_2$ sink ($S_{LAND}$; black with uncertainty in grey shading) showing
individual DGVM model results (green) and multi model mean (olive). (**c**) Total land $CO_2$ fluxes (b –
a; black with uncertainty in grey shading), from DGVM model results (green) and the multi model
mean (olive), atmospheric inversions Chevallier et al. (2005; CAMSv15.2) in purple; Rödenbeck et
al. (2003; Jena CarboScope, s81_v3.8) in violet; Peters et al. (2010; Carbon Tracker, CTE2016-FT) in
salmon; see Table 6, and the carbon balance from Eq. (1) (black). In (**c**) the inversions were
corrected for the pre-industrial land sink of $CO_2$ from river input, by removing a sink of 0.45 GtC yr$^{-1}$
(Jacobson et al., 2007). This correction does not take into account the anthropogenic
contribution to river fluxes (see Sect. 2.7.2).
**Fig. 7**

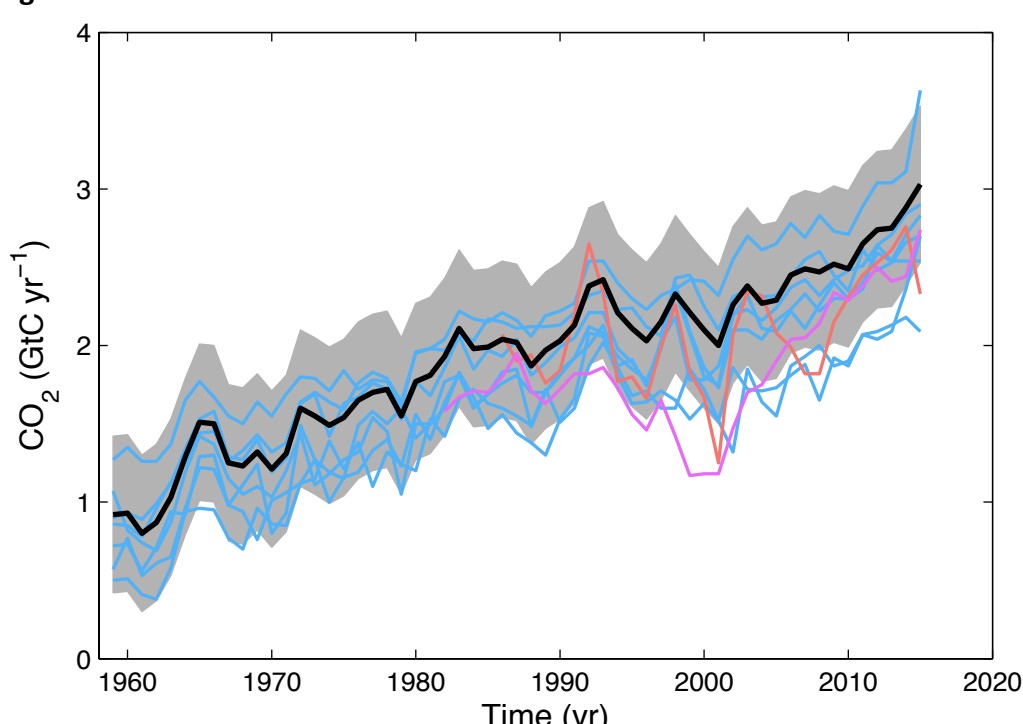

**Figure 7.** Comparison of the anthropogenic atmosphere-ocean $CO_2$ flux showing the budget values
of $S_{OCEAN}$ (black; with ±1σ uncertainty in grey shading), individual ocean models before
normalisation (blue), and the two ocean data-based products (Rödenbeck et al. (2014b) in salmon
and Landschützer et al. (2015) in purple; see Table 6). Both data-based products were adjusted for
the pre-industrial ocean source of $CO_2$ from river input to the ocean, which is not present in the
models, by adding a sink of 0.45 GtC yr$^{-1}$ (Jacobson et al., 2007), to make them comparable to
$S_{OCEAN}$. This adjustment does not take into account the anthropogenic contribution to river fluxes
(see Section 2.7.2).
**Fig. 8**

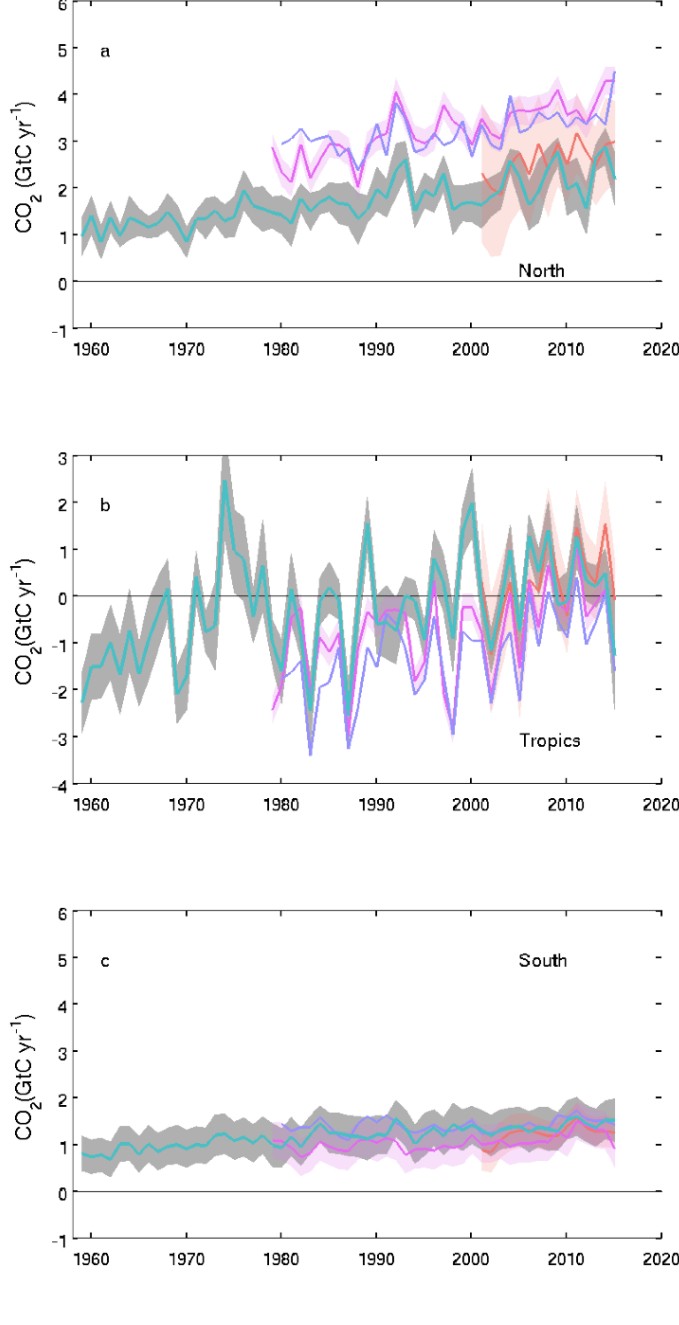

**Figure 8.** $CO_2$ fluxes between the atmosphere and the surface ($S_{OCEAN}$ + $S_{LAND}$ + $E_{LUC}$) by latitude
bands for the (**a**) North (north of 30°N), (**b**) Tropics (30°S-30°N), and (**c**) South (south of 30°S).
Estimates from the combination of the multi-model means for the land and oceans are shown
(turquoise) with ±1σ of the model ensemble (in grey). Results from the three atmospheric
inversions are shown from Chevallier et al. (2005; CAMSv15.2) in purple; Rödenbeck et al. (2003;
Jena CarboScope, s81_v3.8) in blue; Peters et al. (2010; CarbonTracker, CTE2016-FT) in salmon;
see Table 6. Where available the uncertainty in the inversions are also shown.
**Fig. 9**

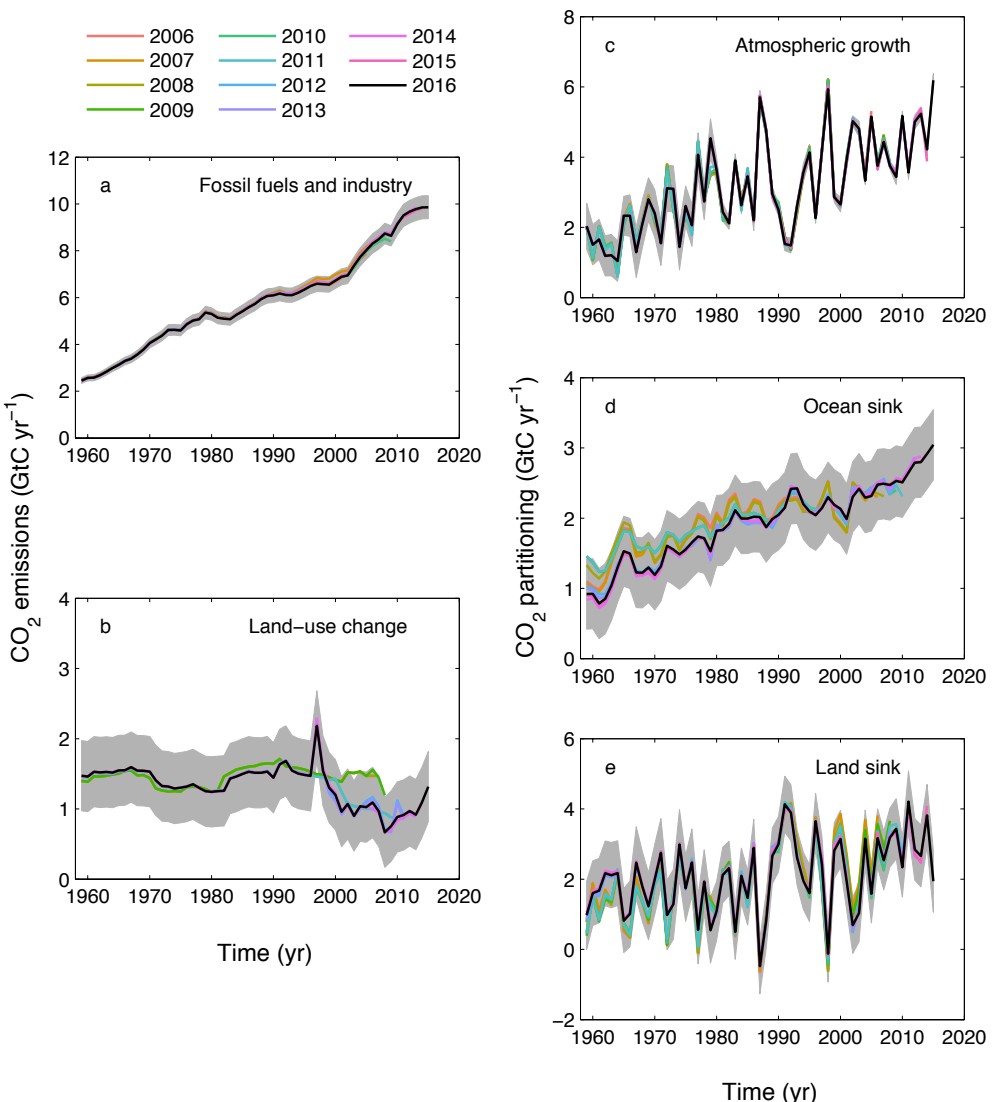

**Figure 9.** Comparison of global carbon budget components released annually by GCP since 2006.
$CO_2$ emissions from both (**a**) fossil fuels and industry ($E_{FF}$), and (**b**) land-use change ($E_{LUC}$), and their
partitioning among (**c**) the atmosphere ($G_{ATM}$), (**d**) the ocean ($S_{OCEAN}$), and (**e**) the land ($S_{LAND}$). See
legend for the corresponding years, with the 2006 carbon budget from Raupach et al.(2007); 2007
from Canadell et al. (2007); 2008 released online only; 2009 from Le Quéré et al.; 2010 from
Friedlingstein et al. (2010); 2011 from Peters et al. (2012b); 2012 from Le Quéré et al. (2013); 2013
from Le Quéré et al. (2014), 2014 from Le Quéré et al. (2015b), 2015 from Le Quéré et al. (2015a),
and this year's budget (2016; this study). The budget year generally corresponds to the year when
the budget was first released. All values are in GtC yr$^{-1}$. Grey shading shows the uncertainty
bounds representing ±1σ.

1 **Appendix** Attribution of $fCO_2$ measurements for year 2015 included in addition to SOCAT v4 (Bakker et al., 2016; Bakker, 2014) to inform ocean

2 data products.

| Vessel | Start date yyy-mm-dd | End date yyy-mm-dd | Regions | No. of samples | Principal investigators | DOI (if available)/comment |
|---|---|---|---|---|---|---|
| Atlantic Companion | 2015-03-03 | 2015-03-10 | North Atlantic | 8496 | Steinhoff, T.; Becker, M.; Körtzinger, A. | 10.3334/CDIAC/OTG.VOS_Atlantic_Companion_Line_2015 |
| Atlantic Companion | 2015-03-30 | 2015-04-07 | North Atlantic | 9265 | Steinhoff, T.; Becker, M.; Körtzinger, A. | 10.3334/CDIAC/OTG.VOS_Atlantic_Companion_Line_2015 |
| Aurora Australis | 2014-12-05 | 2015-01-24 | Southern Ocean | 41463 | Tilbrook, B. | 10.3334/CDIAC/OTG.VOS_AA_2014 |
| Benguela Stream | 2015-01-08 | 2015-01-14 | North Atlantic, Tropical Atlantic | 4664 | Schuster, U.; Jones, S.D.; Watson, A.J. | 10.3334/CDIAC/OTG.VOS_BENGUELA_STREAM_2015 |
| Benguela Stream | 2015-02-05 | 2015-02-12 | North Atlantic, Tropical Atlantic | 4056 | Schuster, U.; Jones, S.D.; Watson, A.J. | 10.3334/CDIAC/OTG.VOS_BENGUELA_STREAM_2015 |
| Benguela Stream | 2015-02-22 | 2015-03-01 | North Atlantic, Tropical Atlantic | 6158 | Schuster, U.; Jones, S.D.; Watson, A.J. | 10.3334/CDIAC/OTG.VOS_BENGUELA_STREAM_2015 |
| Benguela Stream | 2015-04-30 | 2015-05-07 | North Atlantic, Tropical Atlantic | 6125 | Schuster, U.; Jones, S.D.; Watson, A.J. | 10.3334/CDIAC/OTG.VOS_BENGUELA_STREAM_2015 |
| Benguela Stream | 2015-05-17 | 2015-05-24 | North Atlantic, Tropical Atlantic | 6152 | Schuster, U.; Jones, S.D.; Watson, A.J. | 10.3334/CDIAC/OTG.VOS_BENGUELA_STREAM_2015 |
| Benguela Stream | 2015-05-27 | 2015-06-04 | North Atlantic, Tropical Atlantic | 6116 | Schuster, U.; Jones, S.D.; Watson, A.J. | 10.3334/CDIAC/OTG.VOS_BENGUELA_STREAM_2015 |
| Benguela Stream | 2015-06-24 | 2015-07-02 | North Atlantic, Tropical Atlantic | 6538 | Schuster, U.; Jones, S.D.; Watson, A.J. | 10.3334/CDIAC/OTG.VOS_BENGUELA_STREAM_2015 |
| Benguela Stream | 2015-07-11 | 2015-07-19 | North Atlantic, Tropical Atlantic | 6220 | Schuster, U.; Jones, S.D.; Watson, A.J. | 10.3334/CDIAC/OTG.VOS_BENGUELA_STREAM_2015 |
| Benguela Stream | 2015-07-22 | 2015-07-30 | North Atlantic, Tropical Atlantic | 6534 | Schuster, U.; Jones, S.D.; Watson, A.J. | 10.3334/CDIAC/OTG.VOS_BENGUELA_STREAM_2015 |
| Benguela Stream | 2015-08-08 | 2015-08-16 | North Atlantic, Tropical Atlantic | 6727 | Schuster, U.; Jones, S.D.; Watson, A.J. | 10.3334/CDIAC/OTG.VOS_BENGUELA_STREAM_2015 |
| Benguela Stream | 2015-08-19 | 2015-08-27 | North Atlantic, Tropical Atlantic | 6811 | Schuster, U.; Jones, S.D.; Watson, A.J. | 10.3334/CDIAC/OTG.VOS_BENGUELA_STREAM_2015 |
| Cap Blanche | 2015-03-28 | 2015-04-10 | Tropical Pacific, Southern Ocean | 6117 | Cosca C.; Feely R.; Alin S. | 10.3334/CDIAC/OTG.VOS_CAP_BLANCHE_2015 |
| Cap Blanche | 2015-09-30 | 2015-10-12 | Tropical Pacific, Southern Ocean | 5582 | Cosca C.; Feely R.; Alin S. | 10.3334/CDIAC/OTG.VOS_CAP_BLANCHE_2015 |
| Cap Blanche | 2015-11-20 | 2015-12-04 | Tropical Pacific, Southern Ocean | 6677 | Cosca C.; Feely R.; Alin S. | 10.3334/CDIAC/OTG.VOS_CAP_BLANCHE_2015 |
| Cap San Lorenzo | 2015-02-28 | 2015-03-12 | North Atlantic, Tropical Atlantic | 5699 | Lefèvre, N., Diverrès D. | |
| Cap San Lorenzo | 2015-03-31 | 2015-04-06 | Tropical Atlantic | 2654 | Lefèvre, N., Diverrès D. | |
| Cap San Lorenzo | 2015-04-28 | 2015-05-07 | North Atlantic, Tropical Atlantic | 4335 | Lefèvre, N., Diverrès D. | |
| Cap San Lorenzo | 2015-06-20 | 2015-07-01 | North Atlantic, Tropical Atlantic | 5833 | Lefèvre, N., Diverrès D. | |
| Cap San Lorenzo | 2015-07-29 | 2015-08-04 | North Atlantic | 2934 | Lefèvre, N., Diverrès D. | |
| Colibri | 2015-02-26 | 2015-03-10 | North Atlantic, Tropical Atlantic | 4615 | Lefèvre, N., Diverrès D. | |
| Colibri | 2015-03-12 | 2015-03-23 | North Atlantic, Tropical Atlantic | 5561 | Lefèvre, N., Diverrès D. | |
| Colibri | 2015-05-26 | 2015-06-04 | North Atlantic, Tropical Atlantic | 3683 | Lefèvre, N., Diverrès D. | |

| Colibri | 2015-06-07 | 2015-06-18 | North Atlantic, Tropical Atlantic | 5613 | Lefèvre, N., Diverrès D. | |
| Equinox | 2015-02-24 | 2015-03-06 | Tropical Atlantic | 3563 | Wanninkhof, R.; Pierrot, D.; Barbero, L. | 10.3334/CDIAC/OTG.VOS_EQNX_2015 |
| Equinox | 2015-03-07 | 2015-03-11 | Tropical Atlantic | 1588 | Wanninkhof, R.; Pierrot, D.; Barbero, L. | 10.3334/CDIAC/OTG.VOS_EQNX_2015 |
| Equinox | 2015-03-19 | 2015-03-27 | Tropical Atlantic | 2694 | Wanninkhof, R.; Pierrot, D.; Barbero, L. | 10.3334/CDIAC/OTG.VOS_EQNX_2015 |
| Equinox | 2015-03-27 | 2015-04-06 | Tropical Atlantic | 3607 | Wanninkhof, R.; Pierrot, D.; Barbero, L. | 10.3334/CDIAC/OTG.VOS_EQNX_2015 |
| Equinox | 2015-04-06 | 2015-04-17 | Tropical Atlantic | 3750 | Wanninkhof, R.; Pierrot, D.; Barbero, L. | 10.3334/CDIAC/OTG.VOS_EQNX_2015 |
| Equinox | 2015-04-17 | 2015-04-27 | Tropical Atlantic | 3611 | Wanninkhof, R.; Pierrot, D.; Barbero, L. | 10.3334/CDIAC/OTG.VOS_EQNX_2015 |
| Equinox | 2015-04-28 | 2015-05-11 | North Atlantic, Tropical Atlantic | 5151 | Wanninkhof, R.; Pierrot, D.; Barbero, L. | 10.3334/CDIAC/OTG.VOS_EQNX_2015 |
| Equinox | 2015-05-11 | 2015-05-21 | North Atlantic | 2323 | Wanninkhof, R.; Pierrot, D.; Barbero, L. | 10.3334/CDIAC/OTG.VOS_EQNX_2015 |
| Equinox | 2015-05-21 | 2015-06-02 | North Atlantic | 3565 | Wanninkhof, R.; Pierrot, D.; Barbero, L. | 10.3334/CDIAC/OTG.VOS_EQNX_2015 |
| Equinox | 2015-06-02 | 2015-06-04 | North Atlantic | 484 | Wanninkhof, R.; Pierrot, D.; Barbero, L. | 10.3334/CDIAC/OTG.VOS_EQNX_2015 |
| Explorer of the Seas | 2014-12-27 | 2015-01-04 | Tropical Atlantic | 2804 | Wanninkhof, R.; Pierrot, D.; Barbero, L. | 10.3334/CDIAC/OTG.VOS_EXP2014 |
| Explorer of the Seas | 2015-01-04 | 2015-01-09 | Tropical Atlantic | 1698 | Wanninkhof, R.; Pierrot, D.; Barbero, L. | 10.3334/CDIAC/OTG.VOS_EXP2015 |
| Explorer of the Seas | 2015-01-09 | 2015-01-18 | Tropical Atlantic | 3176 | Wanninkhof, R.; Pierrot, D.; Barbero, L. | 10.3334/CDIAC/OTG.VOS_EXP2015 |
| Explorer of the Seas | 2015-01-18 | 2015-01-24 | Tropical Atlantic | 2058 | Wanninkhof, R.; Pierrot, D.; Barbero, L. | 10.3334/CDIAC/OTG.VOS_EXP2015 |
| Explorer of the Seas | 2015-01-24 | 2015-01-29 | Tropical Atlantic | 1587 | Wanninkhof, R.; Pierrot, D.; Barbero, L. | 10.3334/CDIAC/OTG.VOS_EXP2015 |
| Explorer of the Seas | 2015-01-29 | 2015-02-07 | Tropical Atlantic | 3176 | Wanninkhof, R.; Pierrot, D.; Barbero, L. | 10.3334/CDIAC/OTG.VOS_EXP2015 |
| Explorer of the Seas | 2015-02-07 | 2015-02-12 | Tropical Atlantic | 1707 | Wanninkhof, R.; Pierrot, D.; Barbero, L. | 10.3334/CDIAC/OTG.VOS_EXP2015 |
| Explorer of the Seas | 2015-02-12 | 2015-02-15 | Tropical Atlantic | 1289 | Wanninkhof, R.; Pierrot, D.; Barbero, L. | 10.3334/CDIAC/OTG.VOS_EXP2015 |
| F.G. Walton Smith | 2015-01-12 | 2015-01-14 | Tropical Atlantic | 816 | Millero, F.; Wanninkhof, R. | |
| F.G. Walton Smith | 2015-04-09 | 2015-04-10 | Tropical Atlantic | 613 | Millero, F.; Wanninkhof, R. | |
| F.G. Walton Smith | 2015-04-13 | 2015-04-17 | Tropical Atlantic | 2078 | Millero, F.; Wanninkhof, R. | |
| F.G. Walton Smith | 2015-04-22 | 2015-05-02 | Tropical Atlantic | 3514 | Millero, F.; Wanninkhof, R. | |
| F.G. Walton Smith | 2015-05-07 | 2015-05-20 | Tropical Atlantic | 6523 | Millero, F.; Wanninkhof, R. | |
| F.G. Walton Smith | 2015-05-26 | 2015-05-27 | Tropical Atlantic | 684 | Millero, F.; Wanninkhof, R. | |
| F.G. Walton Smith | 2015-06-01 | 2015-06-05 | Tropical Atlantic | 2038 | Millero, F.; Wanninkhof, R. | |
| F.G. Walton Smith | 2015-06-10 | 2015-06-27 | Tropical Atlantic | 7319 | Millero, F.; Wanninkhof, R. | |
| F.G. Walton Smith | 2015-07-14 | 2015-07-15 | Tropical Atlantic | 689 | Millero, F.; Wanninkhof, R. | |
| F.G. Walton Smith | 2015-07-27 | 2015-08-01 | Tropical Atlantic | 2258 | Millero, F.; Wanninkhof, R. | |
| F.G. Walton Smith | 2015-08-22 | 2015-09-04 | Tropical Atlantic | 6600 | Millero, F.; Wanninkhof, R. | |

| | | | | | | |
|---|---|---|---|---|---|---|
| F.G. Walton Smith | 2015-09-21 | 2015-09-25 | Tropical Atlantic | 2096 | Millero, F.; Wanninkhof, R. | |
| F.G. Walton Smith | 2015-09-28 | 2015-10-02 | Tropical Atlantic | 1990 | Millero, F.; Wanninkhof, R. | |
| F.G. Walton Smith | 2015-10-27 | 2015-11-06 | North Atlantic, Tropical Atlantic | 3896 | Millero, F.; Wanninkhof, R. | |
| F.G. Walton Smith | 2015-11-10 | 2015-11-11 | Tropical Atlantic | 271 | Millero, F.; Wanninkhof, R. | |
| F.G. Walton Smith | 2015-11-16 | 2015-11-20 | Tropical Atlantic | 82 | Millero, F.; Wanninkhof, R. | |
| G.O. Sars | 2015-01-17 | 2015-02-10 | North Atlantic | 9661 | Lauvset, S.K. | |
| G.O. Sars | 2015-04-12 | 2015-04-25 | North Atlantic | 11719 | Lauvset, S.K.; Skjelvan, I. | |
| G.O. Sars | 2015-04-29 | 2015-05-01 | North Atlantic | 2939 | Lauvset, S.K.; Skjelvan, I. | |
| G.O. Sars | 2015-07-05 | 2015-07-14 | North Atlantic | 8921 | Lauvset, S.K.; Skjelvan, I. | |
| G.O. Sars | 2015-07-21 | 2015-08-13 | North Atlantic | 20088 | Lauvset, S.K.; Skjelvan, I. | |
| G.O. Sars | 2015-08-18 | 2015-09-05 | North Atlantic | 18076 | Lauvset, S.K.; Skjelvan, I. | |
| G.O. Sars | 2015-09-12 | 2015-09-25 | North Atlantic | 11327 | Lauvset, S.K.; Skjelvan, I. | |
| G.O. Sars | 2015-09-30 | 2015-10-14 | Arctic, North Atlantic | 13610 | Lauvset, S.K.; Skjelvan, I. | |
| G.O. Sars | 2015-10-27 | 2015-11-03 | North Atlantic | 6937 | Lauvset, S.K.; Skjelvan, I. | |
| Gordon Gunter | 2015-03-04 | 2015-03-14 | Tropical Atlantic | 4678 | Wanninkhof, R.; Pierrot, D.; Barbero, L. | 10.3334/CDIAC/OTG.COAST_GU2015_UW |
| Gordon Gunter | 2015-03-18 | 2015-04-02 | Tropical Atlantic | 5015 | Wanninkhof, R.; Pierrot, D.; Barbero, L. | 10.3334/CDIAC/OTG.COAST_GU2015_UW |
| Gordon Gunter | 2015-04-15 | 2015-04-27 | North Atlantic, Tropical Atlantic | 4334 | Wanninkhof, R.; Pierrot, D.; Barbero, L. | 10.3334/CDIAC/OTG.COAST_GU2015_UW |
| Gordon Gunter | 2015-05-16 | 2015-06-05 | North Atlantic | 9118 | Wanninkhof, R.; Pierrot, D.; Barbero, L. | 10.3334/CDIAC/OTG.COAST_GU2015_UW |
| Gordon Gunter | 2015-06-09 | 2015-06-12 | North Atlantic | 1031 | Wanninkhof, R.; Pierrot, D.; Barbero, L. | 10.3334/CDIAC/OTG.COAST_GU2015_UW |
| Gordon Gunter | 2015-06-19 | 2015-07-03 | North Atlantic | 5688 | Wanninkhof, R.; Pierrot, D.; Barbero, L. | 10.3334/CDIAC/OTG.COAST_GU2015_UW |
| Gordon Gunter | 2015-07-08 | 2015-07-24 | North Atlantic, Tropical Atlantic | 7293 | Wanninkhof, R.; Pierrot, D.; Barbero, L. | 10.3334/CDIAC/OTG.COAST_GU2015_UW |
| Gordon Gunter | 2015-07-30 | 2015-08-16 | Tropical Atlantic | 7434 | Wanninkhof, R.; Pierrot, D.; Barbero, L. | 10.3334/CDIAC/OTG.COAST_GU2015_UW |
| Gordon Gunter | 2015-08-23 | 2015-09-06 | Tropical Atlantic | 6452 | Wanninkhof, R.; Pierrot, D.; Barbero, L. | 10.3334/CDIAC/OTG.COAST_GU2015_UW |
| Gordon Gunter | 2015-09-14 | 2015-09-28 | Tropical Atlantic | 6111 | Wanninkhof, R.; Pierrot, D.; Barbero, L. | 10.3334/CDIAC/OTG.COAST_GU2015_UW |
| Gulf Challenger | 2015-03-13 | 2015-03-13 | North Atlantic | 1148 | Vandemark, D.; Salisbury, J. ; Hunt, C. | 10.3334/CDIAC/otg.TSM_UNH_GOM |
| Gulf Challenger | 2015-06-05 | 2015-06-05 | North Atlantic | 1071 | Vandemark, D.; Salisbury, J. ; Hunt, C. | 10.3334/CDIAC/otg.TSM_UNH_GOM |
| Gulf Challenger | 2015-08-26 | 2015-08-26 | North Atlantic | 1127 | Vandemark, D.; Salisbury, J. ; Hunt, C. | 10.3334/CDIAC/otg.TSM_UNH_GOM |
| Gulf Challenger | 2015-10-07 | 2015-10-07 | North Atlantic | 1078 | Vandemark, D.; Salisbury, J. ; Hunt, C. | 10.3334/CDIAC/otg.TSM_UNH_GOM |
| Gulf Challenger | 2015-11-18 | 2015-11-18 | North Atlantic | 960 | Vandemark, D.; Salisbury, J. ; Hunt, C. | 10.3334/CDIAC/otg.TSM_UNH_GOM |
| Healy | 2015-07-14 | 2015-07-24 | Arctic, North Pacific | 4121 | Sutherland, S.C.; Newberger, T.; | 10.3334/CDIAC/OTG.VOS_Healy_Lines_2015 |

| | | | | | | |
|---|---|---|---|---|---|---|
| | | | | | Takahashi, T. | |
| Healy | 2015-08-11 | 2015-10-21 | Arctic, North Pacific | 27033 | Sutherland, S.C.; Newberger, T.; Takahashi, T. | 10.3334/CDIAC/OTG.VOS_Healy_Lines_2015 |
| Healy | 2015-10-26 | 2015-10-28 | North Pacific | 960 | Sutherland, S.C.; Newberger, T.; Takahashi, T. | 10.3334/CDIAC/OTG.VOS_Healy_Lines_2015 |
| Henry B. Bigelow | 2015-03-12 | 2015-03-21 | North Atlantic | 3525 | Wanninkhof, R.; Pierrot, D.; Barbero, L. | 10.3334/CDIAC/OTG.AOML_BIGELOW_ECOAST_2015 |
| Henry B. Bigelow | 2015-03-23 | 2015-04-03 | North Atlantic | 5059 | Wanninkhof, R.; Pierrot, D.; Barbero, L. | 10.3334/CDIAC/OTG.AOML_BIGELOW_ECOAST_2015 |
| Henry B. Bigelow | 2015-04-07 | 2015-04-23 | North Atlantic | 6155 | Wanninkhof, R.; Pierrot, D.; Barbero, L. | 10.3334/CDIAC/OTG.AOML_BIGELOW_ECOAST_2015 |
| Henry B. Bigelow | 2015-04-27 | 2015-05-07 | North Atlantic | 4638 | Wanninkhof, R.; Pierrot, D.; Barbero, L. | 10.3334/CDIAC/OTG.AOML_BIGELOW_ECOAST_2015 |
| Henry B. Bigelow | 2015-05-19 | 2015-06-03 | North Atlantic | 6456 | Wanninkhof, R.; Pierrot, D.; Barbero, L. | 10.3334/CDIAC/OTG.AOML_BIGELOW_ECOAST_2015 |
| Henry B. Bigelow | 2015-06-11 | 2015-06-19 | North Atlantic | 3839 | Wanninkhof, R.; Pierrot, D.; Barbero, L. | 10.3334/CDIAC/OTG.AOML_BIGELOW_ECOAST_2015 |
| Henry B. Bigelow | 2015-06-24 | 2015-07-02 | North Atlantic | 3401 | Wanninkhof, R.; Pierrot, D.; Barbero, L. | 10.3334/CDIAC/OTG.AOML_BIGELOW_ECOAST_2015 |
| Henry B. Bigelow | 2015-07-27 | 2015-08-07 | North Atlantic | 5265 | Wanninkhof, R.; Pierrot, D.; Barbero, L. | 10.3334/CDIAC/OTG.AOML_BIGELOW_ECOAST_2015 |
| Henry B. Bigelow | 2015-08-12 | 2015-08-21 | North Atlantic | 4315 | Wanninkhof, R.; Pierrot, D.; Barbero, L. | 10.3334/CDIAC/OTG.AOML_BIGELOW_ECOAST_2015 |
| Henry B. Bigelow | 2015-09-01 | 2015-09-17 | North Atlantic | 7836 | Wanninkhof, R.; Pierrot, D.; Barbero, L. | 10.3334/CDIAC/OTG.AOML_BIGELOW_ECOAST_2015 |
| Henry B. Bigelow | 2015-09-23 | 2015-09-30 | North Atlantic | 3382 | Wanninkhof, R.; Pierrot, D.; Barbero, L. | 10.3334/CDIAC/OTG.AOML_BIGELOW_ECOAST_2015 |
| Henry B. Bigelow | 2015-10-07 | 2015-10-22 | North Atlantic | 7186 | Wanninkhof, R.; Pierrot, D.; Barbero, L. | 10.3334/CDIAC/OTG.AOML_BIGELOW_ECOAST_2015 |
| Henry B. Bigelow | 2015-10-27 | 2015-11-06 | North Atlantic | 4472 | Wanninkhof, R.; Pierrot, D.; Barbero, L. | 10.3334/CDIAC/OTG.AOML_BIGELOW_ECOAST_2015 |
| Henry B. Bigelow | 2015-11-12 | 2015-11-17 | North Atlantic | 2402 | Wanninkhof, R.; Pierrot, D.; Barbero, L. | 10.3334/CDIAC/OTG.AOML_BIGELOW_ECOAST_2015 |
| Laurence M. Gould | 2014-12-30 | 2015-02-07 | Southern Ocean | 7302 | Sweeney, C.; Takahashi, T.; Newberger, T.; Sutherland, S.C.; Munro, D.R. | 10.3334/CDIAC/OTG.VOS_LM_GOULD_2014 |
| Laurence M. Gould | 2015-02-14 | 2015-03-16 | Southern Ocean | 9450 | Sweeney, C.; Takahashi, T.; Newberger, T.; Sutherland, S.C.; Munro, D.R. | 10.3334/CDIAC/OTG.VOS_LM_GOULD_2015 |
| Laurence M. Gould | 2015-03-21 | 2015-04-03 | Southern Ocean | 2602 | Sweeney, C.; Takahashi, T.; Newberger, T.; Sutherland, S.C.; Munro, D.R. | 10.3334/CDIAC/OTG.VOS_LM_GOULD_2015 |
| Laurence M. Gould | 2015-04-08 | 2015-05-11 | Southern Ocean | 7691 | Sweeney, C.; Takahashi, T.; Newberger, T.; Sutherland, S.C.; Munro, D.R. | 10.3334/CDIAC/OTG.VOS_LM_GOULD_2015 |
| Laurence M. Gould | 2015-05-16 | 2015-06-16 | Southern Ocean | 9497 | Sweeney, C.; Takahashi, T.; Newberger, T.; Sutherland, S.C.; Munro, D.R. | 10.3334/CDIAC/OTG.VOS_LM_GOULD_2015 |
| Laurence M. Gould | 2015-06-21 | 2015-06-30 | Southern Ocean | 2379 | Sweeney, C.; Takahashi, T.; Newberger, T.; Sutherland, S.C.; Munro, D.R. | 10.3334/CDIAC/OTG.VOS_LM_GOULD_2015 |
| Marcus G. Langseth | 2015-04-13 | 2015-04-22 | North Atlantic | 1948 | Sutherland, S.C.; Newberger, T.; Takahashi, T.; Sweeney, C. | 10.3334/CDIAC/OTG.VOS_MG_LANGSETH_LINES_2015 |
| Marcus G. Langseth | 2015-06-01 | 2015-06-23 | North Atlantic | 8608 | Sutherland, S.C.; Newberger, T.; Takahashi, T.; Sweeney, C. | 10.3334/CDIAC/OTG.VOS_MG_LANGSETH_LINES_2015 |
| Marcus G. Langseth | 2015-07-31 | 2015-09-12 | North Atlantic | 14519 | Sutherland, S.C.; Newberger, T.; Takahashi, T.; Sweeney, C. | 10.3334/CDIAC/OTG.VOS_MG_LANGSETH_LINES_2015 |
| Marion Dufresne | 2015-01-07 | 2015-02-06 | Indian Ocean, Southern Ocean | 4529 | Metzl, N.; Lo Monaco, C. | 10.3334/CDIAC/OTG.VOS_OISO_24 |

| Platform | Start | End | Region | Count | Authors | DOI |
|---|---|---|---|---|---|---|
| Mooring | 2014-03-07 | 2015-03-22 | Tropical Atlantic | 3048 | Sutton, A.; Sabine, C.; Manzello, D.; Musielewicz, S.; Maenner, S.; Dietrich, C.; Bott, R.; Osborne, J. | 10.3334/CDIAC/OTG.CHEECA_80W_25N |
| Mooring | 2014-03-07 | 2015-04-03 | Tropical Pacific | 3129 | Sutton, A.; Sabine, C.; Maenner, S.; Musielewicz, S.; Bott, R.; Osborne, J. | 10.3334/CDIAC/otg.TSM_Stratus_85W_20S |
| Mooring | 2014-05-02 | 2015-04-28 | North Pacific | 2630 | Sutton, A.; Sabine, C.; Send, U.; Ohman, M.; Musielewicz, S.; Maenner, S.; Dietrich, C.; Bott, R.; Osborne, J. | 10.3334/CDIAC/OTG.TSM_CCE2_121W_34N |
| Mooring | 2014-05-06 | 2015-01-27 | North Pacific | 2122 | Mathis, J.; Monacci, N.; Musielewicz, S.; Maenner, S. | 10.3334/CDIAC/OTG.TSM_Southeast_AK_56N_134W |
| Mooring | 2014-05-24 | 2015-05-06 | Tropical Pacific | 2447 | Sutton, A.; Sabine, C.; De Carlo, E.; Musielewicz, S.; Maenner, S.; Dietrich, C.; Bott, R.; Osborne, J. | 10.3334/CDIAC/OTG.TSM_Kaneohe_158W_21N |
| Mooring | 2014-07-21 | 2015-07-07 | North Atlantic | 2796 | Sutton, A.; Sabine, C.; Andersson, A.; Bates, N.; Musielewicz, S.; Maenner, S.; Dietrich, C.; Bott, R.; Osborne, J. | 10.3334/CDIAC/OTG.TSM_Crescent_64W_32N |
| Mooring | 2014-10-06 | 2015-01-07 | North Atlantic | 741 | Sutton, A.; Sabine, C.; Maenner, S.; Musielewicz, S.; Bott, R.; Osborne, J. | 10.3334/CDIAC/OTG.TSM_Hog_Reef_64W_32N |
| Nathaniel B. Palmer | 2015-01-06 | 2015-01-18 | Southern Ocean | 4320 | Sutherland, S.C.; Newberger, T.; Takahashi, T.; Sweeney, C. | 10.3334/CDIAC/OTG.VOS_PALMER_2015 |
| Nathaniel B. Palmer | 2015-01-23 | 2015-03-14 | Southern Ocean | 17383 | Sutherland, S.C.; Newberger, T.; Takahashi, T.; Sweeney, C. | 10.3334/CDIAC/OTG.VOS_PALMER_2015 |
| Nathaniel B. Palmer | 2015-03-27 | 2015-04-28 | Southern Ocean | 10623 | Sutherland, S.C.; Newberger, T.; Takahashi, T.; Sweeney, C. | 10.3334/CDIAC/OTG.VOS_PALMER_2015 |
| Nathaniel B. Palmer | 2015-05-12 | 2015-05-28 | Southern Ocean | 5654 | Sutherland, S.C.; Newberger, T.; Takahashi, T.; Sweeney, C. | 10.3334/CDIAC/OTG.VOS_PALMER_2015 |
| Nathaniel B. Palmer | 2015-08-05 | 2015-08-28 | Southern Ocean | 7528 | Sutherland, S.C.; Newberger, T.; Takahashi, T.; Sweeney, C. | 10.3334/CDIAC/OTG.VOS_PALMER_2015 |
| Nathaniel B. Palmer | 2015-09-08 | 2015-10-18 | Southern Ocean, Tropical Atlantic | 13871 | Sutherland, S.C.; Newberger, T.; Takahashi, T.; Sweeney, C. | 10.3334/CDIAC/OTG.VOS_PALMER_2015 |
| New Century 2 | 2014-12-12 | 2015-01-12 | North Pacific, Tropical Pacific | 3221 | Nakaoka, S. | 10.3334/CDIAC/OTG.VOS_New_Century_2_2014 |
| New Century 2 | 2015-03-16 | 2015-03-31 | North Pacific | 1343 | Nakaoka, S. | 10.3334/CDIAC/OTG.VOS_New_Century_2_2015 |
| New Century 2 | 2015-04-01 | 2015-04-14 | North Pacific | 1417 | Nakaoka, S. | 10.3334/CDIAC/OTG.VOS_New_Century_2_2015 |
| New Century 2 | 2015-04-16 | 2015-05-03 | North Pacific | 1668 | Nakaoka, S. | 10.3334/CDIAC/OTG.VOS_New_Century_2_2015 |
| New Century 2 | 2015-05-04 | 2015-05-17 | North Pacific | 1616 | Nakaoka, S. | 10.3334/CDIAC/OTG.VOS_New_Century_2_2015 |
| New Century 2 | 2015-05-20 | 2015-06-04 | North Pacific | 1569 | Nakaoka, S. | 10.3334/CDIAC/OTG.VOS_New_Century_2_2015 |
| New Century 2 | 2015-06-05 | 2015-06-21 | North Pacific | 1545 | Nakaoka, S. | 10.3334/CDIAC/OTG.VOS_New_Century_2_2015 |
| New Century 2 | 2015-06-23 | 2015-07-07 | North Pacific | 1376 | Nakaoka, S. | 10.3334/CDIAC/OTG.VOS_New_Century_2_2015 |
| New Century 2 | 2015-07-07 | 2015-07-20 | North Pacific | 1440 | Nakaoka, S. | 10.3334/CDIAC/OTG.VOS_New_Century_2_2015 |
| New Century 2 | 2015-07-23 | 2015-08-07 | North Pacific | 1538 | Nakaoka, S. | 10.3334/CDIAC/OTG.VOS_New_Century_2_2015 |
| New Century 2 | 2015-08-09 | 2015-08-21 | North Pacific | 1460 | Nakaoka, S. | 10.3334/CDIAC/OTG.VOS_New_Century_2_2015 |

| | | | | | | |
|---|---|---|---|---|---|---|
| New Century 2 | 2015-08-26 | 2015-09-24 | North Pacific, Tropical Pacific | 2422 | Nakaoka, S. | 10.3334/CDIAC/OTG.VOS_New_Century_2_2015 |
| New Century 2 | 2015-09-24 | 2015-10-23 | North Atlantic, North Pacific, Tropical Atlantic, Tropical Pacific | 3157 | Nakaoka, S. | 10.3334/CDIAC/OTG.VOS_New_Century_2_2015 |
| Nuka Arctica | 2015-10-21 | 2015-11-08 | North Atlantic | 5318 | Omar, A.; Olsen, A.; Johannessen, T. | |
| Nuka Arctica | 2015-12-01 | 2015-12-21 | North Atlantic | 10558 | Omar, A.; Olsen, A.; Johannessen, T. | |
| Polarstern | 2014-12-03 | 2015-01-31 | Southern Ocean | 58046 | van Heuven, S.; Hoppema, M. | 10.3334/CDIAC/OTG.OA_VOS_POLARSTERN_2014 |
| Polarstern | 2015-05-19 | 2015-06-27 | Arctic, North Atlantic | 39056 | van Heuven, S.; Hoppema, M. | 10.3334/CDIAC/OTG.OA_VOS_POLARSTERN_2015 |
| Polarstern | 2015-06-29 | 2015-08-14 | Arctic, North Atlantic | 20164 | van Heuven, S.; Hoppema, M. | 10.3334/CDIAC/OTG.OA_VOS_POLARSTERN_2015 |
| Polarstern | 2015-08-18 | 2015-10-11 | Arctic, North Atlantic | 43709 | van Heuven, S.; Hoppema, M. | 10.3334/CDIAC/OTG.OA_VOS_POLARSTERN_2015 |
| Polarstern | 2015-10-30 | 2015-12-01 | North Atlantic, Southern Ocean, Tropical Atlantic | 27178 | van Heuven, S.; Hoppema, M. | 10.3334/CDIAC/OTG.OA_VOS_POLARSTERN_2015 |
| Ronald H. Brown | 2015-01-15 | 2015-01-29 | North Pacific, Tropical Pacific | 4855 | Wanninkhof, R.; Pierrot, D.; Barbero, L. | 10.3334/CDIAC/OTG.VOS_RB_2015 |
| Ronald H. Brown | 2015-01-30 | 2015-02-12 | North Pacific | 5365 | Wanninkhof, R.; Pierrot, D.; Barbero, L. | 10.3334/CDIAC/OTG.VOS_RB_2015 |
| Ronald H. Brown | 2015-03-01 | 2015-03-30 | Tropical Pacific | 13576 | Wanninkhof, R.; Pierrot, D.; Barbero, L. | 10.3334/CDIAC/OTG.VOS_RB_2015 |
| Ronald H. Brown | 2015-04-10 | 2015-05-12 | Tropical Pacific | 15021 | Wanninkhof, R.; Pierrot, D.; Barbero, L. | 10.3334/CDIAC/OTG.VOS_RB_2015 |
| Ronald H. Brown | 2015-05-25 | 2015-06-24 | North Pacific, Tropical Pacific | 13690 | Wanninkhof, R.; Pierrot, D.; Barbero, L. | 10.3334/CDIAC/OTG.VOS_RB_2015 |
| Ronald H. Brown | 2015-07-14 | 2015-07-31 | North Pacific | 5862 | Wanninkhof, R.; Pierrot, D.; Barbero, L. | 10.3334/CDIAC/OTG.VOS_RB_2015 |
| Ronald H. Brown | 2015-08-06 | 2015-08-21 | Arctic, North Pacific | 6365 | Wanninkhof, R.; Pierrot, D.; Barbero, L. | 10.3334/CDIAC/OTG.VOS_RB_2015 |
| Ronald H. Brown | 2015-08-22 | 2015-09-04 | Arctic, North Pacific | 6298 | Wanninkhof, R.; Pierrot, D.; Barbero, L. | 10.3334/CDIAC/OTG.VOS_RB_2015 |
| Ronald H. Brown | 2015-11-22 | 2015-12-18 | Tropical Pacific | 10838 | Wanninkhof, R.; Pierrot, D.; Barbero, L. | 10.3334/CDIAC/OTG.VOS_RB_2015 |
| S.A. Agulhas II | 2014-12-08 | 2015-02-16 | Southern Ocean | 23342 | Monteiro, P.M.S.; Joubert, W.R.; Gregor, L. | 10.3334/CDIAC/OTG.VOS_SA_Agulhas_II_2015 |
| S.A. Agulhas II | 2015-07-23 | 2015-08-12 | Southern Ocean | 16271 | Monteiro, P.M.S.; Joubert, W.R.; Gregor, L. | 10.3334/CDIAC/OTG.VOS_SA_Agulhas_II_2015 |
| S.A. Agulhas II | 2015-09-04 | 2015-10-06 | Southern Ocean | 12371 | Monteiro, P.M.S.; Joubert, W.R.; Gregor, L. | 10.3334/CDIAC/OTG.VOS_SA_Agulhas_II_2015 |
| Simon Stevin | 2015-06-01 | 2015-06-01 | North Atlantic | 445 | Gkritzalis, T.; Cattrijsse, A. | |
| Simon Stevin | 2015-06-04 | 2015-06-04 | North Atlantic | 909 | Gkritzalis, T.; Cattrijsse, A. | |
| Simon Stevin | 2015-06-08 | 2015-06-08 | North Atlantic | 440 | Gkritzalis, T.; Cattrijsse, A. | |
| Simon Stevin | 2015-06-23 | 2015-06-23 | North Atlantic | 749 | Gkritzalis, T.; Cattrijsse, A. | |
| Simon Stevin | 2015-06-24 | 2015-06-24 | North Atlantic | 1234 | Gkritzalis, T.; Cattrijsse, A. | |
| Simon Stevin | 2015-06-25 | 2015-06-25 | North Atlantic | 787 | Gkritzalis, T.; Cattrijsse, A. | |
| Simon Stevin | 2015-06-30 | 2015-06-30 | North Atlantic | 425 | Gkritzalis, T.; Cattrijsse, A. | |

| Simon Stevin | 2015-07-02 | 2015-07-02 | North Atlantic | 154 | Gkritzalis, T.; Cattrijsse, A. |
| Simon Stevin | 2015-07-06 | 2015-07-06 | North Atlantic | 168 | Gkritzalis, T.; Cattrijsse, A. |
| Simon Stevin | 2015-07-10 | 2015-07-10 | North Atlantic | 357 | Gkritzalis, T.; Cattrijsse, A. |
| Simon Stevin | 2015-07-13 | 2015-07-13 | North Atlantic | 223 | Gkritzalis, T.; Cattrijsse, A. |
| Simon Stevin | 2015-07-14 | 2015-07-14 | North Atlantic | 54 | Gkritzalis, T.; Cattrijsse, A. |
| Simon Stevin | 2015-07-15 | 2015-07-15 | North Atlantic | 477 | Gkritzalis, T.; Cattrijsse, A. |
| Simon Stevin | 2015-07-16 | 2015-07-16 | North Atlantic | 465 | Gkritzalis, T.; Cattrijsse, A. |
| Simon Stevin | 2015-07-22 | 2015-07-22 | North Atlantic | 87 | Gkritzalis, T.; Cattrijsse, A. |
| Simon Stevin | 2015-07-23 | 2015-07-23 | North Atlantic | 428 | Gkritzalis, T.; Cattrijsse, A. |
| Simon Stevin | 2015-07-24 | 2015-07-24 | North Atlantic | 299 | Gkritzalis, T.; Cattrijsse, A. |
| Simon Stevin | 2015-07-31 | 2015-07-31 | North Atlantic | 401 | Gkritzalis, T.; Cattrijsse, A. |
| Simon Stevin | 2015-08-03 | 2015-08-03 | North Atlantic | 394 | Gkritzalis, T.; Cattrijsse, A. |
| Simon Stevin | 2015-08-04 | 2015-08-04 | North Atlantic | 412 | Gkritzalis, T.; Cattrijsse, A. |
| Simon Stevin | 2015-08-07 | 2015-08-07 | North Atlantic | 463 | Gkritzalis, T.; Cattrijsse, A. |
| Simon Stevin | 2015-08-10 | 2015-08-10 | North Atlantic | 479 | Gkritzalis, T.; Cattrijsse, A. |
| Simon Stevin | 2015-08-12 | 2015-08-12 | North Atlantic | 341 | Gkritzalis, T.; Cattrijsse, A. |
| Simon Stevin | 2015-08-17 | 2015-08-17 | North Atlantic | 439 | Gkritzalis, T.; Cattrijsse, A. |
| Simon Stevin | 2015-08-18 | 2015-08-18 | North Atlantic | 414 | Gkritzalis, T.; Cattrijsse, A. |
| Simon Stevin | 2015-08-19 | 2015-08-19 | North Atlantic | 470 | Gkritzalis, T.; Cattrijsse, A. |
| Simon Stevin | 2015-08-21 | 2015-08-21 | North Atlantic | 401 | Gkritzalis, T.; Cattrijsse, A. |
| Simon Stevin | 2015-08-24 | 2015-08-24 | North Atlantic | 450 | Gkritzalis, T.; Cattrijsse, A. |
| Simon Stevin | 2015-08-27 | 2015-08-27 | North Atlantic | 373 | Gkritzalis, T.; Cattrijsse, A. |
| Simon Stevin | 2015-08-28 | 2015-08-28 | North Atlantic | 455 | Gkritzalis, T.; Cattrijsse, A. |
| Simon Stevin | 2015-09-02 | 2015-09-02 | North Atlantic | 961 | Gkritzalis, T.; Cattrijsse, A. |
| Simon Stevin | 2015-09-03 | 2015-09-03 | North Atlantic | 450 | Gkritzalis, T.; Cattrijsse, A. |
| Simon Stevin | 2015-09-04 | 2015-09-04 | North Atlantic | 307 | Gkritzalis, T.; Cattrijsse, A. |
| Simon Stevin | 2015-09-08 | 2015-09-08 | North Atlantic | 464 | Gkritzalis, T.; Cattrijsse, A. |
| Simon Stevin | 2015-09-09 | 2015-09-09 | North Atlantic | 436 | Gkritzalis, T.; Cattrijsse, A. |
| Simon Stevin | 2015-09-10 | 2015-09-10 | North Atlantic | 469 | Gkritzalis, T.; Cattrijsse, A. |
| Simon Stevin | 2015-09-11 | 2015-09-11 | North Atlantic | 443 | Gkritzalis, T.; Cattrijsse, A. |

| | | | | | | |
|---|---|---|---|---|---|---|
| Simon Stevin | 2015-09-15 | 2015-09-15 | North Atlantic | 729 | Gkritzalis, T.; Cattrijsse, A. | |
| Simon Stevin | 2015-09-16 | 2015-09-16 | North Atlantic | 1081 | Gkritzalis, T.; Cattrijsse, A. | |
| Simon Stevin | 2015-09-21 | 2015-09-21 | North Atlantic | 366 | Gkritzalis, T.; Cattrijsse, A. | |
| Simon Stevin | 2015-09-25 | 2015-09-25 | North Atlantic | 454 | Gkritzalis, T.; Cattrijsse, A. | |
| Simon Stevin | 2015-09-28 | 2015-09-28 | North Atlantic | 440 | Gkritzalis, T.; Cattrijsse, A. | |
| Simon Stevin | 2015-09-29 | 2015-09-29 | North Atlantic | 701 | Gkritzalis, T.; Cattrijsse, A. | |
| Simon Stevin | 2015-09-30 | 2015-09-30 | North Atlantic | 850 | Gkritzalis, T.; Cattrijsse, A. | |
| Simon Stevin | 2015-10-05 | 2015-10-05 | North Atlantic | 453 | Gkritzalis, T.; Cattrijsse, A. | |
| Simon Stevin | 2015-10-06 | 2015-10-06 | North Atlantic | 491 | Gkritzalis, T.; Cattrijsse, A. | |
| Simon Stevin | 2015-10-07 | 2015-10-07 | North Atlantic | 423 | Gkritzalis, T.; Cattrijsse, A. | |
| Simon Stevin | 2015-10-08 | 2015-10-08 | North Atlantic | 437 | Gkritzalis, T.; Cattrijsse, A. | |
| Simon Stevin | 2015-10-10 | 2015-10-10 | North Atlantic | 488 | Gkritzalis, T.; Cattrijsse, A. | |
| Simon Stevin | 2015-10-11 | 2015-10-11 | North Atlantic | 448 | Gkritzalis, T.; Cattrijsse, A. | |
| Simon Stevin | 2015-10-20 | 2015-10-20 | North Atlantic | 435 | Gkritzalis, T.; Cattrijsse, A. | |
| Simon Stevin | 2015-10-21 | 2015-10-21 | North Atlantic | 319 | Gkritzalis, T.; Cattrijsse, A. | |
| Simon Stevin | 2015-11-01 | 2015-11-01 | North Atlantic | 387 | Gkritzalis, T.; Cattrijsse, A. | |
| Simon Stevin | 2015-11-04 | 2015-11-04 | North Atlantic | 272 | Gkritzalis, T.; Cattrijsse, A. | |
| Simon Stevin | 2015-11-05 | 2015-11-05 | North Atlantic | 415 | Gkritzalis, T.; Cattrijsse, A. | |
| Simon Stevin | 2015-11-06 | 2015-11-06 | North Atlantic | 114 | Gkritzalis, T.; Cattrijsse, A. | |
| Simon Stevin | 2015-11-12 | 2015-11-12 | North Atlantic | 202 | Gkritzalis, T.; Cattrijsse, A. | |
| Simon Stevin | 2015-12-07 | 2015-12-07 | North Atlantic | 217 | Gkritzalis, T.; Cattrijsse, A. | |
| Simon Stevin | 2015-12-08 | 2015-12-08 | North Atlantic | 336 | Gkritzalis, T.; Cattrijsse, A. | |
| Simon Stevin | 2015-12-09 | 2015-12-09 | North Atlantic | 156 | Gkritzalis, T.; Cattrijsse, A. | |
| Soyo Maru | 2015-05-08 | 2015-05-11 | North Pacific, Tropical Pacific | 3972 | Ono, T. | |
| Soyo Maru | 2015-08-01 | 2015-08-07 | North Pacific | 8354 | Ono, T. | |
| Soyo Maru | 2015-10-26 | 2015-11-03 | North Pacific | 10759 | Ono, T. | |
| Tangaroa | 2015-01-28 | 2015-03-10 | Southern Ocean | 34868 | Currie, K. | 10.3334/CDIAC/OTG.VOS_Tangaroa_2015 |
| Tangaroa | 2015-03-27 | 2015-04-14 | Southern Ocean | 15297 | Currie, K. | 10.3334/CDIAC/OTG.VOS_Tangaroa_2015 |
| Tangaroa | 2015-04-17 | 2015-04-22 | Southern Ocean | 4797 | Currie, K. | 10.3334/CDIAC/OTG.VOS_Tangaroa_2015 |
| Tangaroa | 2015-04-23 | 2015-04-30 | Southern Ocean | 5791 | Currie, K. | 10.3334/CDIAC/OTG.VOS_Tangaroa_2015 |

| | | | | | | |
|---|---|---|---|---|---|---|
| Tangaroa | 2015-05-04 | 2015-05-21 | Southern Ocean | 12051 | Currie, K. | 10.3334/CDIAC/OTG.VOS_Tangaroa_2015 |
| Tangaroa | 2015-05-23 | 2015-06-01 | Southern Ocean | 7985 | Currie, K. | 10.3334/CDIAC/OTG.VOS_Tangaroa_2015 |
| Tangaroa | 2015-07-04 | 2015-08-02 | Southern Ocean | 26898 | Currie, K. | 10.3334/CDIAC/OTG.VOS_Tangaroa_2015 |
| Tangaroa | 2015-08-04 | 2015-08-26 | Southern Ocean | 18553 | Currie, K. | 10.3334/CDIAC/OTG.VOS_Tangaroa_2015 |
| Tangaroa | 2015-09-05 | 2015-09-24 | Southern Ocean | 12776 | Currie, K. | 10.3334/CDIAC/OTG.VOS_Tangaroa_2015 |
| Tangaroa | 2015-09-26 | 2015-10-04 | Tropical Pacific | 7207 | Currie, K. | 10.3334/CDIAC/OTG.VOS_Tangaroa_2014 |
| Tangaroa | 2015-10-13 | 2015-10-25 | Southern Ocean | 10658 | Currie, K. | 10.3334/CDIAC/OTG.VOS_Tangaroa_2015 |
| Trans Future 5 | 2015-01-10 | 2015-01-24 | North Pacific, Southern Ocean, Tropical Pacific | 1507 | Nojiri, Y. | 10.3334/CDIAC/OTG.VOS_TF5_2015 |
| Trans Future 5 | 2015-01-31 | 2015-02-10 | North Pacific, Tropical Pacific | 1179 | Nojiri, Y. | 10.3334/CDIAC/OTG.VOS_TF5_2015 |
| Trans Future 5 | 2015-02-11 | 2015-02-24 | Southern Ocean, Tropical Pacific | 922 | Nojiri, Y. | 10.3334/CDIAC/OTG.VOS_TF5_2015 |
| Trans Future 5 | 2015-02-25 | 2015-03-08 | North Pacific, Southern Ocean, Tropical Pacific | 1379 | Nojiri, Y. | 10.3334/CDIAC/OTG.VOS_TF5_2015 |
| Trans Future 5 | 2015-03-14 | 2015-03-24 | North Pacific, Tropical Pacific | 1083 | Nojiri, Y. | 10.3334/CDIAC/OTG.VOS_TF5_2015 |
| Trans Future 5 | 2015-04-25 | 2015-05-05 | North Pacific, Tropical Pacific | 1090 | Nakaoka, S. | 10.3334/CDIAC/OTG.VOS_TF5_2015 |
| Trans Future 5 | 2015-05-06 | 2015-05-20 | Southern Ocean, Tropical Pacific | 913 | Nakaoka, S. | 10.3334/CDIAC/OTG.VOS_TF5_2015 |
| Trans Future 5 | 2015-05-21 | 2015-06-01 | North Pacific, Southern Ocean, Tropical Pacific | 1381 | Nakaoka, S. | 10.3334/CDIAC/OTG.VOS_TF5_2015 |
| Trans Future 5 | 2015-06-06 | 2015-06-15 | North Pacific, Tropical Pacific | 1138 | Nakaoka, S. | 10.3334/CDIAC/OTG.VOS_TF5_2015 |
| Trans Future 5 | 2015-06-16 | 2015-06-28 | Southern Ocean, Tropical Pacific | 911 | Nakaoka, S. | 10.3334/CDIAC/OTG.VOS_TF5_2015 |
| Trans Future 5 | 2015-06-29 | 2015-07-12 | North Pacific, Southern Ocean, Tropical Pacific | 1431 | Nakaoka, S. | 10.3334/CDIAC/OTG.VOS_TF5_2015 |
| Trans Future 5 | 2015-07-18 | 2015-07-30 | North Pacific, Tropical Pacific | 1112 | Nakaoka, S. | 10.3334/CDIAC/OTG.VOS_TF5_2015 |
| Trans Future 5 | 2015-07-30 | 2015-08-11 | Southern Ocean, Tropical Pacific | 884 | Nakaoka, S. | 10.3334/CDIAC/OTG.VOS_TF5_2015 |
| Trans Future 5 | 2015-08-12 | 2015-08-24 | North Pacific, Southern Ocean, Tropical Pacific | 1400 | Nakaoka, S. | 10.3334/CDIAC/OTG.VOS_TF5_2015 |
| Trans Future 5 | 2015-09-26 | 2015-10-07 | North Pacific, Tropical Pacific | 811 | Nakaoka, S. | 10.3334/CDIAC/OTG.VOS_TF5_2015 |
| Trans Future 5 | 2015-10-07 | 2015-10-19 | Southern Ocean, Tropical Pacific | 889 | Nakaoka, S. | 10.3334/CDIAC/OTG.VOS_TF5_2015 |
| Trans Future 5 | 2015-10-21 | 2015-11-01 | North Pacific, Southern Ocean, Tropical Pacific | 1427 | Nakaoka, S. | 10.3334/CDIAC/OTG.VOS_TF5_2015 |
| Wakataka Maru | 2015-06-30 | 2015-07-04 | North Pacific | 6356 | Kuwata, A.; Tadokoro, K., Ono, T. | |
| Wakataka Maru | 2015-07-11 | 2015-07-21 | North Pacific | 14479 | Kuwata, A.; Tadokoro, K., Ono, T. | |
| Wakataka Maru | 2015-07-29 | 2015-08-05 | North Pacific | 9773 | Kuwata, A.; Tadokoro, K., Ono, T. | |
| Wakataka Maru | 2015-09-30 | 2015-10-15 | North Pacific | 15111 | Kuwata, A.; Tadokoro, K., Ono, T. | |

