# Peer review of "Global Carbon Budget 2016"

_Earth System Science Data, 2016_

## Editor Comment (EC1) · D. J. Carlson (Editor) · 12 Oct 2016

ESSD seeks to endorse and honour the strong and clear open access policy of the Copernicus publication system. In the case of ESSD we extend that open access policy to all the data and data descriptions that we publish. In very rare cases we allow a short period of temporary closed access during a review process, to prevent premature disclosure in the case of high impact data and to facilitate collaboration with a non-open-access science journal - authors in that case connect a research paper in the science journal to the data in ESSD and vice versa. For both reasons we will process this Global Carbon Budget for 2016 without open access for a period of a few weeks as we and the partner journal complete our usual review processes. All ESSD materials - original manuscript, reviewer comments, and author responses, revised data set - will become fully open access at the time of final publication.

---

## Referee Report (RR1)

Review global carbon budget ESSD-2016-51

Overall a very strong product with substantial scientific and political impact. All data well organised and easily and freely accessible. I applaud the global carbon project for this continuing effort to deliver an accurate annual account of the carbon system. Apparently a good fit to the journal as well.

I note author name and sequence changes from prior (2015) description. Assume changes and updates in references as well. None of these changes highlighted in red? Assume editor(s), journal publishers and authors will sort this? To check to ensure full proper author list for citation and confirm fidelity of references list to references cited in narrative?

This reviewer has a small conceptual worry about the automatic attribution of all annual variations to, by definition, $S_{LAND}$. Any annual budget calculates two emission terms, measures the atmospheric concentration, and calculates with reasonable accuracy the ocean sink, $S_{OCEAN}$. $S_{LAND}$, as the residual and least tightly constrained (plus / minus 0.9 GtC, compared to at worst 0.5 GtC for any other term) factor automatically sweeps up all the annual variation. Thus a statement such as in the abstract lines 31 and 32 "is expected to be near record-high because of the smaller residual terrestrial sink ($S_{LAND}$) in response to El Nino conditions in 2015-2016" becomes a bit circular? In crude terms, if we assign all unexplained variability to $S_{LAND}$ as a default, then $S_{LAND}$ necessarily always becomes the preferred reason for annual variability? I suppose the authors argue correctly that reliable and persistent uncertainty estimates on the other four terms of the budget justify this assignment to $S_{LAND}$, but this reviewer wonders if we have missed something, or if we conveniently rarely ask the question of what else we might have missed based on the convenient option to always assign an imbalance to $S_{LAND}$? The authors correctly provide several hints about other factors (fugitive CH4 emissions or lateral fluxes across coastal boundaries mentioned on page 5, nutrient-dependent changes in ocean carbon cycling mentioned on page 26, etc.) but seem confident in discarding those factors. Later statements highlighted this concern, especially that the authors report medium confidence for both the ocean sink and land sink but assign all explained variability to land sink. I note that section 2.7 does a very good job of addressing these issues, but strangely again ends mostly in affirmation of prior assumptions. In addition to the quantitative summaries of uncertainties in the overall budget terms, we need some small positive or cautionary sentence or two about what we know and where we remain most vulnerable to what we don't know?

Page 14, line 27: "assuming a 2% improvement in coal energy content". Do the authors mean a revision in the thermodynamic energy produced by coal consumption, e.g. kjoules per kg burned, or a change in the carbon intensity of the Chinese economy as discussed earlier in this paragraph, those changes due to socioeconomic factors?

Page 38, lines 8 and 9: "to improvements in energy content of coal at the top of the range." Same issue as above? Does this imply a change, perhaps a deliberate change, in the thermodynamic energy content per kg of coal burned, or a social change in the efficiency of using that coal energy? I do not understand "at the top of the range"?

Section 3.2.2, especially page 39: As I remember Betts et al. 2016 (cited elsewhere in this paper but apparently quite relevant here as well), who primarily relied on CO2 extrapolations based on global average SST rather than the careful budget accounting as reported here, estimated 1 ppm (with a large uncertainty?) increase on top of the 2.1 ppm expected annual CO2 increase assignable to ENSO conditions and processes. But later this paper reports increase of 2.1 ppm for just the first 6 months of 2016? A final accounting of increased $G_{ATM}$ for 2016 will certainly hit 3.0 but plausibly might hit closer to 3.5 ppm? Can we really assign that all to change in land surface sources and sinks caused by ENSO? For example Betts et al mention large-scale fires in Indonesia having an impact of 'only' 0.2 ppm. Should we, would we not have already observed a land disruption of sufficient magnitude to cause a CO2 increase 1 or 1.5 ppm?

Page 41, line 7.  Up to this point the reader has encountered many CO2 concentration reports as ppm or GtC, always on annual or even decadal time scales.  Here for the first time we encounter a 6-month estimate.  The authors could help the readers with a small adjustment to make that point earlier and clearer in the sentence?  E.g "*in the 6 month period* between December 2015 and June 2016 was already 2.1 ppm (Dlugokencky and Tans, 2016) after seasonal adjustment".

Page 41, lines 10 to 17.  This represents a good and necessary discussion! But in this section the authors have abandoned all confidence limits or uncertainty estimates.  Perhaps a concluding sentence here, about the medium or low confidence assigned to this assumption (that all 2016 changes will occur due to land surface process and none to ocean or other uncertainties) would satisfy my earlier concern about where the cumulative uncertainties leave us vulnerable to mis-interpretation.  E.g the substantial uncertainties in net land and ocean CO2 fluxes in NH in recent years (Figure 8)?

---

## Referee Report (RR2)

**Date:** 17 October 2016 at 17:11
**Topic:** ESSDD Carbon Budget 2016

The authors present an update of the "global carbon budget" through 2015 and a forecast for the year 2016 primarily based on the methodology used in the assessments in the previous years. Clearly the main data set provided by this study is robust and valid, as witnessed by relatively minor changes in the main budget terms during the present and past updates. It is an important data resource to Earth system science and a timely update. Since there is no real "new" science in this manuscript, I am still puzzled why the authors require ESSD to abolish its open review process for this manuscript.

As last year just have a few minor comments regarding presentation, clarification and documentation as given below. The manuscript still contains a few mistakes also in the unchanged text, which were not caught in last year's edition.

p5 line 9: correct would be "which we convert to units of carbon mass per year"

p25 line 18: The reference to Zeng et al. 2005 can't be right here. A reference to a paper by Gruber or Wanninkhof might be appropriate here.

p26, line 24. better: "… lead to an increase of the ocean sink of up to …."

p29, line 21. better: "The standard deviation of the annual CO2 sinks across…"

p31, line 28-30. both formulas have a sign error since E_LUC and S_LAND are defined as positive quantities. Should probably be written as S_LAND - E_LUC and S_LAND + S_OCEAN - E_LUC in order to be consistent with Figure 6.

p33, line 12. The way the > sign is shown here is ambiguous. Does this mean that the error in F_HO is larger than 0.05 GtC/yr?

p33, line 21. Why is the error of E_LOAC now 0.3 GtC/yr? In line 9 above it was 0.2 GtC/yr.

p34, line 11. missing % sign after "91"

p36, line 1. Same sign error as on page 31

p39, line 19-20: incomplete sentence

p41, line 4: why is here specified r^2? Earlier correlations were stated using r.

p64, Table 6: a reference to the DLEM model is missing

p67, Table 8 and Figure 2: Perhaps a minor point, but if we do exact science, as last year, I claim that the uncertainty of the residual land sink for the decade 2006-2015 rounded is 0.9 GtC/yr and not 0.8 as stated here:

The emission numbers in the spreadsheet unrounded average to 9.3184 of which 5% is 0.4659 GtC/yr.
The atmospheric growth rate uncertainty (using the formula on page 24, line 18) is 0.0495 ppm/yr = 0.1049 GtC/yr using the 2.12 scaling factor from ppm to GtC. The uncertainties for the ocean sink and the land use emissions are each 0.5 GtC/yr (stated in the text and in the spreadsheet).

Thus: $\sqrt{0.4659^2 + 0.1049^2 + 0.5^2 + 0.5^2} = 0.853$ GtC/yr which rounds up to 0.9 GtC/yr.

p72, Figure 1: Include a thin dashed horizontal line at 400 ppm, since this threshold is prominently mentioned in the text.

p75, Figure 4: It's a pity that there is so much white space in these panels. I understand that the comparability between the curves is to be ensured by having the same GtC/yr per cm on the vertical axis. It seems to me that all graphs could be plotted with a vertical range of 9 GtC/yr instead of 12. If the fossil fuel emissions in panel (a) really have to start at 0 GtC/yr, then it could be made a bit larger in the vertical direction, so that the scale is preserved.

p78, Figure 6: For the casual reader who only looks at the graphics, the caption should explicitly mention that the fossil emissions (which also originate on land) are not included here. The term "Atmosphere-land CO2 flux" is not correct for the top panel. Perhaps at least explicitly write in the top panel "Land-use change emissions".

p81, Figure 8: There's again a sign error in the formula in the figure caption.

---

## Author Response (AR2)

Review global carbon budget ESSD-2016-51
Reviewers' comments are in blue
Author's reply are in black

**Reviewer 1**

Overall a very strong product with substantial scientific and political impact. All data well organised and easily and freely accessible. I applaud the global carbon project for this continuing effort to deliver an accurate annual account of the carbon system. Apparently a good fit to the journal as well.

Thank you for your comments on our manuscript update.

I note author name and sequence changes from prior (2015) description. Assume changes and updates in references as well. None of these changes highlighted in red? Assume editor(s), journal publishers and authors will sort this? To check to ensure full proper author list for citation and confirm fidelity of references list to references cited in narrative?

Indeed the author list and order changes every year to reflect the contributions to the specific years. This is handled with Endnote in the manuscript, which keeps track of paper versions. References to previous updates have been checked at submission, and will be checked again during proofs. The text in red in the submitted version was used for changes that affect the data, methods or their interpretation. The *updates* are not generally indicated in red for clarity of the text because there are too many.

This reviewer has a small conceptual worry about the automatic attribution of all annual variations to, by definition, SLAND. Any annual budget calculates two emission terms, measures the atmospheric concentration, and calculates with reasonable accuracy the ocean sink, SOCEAN. SLAND, as the residual and least tightly constrained (plus / minus 0.9 GtC, compared to at worst 0.5 GtC for any other term) factor automatically sweeps up all the annual variation. Thus a statement such as in the abstract lines 31 and 32 „Äúis expected to be near record-high because of the smaller residual terrestrial sink (SLAND) in response to El Nino conditions in 2015-2016‚Äù becomes a bit circular? In crude terms, if we assign all unexplained variability to SLAND as a default, then SLAND necessarily always becomes the preferred reason for annual variability?

The reviewer is correct that assigning all the variability to $S_{LAND}$ is problematic in some cases because we assign all unexplained fluxes on land. We address this comment in depth just below. For the specific example mentioned in the abstract though we think it is legitimate to mention specifically $S_{LAND}$. This is because several different lines of analysis point to a smaller $S_{LAND}$ during El Niño events. First, the anthropogenic emissions cannot account for this growth rate as we show in our paper, so the anomaly must be caused by the ocean and/or land sinks. Ocean data show that the ocean $CO_2$ sink usually increases in response to El Nino, with a magnitude of change of a few tenths of GtC that could not account for the atmospheric variability even if the sign was different. Second, DGVMs, including those used here over year 2015, respond to El Niño by reducing their $CO_2$ sink intensity because of the high temperatures and lower rainfall over tropical land. Thus $S_{LAND}$ is by far the most likely explanation for the projected high growth in atmospheric $CO_2$ in 2016. We have modified the sentence to highlight the link with the end of the El Niño in 2015, and toned down the expectation from 'near record-high' to 'relatively high' to reflect the addition 2 months of data that we were able to include in the 2016 projection.

The new sentence now reads: "In spite of an unchanged $E_{FF}$ in 2016, the growth rate in atmospheric $CO_2$ concentration is expected to be relatively high because of the persistence of the smaller residual terrestrial sink ($S_{LAND}$) in response to El Niño conditions of 2015 - 2016."

I suppose the authors argue correctly that reliable and persistent uncertainty estimates on the other four terms of the budget justify this assignment to SLAND, but this reviewer wonders if we have missed something, or if we

conveniently rarely ask the question of what else we might have missed based on the convenient option to always assign an imbalance to SLAND? The authors correctly provide several hints about other factors (fugitive CH4 emissions or lateral fluxes across coastal boundaries mentioned on page 5, nutrient-dependent changes in ocean carbon cycling mentioned on page 26, etc.) but seem confident in discarding those factors. Later statements highlighted this concern, especially that the authors report medium confidence for both the ocean sink and land sink but assign all explained variability to land sink. I note that section 2.7 does a very good job of addressing these issues, but strangely again ends mostly in affirmation of prior assumptions. In addition to the quantitative summaries of uncertainties in the overall budget terms, we need some small positive or cautionary sentence or two about what we know and where we remain most vulnerable to what we don‚Äôt know?

This is a very important comment. We have addressed it by adding a figure (Figure 10) that shows the carbon that is currently not accounted for through our quantitative analysis of each component separately. We also completely re-wrote one of the paragraphs of the discussion to emphasise this unaccounted carbon instead of emphasising the accounted carbon. We hope this change will give the appropriate visibility to the comment of the reviewer and help trigger research to resolve the missing carbon. The figure and new paragraph are copied here:

"Our capacity to constrain the global carbon budget can be evaluated by adding the five components of Equation (1) using DGVM estimates for $S_{LAND}$, thus using largely independent estimates for each component (Figure 10). This residual global budget represents all the carbon unaccounted currently. Figure 10 shows that the mean global residual is zero, and there is no trend over the entire time period. However it also highlights periods of multiple years where the sum of the estimates differs significantly from zero. These include an unaccounted flux from the surface to the atmosphere (or under-estimated emissions) during 1973-1979 and 1997-2001 and an unaccounted sink from the atmosphere to the surface (or over-estimated emissions) during 1961-1965 and 1990-1992. This unaccounted variability could come from errors in our estimates of the five components of Equation (1; Li et al. 2016), or from missing factors in the Global Carbon Budget, including but not limited to those discussed in Section 2.7. This unaccounted variability limits our ability to verify reported emissions and limits our confidence in the underlying processes regulating the carbon cycle feedbacks with climate change. "

[Figure]

**Figure 10.** Unaccounted carbon in the global carbon budget (GtC $yr^{-1}$), calculated as the sum of $G_{ATM}$ plus $S_{OCEAN}$, minus $E_{FF}$ and $E_{LUC}$ as described in Figure 4, plus $S_{LAND}$ as estimated with DGVM models as in Figure 6b. The uncertainty is the annual uncertainty as described in the text added in quadrature. Positive values indicate an unaccounted surface-to-atmosphere flux of $CO_2$ or an under-estimation of the emissions.

Page 14, line 27: "assuming a 2% improvement in coal energy content". Do the authors mean a revision in the thermodynamic energy produced by coal consumption, e.g. kjoules per kg burned, or a change in the carbon intensity of the Chinese economy as discussed earlier in this paragraph, those changes due to socioeconomic factors?

The 2% improvement reflects improvements in the quality of the coal used. We clarified to: "assuming a 2% increase in the energy (and thus carbon) content of coal for 2016 resulting from improvements in the quality of the coal used, in line with the trends reported by the National Bureau of Statistics for recent years."

Page 38, lines 8 and 9: "to improvements in energy content of coal at the top of the range." Same issue as above? Does this imply a change, perhaps a deliberate change, in the thermodynamic energy content per kg of coal burned, or a social change in the efficiency of using that coal energy? I do not understand "at the top of the range"?

Yes this is the same process as above. Last year we considered possible improvements between 0 and 2%. The actual improvements were 2%, so at the top of the range considered. This has been clarified as follows: "This is due to lower decline in coal production in the last four months of the year compared to January-August and to improvements in energy content of coal through improvements in the quality of the coal used which were at the top of the range of improvements considered in our projection."

Section 3.2.2, especially page 39: As I remember Betts et al. 2016 (cited elsewhere in this paper but apparently quite relevant here as well), who primarily relied on CO2 extrapolations based on global average SST rather than the careful budget accounting as reported here, estimated 1 ppm (with a large uncertainty?) increase on top of the 2.1 ppm expected annual CO2 increase assignable to ENSO conditions and processes. But later this paper reports increase of 2.1 ppm for just the first 6 months of 2016? A final accounting of increased GATM for 2016 will certainly hit 3.0 but plausibly might hit closer to 3.5 ppm? Can we really assign that all to change in land surface sources and sinks caused by ENSO? For example Betts et al mention large-scale fires in Indonesia having an impact of 'only' 0.2 ppm. Should we, would we not have already observed a land disruption of sufficient magnitude to cause a CO2 increase 1 or 1.5 ppm?

We are unsure what the reviewer is asking. The comment refers to section 3.2.2 (page 39), which analyses the partitioning for year 2015, whereas the numbers seem to be referring to year 2016 projection, which is dealt in section 3.3.2 (page 41). We think the layout will be clear in the final version of the paper and have thus done no change in section 3.2.2 on the interpretation for year 2015. We have addressed the comment related to the El Niño in our first response above and added a new figure to stress that $S_{LAND}$ should perhaps not account for all the uncertainty. We address the comments related to the variability over land and the projection for year 2016 in detail immediately below. We do not attempt here to make our own forecast for year 2016 but rather rely on Betts et al. It may be possible to assess the state of land disruption for the current year (here 2016) using satellite observations but we are not in a position to provide this information for this year, above the mention of fire-based emissions so far in 2016 that is in the text.

Page 41, line 7. Up to this point the reader has encountered many CO2 concentration reports as ppm or GtC, always on annual or even decadal time scales. Here for the first time we encounter a 6-month estimate. The authors could help the readers with a small adjustment to make that point earlier and clearer in the sentence? E.g "in the 6 month period between December 2015 and June 2016 was already 2.1 ppm (Dlugokencky and Tans, 2016) after seasonal adjustment".

Modified as suggested. We also updated this paragraph as we now have 8 months available, and added a bit more information on the expectation for the last 4 months not yet observed. The new text reads: "Therefore, the global growth rate in atmospheric $CO_2$ concentration is also expected to be high in 2016. In the 8 month period between December 2015 and August 2016, the observed global growth in atmospheric $CO_2$ concentration was already 2.3 ppm (Dlugokencky and Tans, 2016) after seasonal adjustment, supporting the projection of Betts et al. (2016). Even with a return to El Niño neutral or possible emerging La Niña conditions for the second half of 2016, positive growth in atmospheric $CO_2$ would still be expected during the last 4 months of the year because of the continuing persistent emissions. For example, during the transitions from El Niño to La Niña of 1986-1987, 1998-1999, and 2010-2011, atmospheric $CO_2$ growth of 0.3, 0.6, and 0.9 ppm were observed, respectively, in the last 4 months of the year."

Page 41, lines 10 to 17. This represents a good and necessary discussion! But in this section the authors have abandoned all confidence limits or uncertainty estimates. Perhaps a concluding sentence here, about the medium or low confidence assigned to this assumption (that all 2016 changes will occur due to land surface process and none to ocean or other uncertainties) would satisfy my earlier concern about where the cumulative uncertainties leave us vulnerable to misinterpretation. E.g the substantial uncertainties in net land and ocean CO2 fluxes in NH in recent years (Figure 8)?

We added a mention of the uncertainty as follows: "This is consistent with our understanding of the response of the terrestrial vegetation to El Niño conditions and increasing atmospheric $CO_2$ concentrations, though the uncertainties in $G_{ATM}$ and the partitioning among $S_{LAND}$ and $S_{OCEAN}$ are substantial. "

**Reviewer 2**

In general, the authors have again done a great job in pulling everything together. I have a few comments to make, however. One is general and maybe something for the next version. As it stands the paper generally keeps the older texts and it thus keeps on expanding. Some of the text bits are there because of recent new findings and have become less relevant through time. I would advise the authors next time to check with the editor if some pruning of old text may be allowed and useful. I feel that this will improve the readability.

Thank you for your comments on our manuscript. We take on-board this comment that the manuscript is ever expanding from additions to the updates. For this revised version, we have greatly simplified Section 2.7.2 on 'Anthropogenic carbon fluxes in the land to ocean aquatic continuum', deleted Section 2.2.4 on 'Other published ELUC methods' and instead referred to the 2015 carbon budget publications for other methods. We have also cut one paragraph of the discussion (although it was immediately replaced by a suggestion from Reviewer 1). We will aim to reduce the manuscript more in depth in the next annual update after consultation with the Editor.

Page 5 line 5. Maybe also something for the next version to consider. I appreciate the history of the different carbon communities, but to an outside reader it may appear a bit strange if one talks about atmospheric growth rate and ocean and land sinks. While the units are the same, the public‚Äôs perception is not, and I would think using the term sink (or source) also for the atmosphere would help. Or alternatively one could start using growth rate for the land and ocean stores, but that is probably a bit far fetched.

We slightly modified this paragraph to regroup the partitioning between the atmosphere, ocean and land. This goes a little way to address the reviewer suggestion. Moving away from the $CO_2$ sink needs some reflection within the community.

"The components of the $CO_2$ budget that are reported annually in this paper include separate estimates for the $CO_2$ emissions from (1) fossil fuel combustion and oxidation and cement production ($E_{FF}$; GtC yr$^{-1}$) and (2) the emissions resulting from deliberate human activities on land leading to land-use change ($E_{LUC}$; GtC yr$^{-1}$); and their partitioning among (3) the growth rate of atmospheric $CO_2$ concentration ($G_{ATM}$; GtC yr$^{-1}$), and the uptake of $CO_2$ by the '$CO_2$ sinks' in (4) the ocean ($S_{OCEAN}$; GtC yr$^{-1}$) and (5) on land ($S_{LAND}$; GtC yr$^{-1}$)."

Page 8 line 8. It is good that the new statistics are being used. However, as a reader, knowing that this is a key uncertainty in the budget, I would like to know rather exactly how much the BP estimates differ from the CDIAC ones.

We added the sentence: "The revised emissions are higher by 5% on average between 1990 and 2015 for a total additional emissions of 2.0 GtC during that period (41.3 GtC using the BP statistics and methodology compared to 39.3 provided by CDIAC). The two estimates converge to similar values from 2011 onwards (<2% difference)." (Note this comment refers to page 9 rather than page 8).

Page 32. The new paper by Scwietzke et al (Nature 2016, 538: 88-91) needs to be mentioned. Ideally the consequences for the fossil fuel methane budget need to be propagated through in the CO2 budget.

This is a rapidly moving field and large uncertainties exist in $CH_4$ emissions (see a sister paper on the Global Methane Budget in the discussion phase in ESSD http://www.earth-syst-sci-data-discuss.net/essd-2016-25/). Scwietzke et al 2016 based on new characterization of $^{13}C$ in $CH_4$ sources recently suggested that the total fossil $CH_4$ source should be revised upwards to nearly 200 Tg $CH_4$ $y^{-1}$ instead of 100 $TgCH_4$ $y^{-1}$ as used in this paper to calculate the contribution of anthropogenic $CH_4$ being oxidized to the $CO_2$ growth rate. We do not know the fraction of this new estimate of the fossil $CH_4$ source which is natural geological fossil $CH_4$ (does not contributes to $G_{ATM}$) vs. anthropogenic fossil $CH_4$ from coal, gas and oil production. Taking the conclusion of Scwietzke et al 2016 of a 20 to 60% higher $CH_4$ emission from natural gas, oil and coal production would lead to a revision from 0.06 GtC $y^{-1}$ to 0.07 – 0.1 GtC $y^{-1}$ to the mean contribution of anthropogenic fossil $CH_4$ emissions to $G_{ATM}$. We do not account for this carbon in our budget because it is a small contribution and the uncertainties are large. Even with the revision of the Scwietzke paper the emissions associated with $CH_4$ are still small compared with other uncertainties in the carbon budget. We have added this information in the text as follows to propagate the findings of this new study in our assessment presented in section 2.7.1: "Assuming steady state, these emissions are all converted to $CO_2$ by OH oxidation, and thus explain 0.06 GtC $yr^{-1}$ of the global $CO_2$ growth rate in the past decade, or 0.07-0.1 GtC $yr^{-1}$ using higher $CH_4$ emissions reported recently (Schwietzke et al., 2016)."

Page 34 line 23-24. A change in the mean from 1.5 to 1 GtonC yr-1 is not exactly 'remain constant'. The subsequent paragraph over the high uncertainty then also stands in contrast. I am not convinced that using a DGVM ensemble is te best way to "beat" a satellite based estimate. It would help if the authors can have a look at how to phrase this appropriately. At the moment is reads a bit like "we don't really understand all of this".

It is correct to say that we don't really know if there is a decrease or not. We have clarified the text as follows: "In contrast, $CO_2$ emissions from land-use change have remained relatively constant at around 1.3 ± 0.5 GtC $yr^{-1}$ during 1960-2015. A decrease in emissions from land-use change is suggested between the 1990s and 2000s by the combination of bookkeeping and fire-based emissions used here (Table 7), but it is highly uncertain due to uncertainty in the underlying land cover change data."
The uncertainty is more related to the uncertainty in the land cover change data. The FAO 2015 will help to get a consistent estimation of ELUC over 1960-2015 when it is available. However there will still be uncertainty in the land cover change data which we don't know how to resolve at this stage.

Page 39 line 16. Add 'and la Nina' or refer to the ENSO cycle. The variability goes both ways.

added 'La Nina' as suggested.

**Reviewer 3**

The authors present an update of the 'global carbon budget' through 2015 and a forecast for the year 2016 primarily based on the methodology used in the assessments in the previous years. Clearly the main data set provided by this study is robust and valid, as witnessed by relatively minor changes in the main budget terms during the present and past updates. It is an important data resource to Earth system science and a timely update. Since there is no real 'new' science in this manuscript, I am still puzzled why the authors require ESSD to abolish its open review process for this manuscript.

Thank you for your comments on our manuscript. The reason why we request a closed review process is to ensure when the data is downloaded it is the final published dataset scrutinised by the reviewers. For example this year we have introduced a significant change in the way we calculate China's emissions and a projection for atmospheric growth rate for 2016. If the reviewers had questioned our choice we would not have published

these two new revisions/additions. The role of the reviewers in this process is very important to make our analysis as robust as possible. Note that the reviews will be publicly available when the manuscript is published.

As last year just have a few minor comments regarding presentation, clarification and documentation as given below. The manuscript still contains a few mistakes also in the unchanged text, which were not caught in last year's edition.

p5 line 9: correct would be 'which we convert to units of carbon mass per year'

Modified as suggested.

p25 line 18: The reference to Zeng et al. 2005 can‚Äôt be right here. A reference to a paper by Gruber or Wanninkhof might be appropriate here.

Modified as suggested (replaced with a reference to Wanninkhof, indeed the reference to Zeng was introduced by mistake).

p26, line 24. better: ‚Äú,Ä¶ lead to an increase of the ocean sink of up to ‚Ä¶.‚Äù

Modified as suggested.

p29, line 21. better: ‚ÄúThe standard deviation of the annual CO2 sinks across‚Ä¶‚Äù

Modified as suggested.

p31, line 28-30. both formulas have a sign error since E_LUC and S_LAND are defined as positive quantities. Should probably be written as S_LAND - E_LUC and S_LAND + S_OCEAN - E_LUC in order to be consistent with Figure 6.

Corrected.

p33, line 12. The way the > sign is shown here is ambiguous. Does this mean that the error in F_HO is larger than 0.05 GtC/yr?

This text was deleted in response to a comment from Reviewer 2.

p33, line 21. Why is the error of E_LOAC now 0.3 GtC/yr? In line 9 above it was 0.2 GtC/yr.

This text was deleted in response to a comment from Reviewer 2.

p34, line 11. missing % sign after ‚Äú91‚Äù

Corrected.

p36, line 1. Same sign error as on page 31

Corrected.

p39, line 19-20: incomplete sentence

The sentence was rephrased.

p41, line 4: why is here specified r^2? Earlier correlations were stated using r.

Changed to r (r=0.95).

p64, Table 6: a reference to the DLEM model is missing

Now added the reference.

p67, Table 8 and Figure 2: Perhaps a minor point, but if we do exact science, as last year, I claim that the uncertainty of the residual land sink for the decade 2006-2015 rounded is 0.9 GtC/yr and not 0.8 as stated here:

The emission numbers in the spreadsheet unrounded average to 9.3184 of which 5% is 0.4659 GtC/yr. The atmospheric growth rate uncertainty (using the formula on page 24, line 18) is 0.0495 ppm/yr = 0.1049 GtC/yr using the 2.12 scaling factor from ppm to GtC. The uncertainties for the ocean sink and the land use emissions are each 0.5 GtC/ yr (stated in the text and in the spreadsheet).

Thus: Sqrt(0.4659^2 + 0.1049^2 + 0.5^2 + 0.5^2) = 0.853 GtC/yr which rounds up to 0.9 GtC/yr.

Corrected.

p72, Figure 1: Include a thin dashed horizontal line at 400 ppm, since this threshold is prominently mentioned in the text.

We tried to introduce thick and thin dashed lines to address the reviewer's comment, but unfortunately Matlab (used to produce the figure) does not allow tick lines of varying sizes. We prefer in this paper to keep to a standard figure format for scientific papers but we will provide a version with the 400 ppm line as suggested separately in the powerpoint presentation that will be provided with the data.

p75, Figure 4: It‚Äôs a pity that there is so much white space in these panels. I understand that the comparability between the curves is to be ensured by having the same GtC/yr per cm on the vertical axis. It seems to me that all graphs could be plotted with a vertical range of 9 GtC/yr instead of 12. If the fossil fuel emissions in panel (a) really have to start at 0 GtC/yr, then it could be made a bit larger in the vertical direction, so that the scale is preserved.

We are keen to start emissions at 0 so the reader gets a sense of the emissions prior to 1960. This leaves only modifications in the vertical direction which bring relatively little improvements in readability. Note that the land and ocean sinks are provided separately in panels with less white space, and the full data is provided for users to draw figures that meet their own needs.

p78, Figure 6: For the casual reader who only looks at the graphics, the caption should explicitly mention that the fossil emissions (which also originate on land) are not included here. The term ‚ÄúAtmosphere-land CO2 flux‚Äù is not correct for the top panel. Perhaps at least explicitly write in the top panel ‚ÄúLand-use change emissions‚Äù.

We have clarified the first sentence to: "CO$_2$ exchanges between the atmosphere and the terrestrial biosphere", which we hope makes it clear at the onset what is shown. We also added 'emissions' in the top panel as suggested and changed the label in the middle panel to 'Residual land sink' to clarify the information.

p81, Figure 8: There's again a sign error in the formula in the figure caption.

Corrected.